# Self-supervised predictive learning accounts for cortical layer-specificity

Kevin Kermani Nejad[1,2], Paul Anastasiades [3], Loreen Hertäg [4,5] &
Rui Ponte Costa [1,2,5] ✉

The neocortex constructs an internal representation of the world, but the underlying circuitry and computational principles remain unclear. Inspired by self-supervised learning algorithms, we propose a computational theory in which layer 2/3 (L2/3) integrates past sensory input, relayed via layer 4, with top-down context to predict incoming sensory stimuli. Learning is self-supervised by comparing L2/3 predictions with the latent representations of actual sensory input arriving at L5. We demonstrate that our model accurately predicts sensory information in context-dependent temporal tasks, and that its predictions are robust to noisy and occluded sensory input. Additionally, our model generates layer-specific sparsity, consistent with experimental observations. Next, using a sensorimotor task, we show that the model's L2/3 and L5 prediction errors mirror mismatch responses observed in awake, behaving mice. Finally, through manipulations, we offer testable predictions to unveil the computational roles of various cortical features. In summary, our findings suggest that the multi-layered neocortex empowers the brain with self-supervised predictive learning.

Internal models of the external world are believed to endow the brain with the ability to predict incoming sensory information and select appropriate action-outcome contingencies[1]. Internal models are widely believed to be encoded in the neocortex[2,3], whose hallmark feature is its laminar organization, comprising six distinct layers. Although much has been learned about the underlying cellular heterogeneity and connectivity of individual cortical layers, why the neocortex relies on a multi-layered structure remains unclear[4]. Unraveling its function could shed light on the neocortical algorithms responsible for building rich internal representations of the world.

Historically, it has been proposed that *unsupervised* learning in sensory cortices underpins the development of intricate sensory representations that are critical for driving behavior[5–7]. Self-supervised learning is a form of unsupervised learning that leverages the inherent structure or patterns within the data as the target for learning. A common application of self-supervised learning is to predict the incoming input given past information[8–12]. Importantly, self-supervised learning algorithms learn representations that capture experimentally observed latent representations while resulting in richer models of input statistics[12–16]. However, learning in these models is often treated as a black box; therefore, it remains to be determined whether the brain is capable of employing such learning principles.

The traditional view of the neocortical microcircuit postulates a sequential flow of sensory information. In this canonical view, sensory input is relayed via the thalamus to layer 4 (L4) of the neocortex[17,18]. L4 subsequently forwards this information to layer 2/3 (L2/3), which is thought to integrate ascending sensory information with top-down modulatory input from higher-order cortical areas[19–21]. L2/3 in turn projects to layer 5 (L5), which transmits the information to other brain areas (Fig. 1a). However, growing evidence suggests that this model

[1]Centre for Neural Circuits and Behaviour, Department of Physiology, Anatomy and Genetics, University of Oxford, Oxford, United Kingdom. [2]Bristol Computational Neuroscience Unit, Intelligent Systems Lab, Faculty of Engineering, University of Bristol, Bristol BS8 1TH, United Kingdom. [3]Department of Translational Health Sciences, University of Bristol, Whitson Street, Bristol BS1 3NY, United Kingdom. [4]Technische Universität Berlin & Bernstein Center for Computational Neuroscience Berlin, 10115 Berlin, Germany. [5]These authors jointly supervised this work: Loreen Hertäg and Rui Ponte Costa. ✉e-mail: rui.costa@dpag.ox.ac.uk

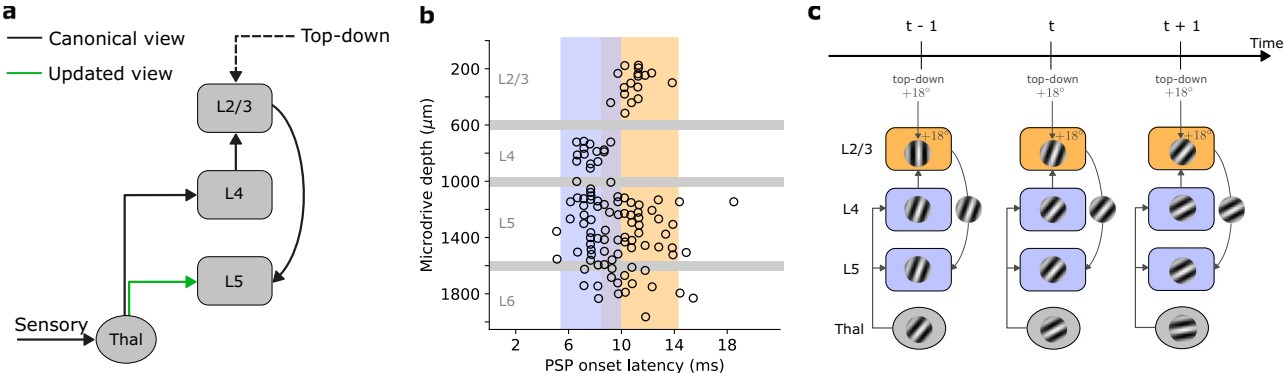

**Fig. 1 | Information flow in neocortical circuits. a** The canonical and updated view of the neocortical microcircuit. Sensory input is initially processed by the thalamus, which, in the classical view, exclusively targets layer 4 (L4). L4 subsequently relays this information to layer 2/3 (L2/3). L2/3, in turn, combines L4 input with top-down contextual input that is fed forward to layer 5 (L5). However, recent studies have emphasized the need to update this view due to direct projections from sensory thalamic nuclei to L5 pyramidal cells[26] (green arrow). For the sake of clarity, we omitted feedback connections from the schematic, which in our self-supervised model are responsible for carrying error signals that drive learning (see main text and Methods). **b** Onset latencies of postsynaptic potentials (PSP) by cortical depth. An onset latency of 0 ms denotes the timing of sensory input (whisker deflection).

These results demonstrate the simultaneous activation of L4 and L5 neurons by the thalamus (blue bands), indicating a direct thalamic input to L5, and a delayed activation of L2/3 neurons (orange band). **c** Illustration of information flow of the proposed self-supervised temporal learning in the neocortical microcircuit. L2/3, informed by past sensory input from L4 and top-down contextual input, predicts the current sensory input arriving in L5. The direct thalamic inputs to L5 provide sensory input, which is used as a teaching signal to instruct the L2/3 predictive model. Gabor-like gratings represent neuronal encoding of the sensory input, or its prediction in the case of L2/3-to-L5 connections. Panel b adapted from *Christine M Constantinople and Randy M Bruno. Deep cortical layers are activated directly by the thalamus. Science 340 (2013)*; reprinted with permission from AAAS.

does not capture the full diversity of connections in the neocortical microcircuit[18]. A body of experimental works suggests that L5 pyramidal cells receive direct thalamic input that can drive short-latency, sensory-evoked responses independently of activity within the cortical network (Fig. 1b)[22–26]. These observations imply two distinct sensory-driven pathways within the neocortex, one targeting L4 and the other L5 (Fig. 1a). However, why the cortex requires multiple inputs and the computations supported by such parallel pathways remain unknown.

Inspired by this refreshed view of the canonical microcircuit and the predictive capabilities of self-supervised machine learning algorithms[9,27], we propose a model in which L2/3, informed by past sensory input from L4 and top-down context from higher-order cortical areas, predicts incoming sensory input. In this model, the delay from L4 to L2/3 enables L2/3 to generate predictions based on previous sensory information (Fig. 1c). Direct thalamic input to L5 provides the new sensory information, which serves as an implicit target to compare with the predictions generated by L2/3-to-L5 connections. When the model's predictions are violated, this comparison triggers errors in both L5 and L2/3, thus driving circuit plasticity in a self-supervised manner. This perspective of neocortical circuitry suggests that the L4-L2/3-L5 laminar structure with parallel thalamic innervation enables the brain to learn rich temporal representations.

We first show that our learning rule for L2/3-to-L5 connections closely resembles the long-term synaptic plasticity experimentally observed[28]. Next, we demonstrate that by using self-supervised learning, our model can learn and predict sequential Gabor-like inputs in a context-dependent manner, highlighting its ability to capture structured patterns. By ablating individual components of the model and evaluating their impact on performance, we reveal how the neocortical circuit components collaboratively enable self-supervised learning. Next, we demonstrate that self-supervised learning leads to predictions that are robust to sensory noise and occlusions. Moreover, the model captures the relative differences in sparsity across layers, aligning qualitatively with experimental findings in sensory systems. Additionally, we demonstrate in a visuomotor task that violations of predictions result in layer-specific mismatch errors, consistent with mismatch responses observed in awake, behaving animals. Finally, we suggest a set of optogenetic experiments capable of testing the core predictions of our self-supervised learning model. Collectively, our

findings support the notion that the L4 → L2/3 → L5 pathway is instrumental in enabling the brain to engage in temporal self-supervised learning, highlighting its potential significance in neural mechanisms of predictive learning.

## Results

### Neocortical layers can implement self-supervised predictive learning

To understand how neocortical microcircuits process temporal information and learn latent representations in a self-supervised manner, we created a model that emulates the properties of cortical circuits. Our model contains three subnetworks with nonlinear neurons separated into layers (L2/3, L4, and L5) to reflect the laminar architecture of the neocortex (Fig. 2a). Within this framework, L4 receives ascending sensory information, $\mathbf{x}$, at timestep $t$ through input weights $W_{\text{Thal.}\rightarrow\text{L4}}$. At the same time, L2/3 receives delayed thalamic input via L4 through $W_{\text{L4}\rightarrow\text{L2/3}}$ synapses, as well as top-down contextual input via the weights $W_{\text{top-down}\rightarrow\text{L2/3}}$. We hypothesize that this combination of inputs enables L2/3 to make predictions about upcoming sensory information. In our model, we define the predictions as the output of L2/3, that is, $W_{\text{L2/3}\rightarrow\text{L5}} \cdot \mathbf{z}_t^{\text{L2/3}}$. These predictions are compared with the activity of L5 neurons $\mathbf{z}_t^{\text{L5}}$ ("target") that also receive the actual sensory input at timestep $t$. This comparison results in a self-supervised *error* when L2/3 predictions and L5 targets are not equal, which is defined as $\mathcal{C}_{\text{L2/3}\rightarrow\text{L5}} = \frac{1}{2}(W_{\text{L2/3}\rightarrow\text{L5}} \cdot \mathbf{z}_t^{\text{L2/3}} - \mathbf{z}_t^{\text{L5}})^2$. In our model, this error is fed back via L5-to-L2/3 connections to adjust the predictive model of the incoming inputs. However, using this self-supervised error alone would lead to a degenerate solution, as the model could learn to output a constant value regardless of the input (known as representational collapse[12,29]). To prevent this, in our model, L5, but not L2/3 or L4, is also trained to reconstruct its own input ("reconstruction cost", see Methods). This strategy offers a solution to the representational collapse problem in cortical circuits.

During learning, we modify connections to minimize the cost function and facilitate the encoding of sensory input. Unlike predictive coding approaches[30], we do not use separate representation and error neurons. Instead, each neuron in our model generates activity during the forward pass (L4-to-L2/3-to-L5), which is then used to compute its respective error signals through gradient calculations (Fig. S1; see

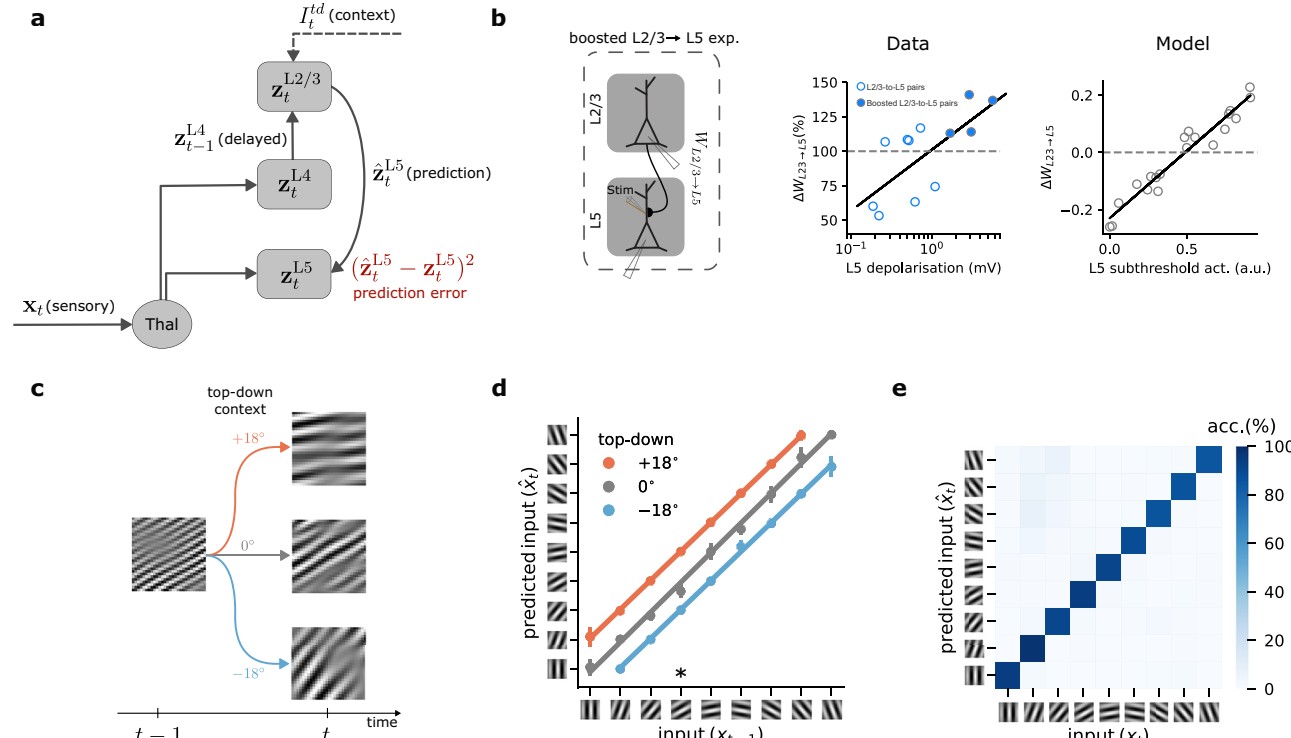

**Fig. 2 | A model of temporal self-supervised learning in cortical circuits.**
**a** Schematic of model of self-supervised learning in cortical layers (black denotes elements used for inference and red for learning). **b** Left: schematic of the experimental setup in which an extracellular electrode was used to boost L5 activity while inducing long-term synaptic plasticity on L2/3-to-L5 connections[28]. Middle: observed changes in synaptic weights as a function of L5 depolarization (scatter plot: individual data points, solid line: linear fit to the data). Right: L2/3-to-L5 learning rule as predicted by our model as a function of L5 activity for multiple randomly drawn samples of L2/3 and L5 activity (circles), and linear fit to the data

points (solid line). **c**, Schematic of a sequential Gabor task. The generative factor provided to the model as top-down context at timestep $t$ determines the orientation of the next Gabor patch at timestep $t + 1$. **d** Decoding accuracy of a linear model trained on the output of L2/3. For a given input, L2/3 predicts the incoming sensory input with high accuracy. Colors represent the three possible conditions (−18°, 0°, and +18°). * points to the example illustrated in (**b**). Error bars represent the standard error of the mean over five different initial conditions. **e** Confusion matrix for classification accuracy of a linear model trained on the output of L5. The matrix is calculated over five different initial conditions.

Methods). Consequently, our model requires both feedforward connections from L4 → L2/3, which relay sensory information, as well as feedback connections from L5 back to L2/3, to transmit self-supervised error signals. All weights are optimized via gradient descent. This approach allows us to recapitulate learning rules observed in experimental data. We compared the learning rule for the weights connecting L2/3 and L5 in our model, $W_{L2/3 \to L5}$ (Methods Eq. (7)), with observed long-term synaptic plasticity in primary sensory cortices (Fig. 2b)[28]. Our learning rule predicts a depression-to-potentiation switch as the activity of L5 neurons increases. This is in line with experimental observations showing a similar depression-to-potentiation switch of $W_{L2/3 \to L5}$ connections with increasing depolarization of L5 pyramidal cells[28]. Hence, this experimental evidence corroborates our model's learning rule, showing that the model is consistent with known synaptic plasticity mechanisms in primary sensory cortices.

To demonstrate our model's ability to learn predictive representations, we created a sequence of Gabor patches, commonly used to evoke responses in the primary visual cortex (see Methods)[31,32]. Starting with a random Gabor patch, the orientation of the patch changes based on randomly generated top-down contextual input (see Methods). This higher-order contextual cue to L2/3 is provided at each time step and conveys contextual information, for instance, an animal's own locomotion in a sensorimotor task, which may be provided by the motor cortex[33]. Given this top-down input, the Gabor patch either rotates anti-clockwise by −18°, remains the same, or rotates clockwise by +18° (Fig. 2c). For example, if the current Gabor patch has 0° orientation, the orientation of the subsequent input can be −18°, 0°,

or +18° for contextual cue values of −18°, 0°, and +18°, respectively (Fig. 2c).

To evaluate the representations learned by our model, we use a linear decoder[8,34,35]. We trained this decoder on L2/3's output to predict the orientation of the Gabor patches received by L5 at any given timestep. L2/3 output effectively learns to predict upcoming Gabor patches with near-perfect accuracy (93%) using the previous input, provided by L4, and the top-down context value (Fig. 2d). Next, we applied a linear classifier to L5's output. On average, L5 achieves a test accuracy of 89% (classification accuracy on a random model is 11%), indicating that L5 successfully identifies and encodes each Gabor patch's distinct features (Fig. 2e). These results directly follow from our model, where L5 encodes the input while L2/3 predicts the incoming input, which are in line with experimental findings showing that L2/3 can learn to predict image sequences in a passive task[32]. In addition, we obtain similar results in a more complex task in which hand-written digits are used as input (Fig. S2).

In summary, we have shown the model's ability to perform self-supervised learning in a temporal task and its consistency with synaptic plasticity observations. However, we have yet to explore the precise contribution of each circuit element to self-supervised learning.

## Neocortical circuitry jointly underlies self-supervised learning
In our model, different cortical layers give rise to distinct computational roles. To demonstrate the contribution of different circuit components, while generating experimentally testable predictions, we systematically ablated individual connections, allowing us to quantify

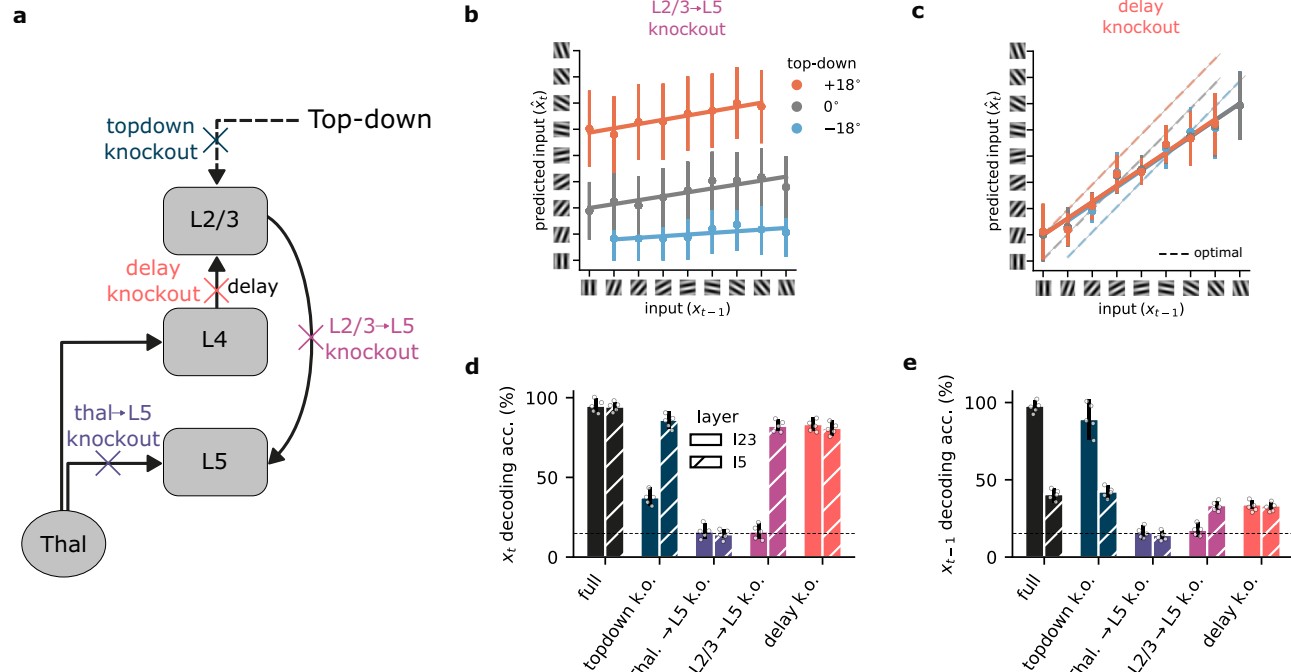

**Fig. 3 | Neocortical circuitry jointly enables self-supervised learning.**
**a** Schematic of the model with individual components knocked out (colored crosses) within the neocortical microcircuit architecture. **b** Connections from L2/3 to L5 are necessary for L2/3 to learn a predictive representation of the input. **c** Impact of L4-mediated delay in self-supervised learning (dashed lines represent the optimal prediction). **d** Summary of decoding accuracy of the current input for L2/3 and L5 when specific connections are knocked out. The x-axis indicates the specific ablation, while the y-axis indicates the decoding accuracy for the current input ($x_t$). **e** Similar to (**d**), but for the past input ($x_{t-1}$). Knockout components in (**d**, **e**) are color-coded as in (**a**). Horizontal dashed lines in **d**, **e** represent chance decoding accuracy. Error bars represent the standard error of the mean over five different initial conditions.

their impact on both representational capability and performance (Fig. 3a).

When we knocked out the L2/3-to-L5 connection, L2/3 could no longer learn to predict upcoming sensory information (Figs. 2c vs. 3b). However, due to the continued presence of top-down contextual input, L2/3 still displayed contextual segregation. Note that there are also implicit feedback connections from L5-to-L2/3, which mediate learning, and which we study in the next section.

Our model proposes a key function for the delay introduced by L4 as information propagates to L2/3. This delay creates a temporal discrepancy between the information available to L2/3 (past input) and L5 (present input), which enables L2/3 to anticipate the incoming sensory input. When this delay is removed, which would be equivalent to L2/3 receiving direct thalamic input, the entire network operates on incoming sensory input, i.e., at timestep $t$. Consequently, the network can no longer generate meaningful predictions of future inputs. This is evident in our model, where removing the delay rendered L2/3 unable to reliably distinguish between potential future outcomes (Fig. 3c). This result highlights the key role that temporal delays may have in shaping predictive learning within the neocortical microcircuit. The L4 to L2/3 delay is essential for biasing L2/3 representations toward the future. Without it, both the Thal. → L4 → L2/3 → L5 and Thal. → L5 pathways end up representing the current sensory input, causing the former pathway to be redundant.

Next, we investigate how ablations of these different circuit elements affect the ability to decode current sensory information from both L2/3 and L5 representations. For current input decoding (Fig. 3d), L5 demonstrated robust accuracy as long as it retained access to thalamic sensory input. This aligns with its role as the primary recipient of sensory data, together with L4[23,26,36]. L2/3 accuracy, however, was more dependent on the overall circuit properties. While top-down input to L2/3 provided useful context-dependent input (Fig. S3), any disruption to the

core pathways within the microcircuit, except the delay knockout, compromised L2/3's ability to represent the current sensory input.

Decoding the previous input (Fig. 3e) further differentiated L2/3 and L5. As anticipated, L5 exhibited limited information about previous inputs due to its exclusive focus on current thalamic information. L2/3, however, encodes information about the past as a result of the delay introduced by L4. Complete loss of this past-input representation occurred only in two scenarios: when critical learning pathways were ablated (Thal. → L5, L2/3 → L5), or when removing the delay synchronized the inputs to L2/3 and L5.

Finally, we demonstrate the importance of the cost functions used in our model, specifically the role of the L5 reconstruction cost and the L2/3 self-supervised cost. Removing the L5 reconstruction loss leads to representational collapse[12,29], a state where L2/3 and L5 converge to similar outputs regardless of input (Fig. S5). While adding a reconstruction loss effectively prevents representational collapse, other mechanisms, such as variance maximization[37], have also been successful. Replacing the L2/3 self-supervised cost function with a simpler regression task impairs L2/3's predictive ability, highlighting the importance of an appropriate self-supervised objective (Fig. S5).

Our ablation and decoding analyses suggest that predictive learning within the neocortical microcircuit depends on a complex interplay between the layers. While the L2/3-to-L5 connection is essential for the model to learn predictive representations, the temporal delay between L4 and L2/3 is crucial for generating future-oriented predictions, but not current representations. In terms of decoding past and present sensory input, our results demonstrate that L2/3 specializes in representing temporal context, while L5 primarily encodes immediate sensory information. This result aligns with the experimental observations showing that L2/3 effectively encodes temporal information with high precision[38].

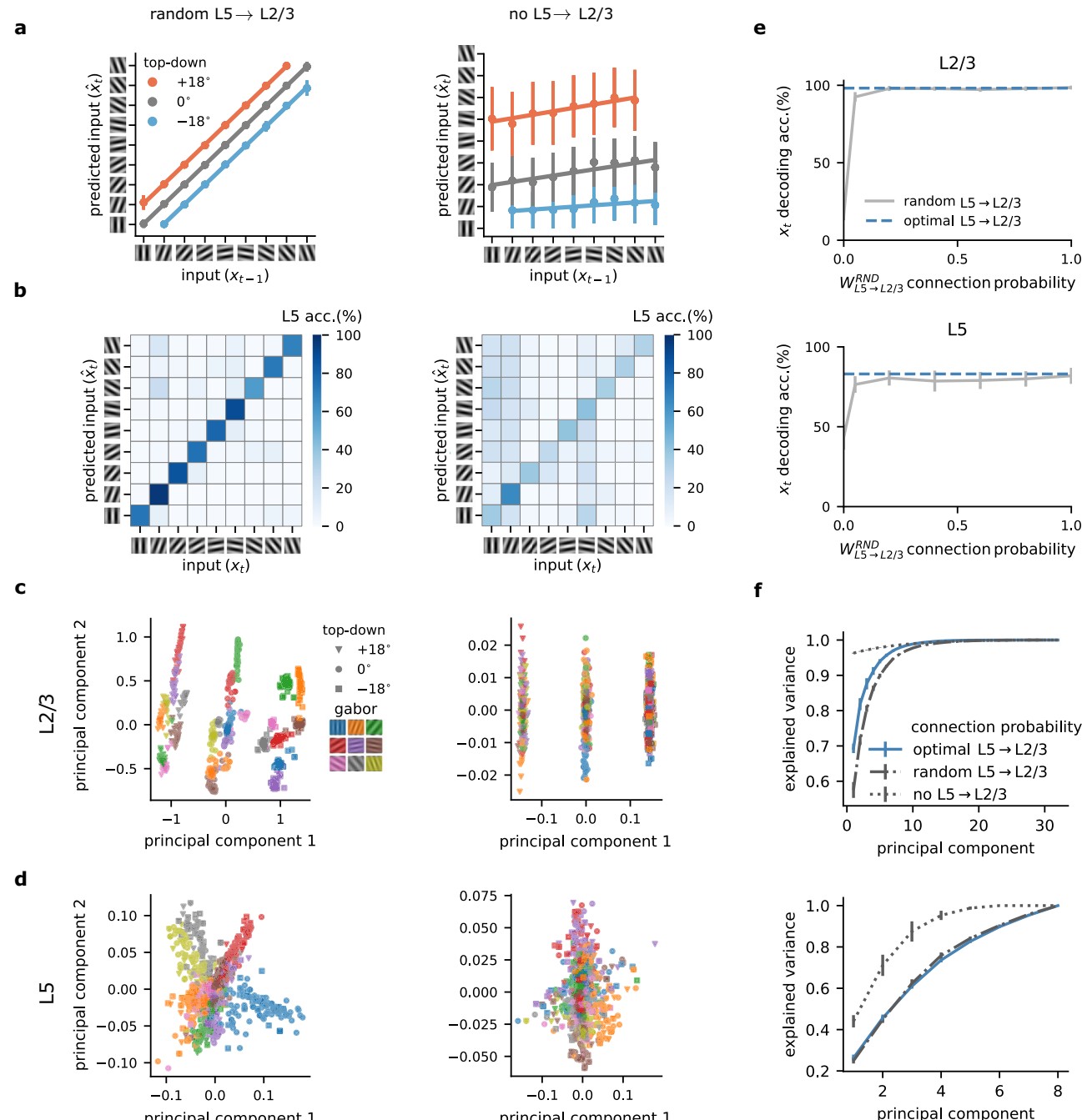

**Fig. 4 | Role of L5-to-L2/3 feedback connections in self-supervised predictive learning. a** L2/3 learns to predict the input in the presence of random feedback (left) but fails to do so without L5-to-L2/3 feedback (right). **b** L5 learns to represent the inputs accurately with random feedback (left) but shows lower decoding accuracy without feedback (right). **c** Two main principal components of L2/3 representations for random (left) and no feedback (right) across different top-down contexts (symbols) and input Gabor orientations (colors). **d** Two main principal components of L5 representations for random (left) and no feedback (right) across different top-down contexts (symbols) and input Gabor orientations (colors). **e** L2/3 (top) and L5 (bottom) decoding accuracy for different degrees of L5-to-L23 feedback. **f** Explained variance of L2/3 (top) and L5 (bottom) learnt representations. Error bars represent the standard error of the mean over five different initial conditions.

## L5 → L2/3 feedback is required for self-supervised learning

A cardinal feature of self-supervised learning models is that they require an error or teaching signal to guide plasticity across the network. This error signal drives adjustments in synaptic weights, refining the network's predictive capabilities. In our model, the learning-driving error signal originates in L5. Hence, this error signal must be fed back to L2/3 to refine the predictive model. This suggests the need for a feedback connection that propagates this information from L5 to L2/3. Although the vast majority of work on neocortical circuits has disregarded

feedback connections from L5 to L2/3 pyramidal cells[17,18], growing evidence shows that they are more abundant than previously assumed[39,40].

Here, we explored the importance of the L5 to L2/3 feedback connection for learning in our model. In particular, we contrast optimal feedback, as used in previous figures, with random and no feedback. The optimal feedback condition corresponds to a setting in which the feedback weights mirror the feedforward weights (i.e., $W_{L5 \to L2/3} = W_{L2/3 \to L5}^T$), whereas in the random feedback condition, the feedback weights are set to a random weight matrix[41].

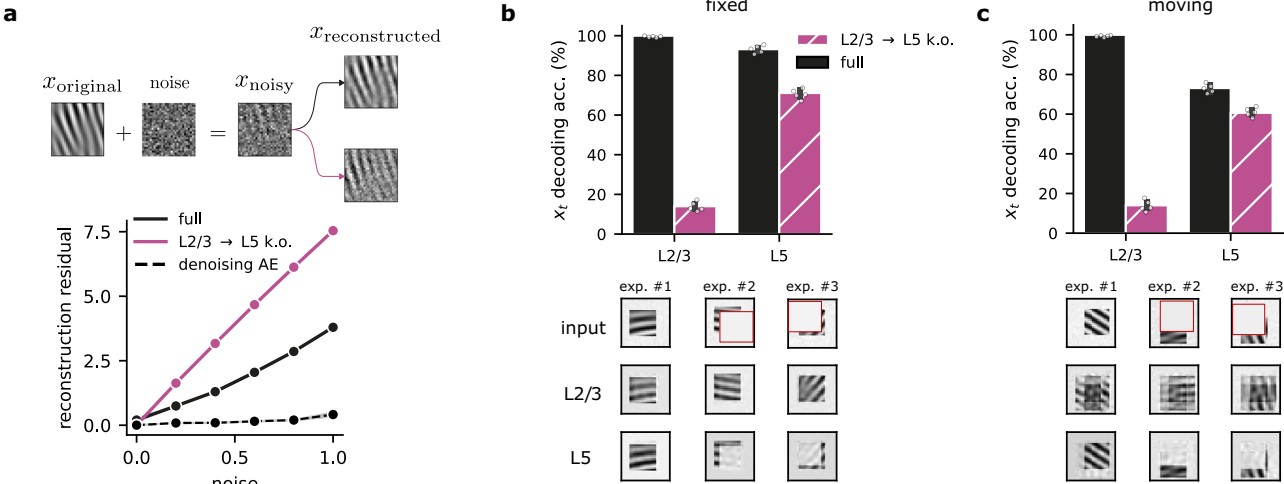

**Fig. 5 | L2/3-to-L5 predictions are crucial for denoising and resolving occluded stimuli. a** L2/3-to-L5 connections promote noise suppression in L5 representations. Top: Schematic of noise added to the original inputs. Bottom: Noise-corrupted input samples lead to higher L5 reconstruction residuals ($\hat{x}_t - x_t$) when the L2/3 → L5 pathway is ablated (purple) compared to the full model (solid black). We also provide the reconstruction residual for an autoencoder that was explicitly trained to denoise the input (dashed black line). **b** Top: Decoding accuracy with and without L2/3 → L5 for a Gabor task with occlusion. Bottom: Three examples depicting L2/3's ability to recover occluded information, compared to L5's incomplete reconstructions (top row: original occluded input; middle row: L2/3 prediction; bottom row: L5 reconstruction). **c** Top: Accuracy with and without L2/3 → L5 connections for a task in which Gabor patches move randomly. Bottom: Examples illustrating the robustness with moving Gabor patches (top row: original input with motion (cf. panel b); middle row: L2/3 prediction; bottom row: L5 reconstruction). L2/3 reconstruction encodes uncertainty about the future possible input location. Error bars represent the standard error of the mean over five different initial conditions.

Inspired by work showing that random feedback weights are sufficient for credit assignment[41], we tested whether this form of unstructured feedback was sufficient for L2/3 to learn. We observed that L2/3 output was indeed able to learn to predict the input with random feedback weights (Figs. 4a and 2c). These results suggest that unstructured feedback may be sufficient in enabling L2/3 output to develop useful predictive representations. Furthermore, our findings demonstrate a significant drop in the decoding accuracy of L5 when feedback connections from L5-to-L2/3 were removed (Fig. 4b). This decline is due to L2/3's inability to learn, causing L5 to adopt erroneous representations influenced by L2/3's unlearned state. These results demonstrate the need for L5 → L2/3 feedback.

To study the neuronal representations learned by the different layers, we analyzed the two main principal components of L2/3. This revealed a notable difference in the structural organization across feedback conditions (that is, random and no feedback). With feedback, L2/3 representations were differentiated based on the identity of the Gabor patch as well as the top-down context (Fig. 4c, left). Without feedback, the L2/3 representations were only distinguished based on top-down inputs, indicating a limitation in the network's learning capability (Fig. 4c, right).

Similarly, analysis of L5 revealed that with random feedback, representations are organized according to Gabor patch features, suggesting a structured learning process (Fig. 4d, left; similar to the optimal feedback condition, Fig. S6). These observations are in line with the increased sparsity we observed in L2/3 compared to L5 (see below). In contrast, when feedback was absent, L5 representations were less organized (Fig. 4d, right).

Classically, feedback connections within the neocortex occur at lower probabilities than the corresponding feedforward pathway[39,40]. To test how connection density influences the properties of the network, we next explored how the linear decoding accuracy of both L5 and L2/3 varies with the probability of feedback connections from L5 to L2/3. An increase in connection probability corresponded to enhanced decoding accuracy (Fig. 4e). Therefore, while a very low feedback connection probability was sufficient for learning the task considered here, for more complex tasks, higher connection probabilities may be required for optimal performance (Fig. S7).

Finally, to determine how distributed information was, we examined how the explained variance, as assessed by the number of principal components (PCs), changed with varying feedback probabilities. The absence of feedback required a greater number of PCs to explain the data effectively, while random feedback closely mirrored the efficiency of optimal feedback connections (Fig. 4f). This increase in the number of PCs to capture the same variance is consistent with our findings, showing the importance of feedback in organizing sensory information in superficial layers (Fig. 4c, d).

This analysis highlights the crucial role of feedback connections in neural networks, particularly in improving predictive capabilities and structuring neural representations. The nuanced differences observed across various types and intensities of feedback offer insight into the role of L5-to-L2/3 feedback connections[39,40] for learning and information processing in cortical networks.

## Self-supervised learning leads to robustness to noise and occlusion in cortical networks

As a consequence of learning a robust predictive model of the sensory input, the cortical network should disregard unpredictable aspects like noise. Therefore, we hypothesized that the self-supervised L2/3 → L5 component would help L5 filter out sensory noise. To test this, we ablated the L2/3 → L5 projection in our model and used input with varying noise levels. The simulated ablation shows that removing the self-supervised component dramatically reduces robustness to different noise levels (Fig. 5a; similar results are obtained when ablating the L4-to-L2/3 delay, Fig. S4). This denoising capability naturally emerges from the model's design, despite not being explicitly designed for this purpose, making it comparable to a near-optimal denoising autoencoder network.

To further investigate the model's robustness to input perturbations, we tested its ability to reconstruct input patterns during partial input occlusions. After training the model without occlusions, we evaluated the robustness of the learned representations in each layer

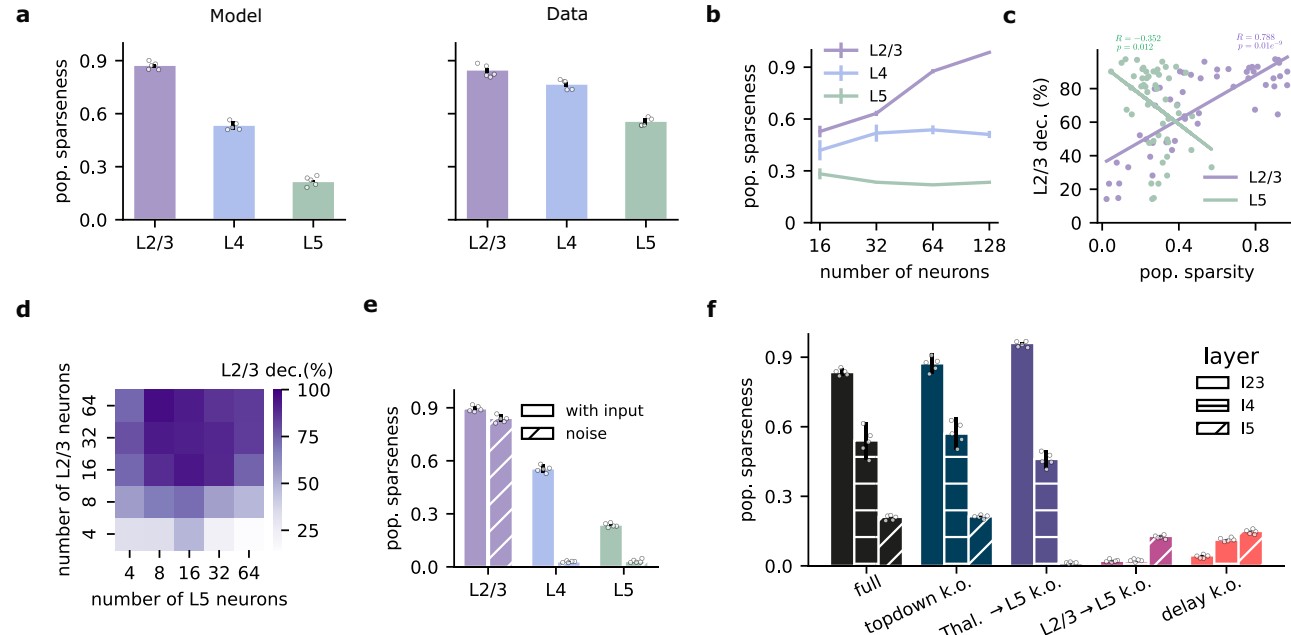

**Fig. 6 | Population sparseness depends on the neocortical layer. a** Population sparseness across layers in the model (left; full histogram in S8) and experimental data[47] (right). **b** Population sparseness as a function of the number of neurons. The qualitative relationship between layers is preserved, but L2/3 sparseness increases with network size. **c** Decoding accuracy of current input as a function of the population sparsity of L2/3 and L5 (L2/3: $r = 0.78$, $p = 2.1e{-}11$; L5: $r = -0.35$, $p = 0.01$). **d** L2/3 decoding accuracy of the current input as a function of the number of neurons in L2/3 and L5. **e** Population sparseness with or without sensory input (noise condition) after learning. L2/3 remains sparse, while L4 and L5 show a strong reduction in response sparsity. **f** Population sparseness following the ablation of various model components during learning. Top-down input ablation slightly increases L2/3 sparseness. Thalamic input ablation to L5 decreases L2/3 sparseness while increasing L5 sparseness. Ablation of L2/3-to-L5 connections abolishes sparseness across all layers. Bars are color-coded following Fig. 3a, d, e. Statistical tests are two-sided, and no adjustments were made for multiple comparisons. Error bars represent the standard error of the mean over five different initial conditions.

by testing their ability to classify sensory input with randomly occluded segments (Fig. 5b). We observed that L2/3 achieves higher decoding accuracy compared to L5, and that L5 decoding remained unaffected by L2/3 → L5 knockout (Fig. 5b, top). Further analysis shows that L2/3 can fully reconstruct the input while L5 is only able to reconstruct the observable parts of the input (Fig. 5b, bottom). These results support the idea that a strong predictive model leads to representations that are robust to several perturbations.

Finally, we explored whether L2/3 can also encode the uncertainty about the possible input locations. To test this, we introduced random shifts in the position of the Gabor patch on the blank canvas during the task. Decoding performance remained similar to the fixed-position task(Fig. 5c, top), but reconstructions were different. L2/3 representations reflect the input's positional uncertainty (blurred reconstructions across possible locations), while L5 again encodes only the visible parts (Fig. 5c, bottom). This suggests that self-supervised learning also leads to useful L2/3 representations in the presence of sensory uncertainty.

Collectively, these results underscore the robustness exhibited by the proposed neocortical predictive learning model across diverse input conditions. Consequently, our model offers valuable insights into the mechanisms through which cortical circuits deal with the considerable variability inherent in naturalistic environments.

## Layer-specific sparseness emerges from self-supervised learning

Sparse coding, in which only a small subset of neurons are strongly active for a given stimulus, is a widespread phenomenon across the neocortex[42–46]. This sparsity is particularly pronounced in superficial layers (L2/3) compared to deeper layers (e.g., L5)[47]. However, it is unclear why the degree of sparsity varies across cortical layers and how it relates to their computational role.

We wondered whether our network, equipped for temporal self-supervised learning, could reproduce experimentally observed

sparsity distributions. Moreover, we wanted to investigate how different network features may control sparsity across different neocortical layers. To this end, we trained our model on the sequential Gabor task. After training, we measured population sparseness across layers using established metrics[48] (see Methods). Interestingly, our results capture qualitative differences in the level of sparsity, showing a trend that is similar to experimental findings[47]: L2/3 presents the highest sparseness, followed by L4 and then L5 (Fig. 6a). This alignment suggests that self-supervised learning, focused on input prediction, could be a key factor driving sparsity in biological neural networks.

Layer 2/3 has undergone rapid expansion relative to other layers within the human evolutionary lineage[49,50]. Could this expansion support greater predictive learning capabilities? We found that L2/3 sparseness increased with network size, while sparsity in L4 and L5 remained relatively stable (Fig. 6b). Consistent with the increased sparsity in L2/3, we find that an increase in the number of L2/3 neurons also results in improved L2/3 decoding accuracy of upcoming sensory inputs (Fig. 6c, d), in line with previous work[51]. In contrast, the relationship between the number of neurons, sparsity, and decoding accuracy was not present in L5 neurons (Figs. 6c, d and S9). These qualitative results do not depend on the type of optimizer and learning rate used (Fig. S10).

To determine whether sparsity is due to the encoding of sensory input or is simply an underlying feature of the circuitry, we trained the network with Gaussian noise instead of Gabor-like inputs. When sensory input was replaced with background noise, L2/3 retained high sparsity, whereas L4 and L5 responses showed a strong decrease in response sparsity (Fig. 6e). These results suggest that L2/3 sparsity is a consequence of learning to predict sensory input (Fig. S9d). One possible explanation is that L2/3 is forced to extract and rely only on the most salient temporal features from the previous input in order to forecast the current latent state, whereas L5 is free to capture the full complexity, and a more distributed representation of the input. In

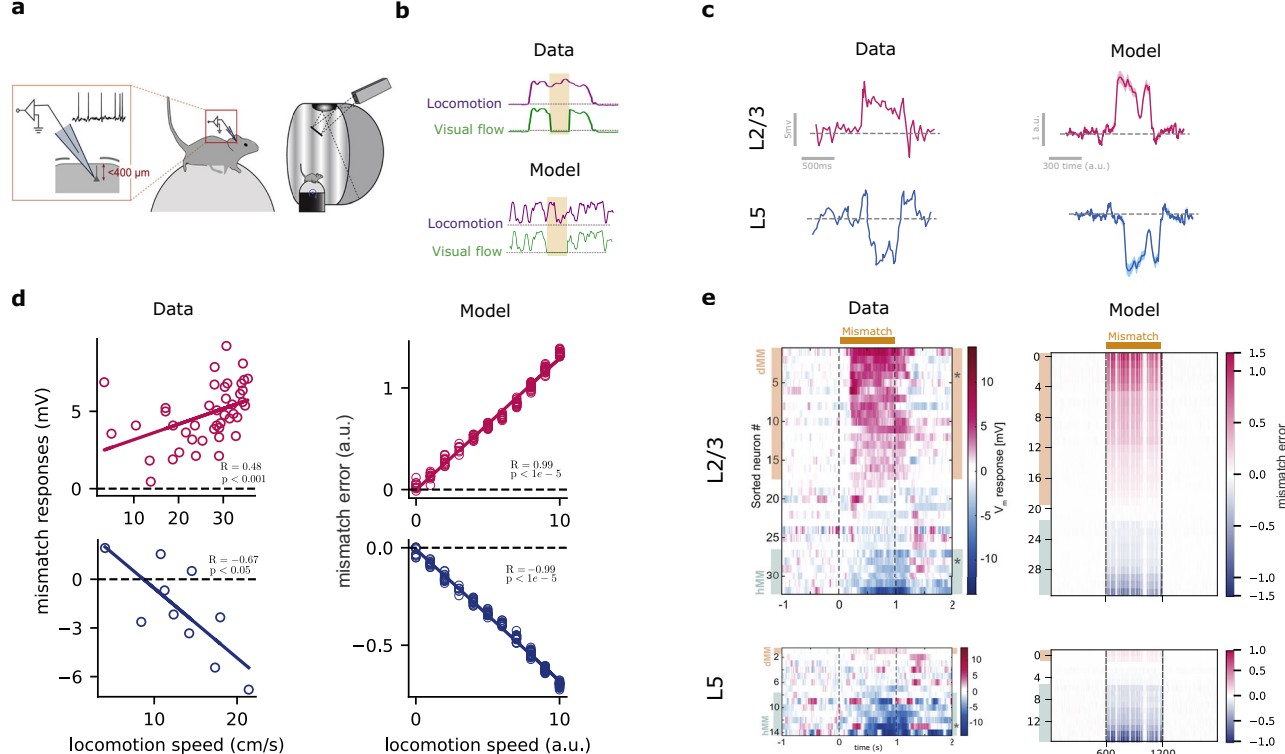

**Fig. 7 | Model generates sensorimotor mismatch prediction errors in line with experimental observations. a** Illustration of visuomotor task used by Jordan and Keller[19] in which mice learned to associate visual flow with locomotion. **b** Sample of training data from experiments (top) and our synthetic dataset (bottom). As in the experimental setup, we randomly halt the visual flow (flat green line) to generate visuomotor mismatches. **c** When the visual flow is halted, a sample neuron in L2/3 of the mouse visual cortex shows depolarization (magenta), while a sample neuron in L5 shows hyperpolarization (blue; left). In our model, in line with the data, L2/3 shows a positive mismatch error, while L5 shows a negative mismatch error

(right). Shaded areas represent the standard deviation over five runs. **d** The mismatch error signals in L2/3 of the model are correlated with the modeled locomotion speed when the visual flow is halted (right), in line with experimental observations[19,52] (left). **e** The model generates a distribution of mismatch errors which are biased towards positive errors in L2/3 and negative errors in L5 (right), in line with mismatch responses observed in primary visual cortex[19] (left). Panels (**a**, **c**–**e**) were partially reprinted from[19], with permission from Elsevier. Statistical tests are two-sided, and no adjustments were made for multiple comparisons. Error bars represent the standard error of the mean over five different initial conditions.

effect, L2/3 is tasked with filtering out the redundant (such as extraneous details or noise) or less informative parts of the previous encoding (by L4), which leads to a naturally sparser pattern of activation.

Next, we ablated different model elements during learning to test their contribution to the emergence of response sparsity (Fig. 6f). Removing top-down input had a minimal effect on sparseness. However, ablating thalamic input to L5 during learning selectively decreases L5 sparseness, likely due to the randomization of L5 responses as a result. Finally, ablating L2/3-to-L5 connections or the delay component completely abolished sparsity across all layers, demonstrating their crucial role in encouraging sparseness over learning.

Overall, these results show that sparsity emerges as a function of input-driven predictive learning as postulated by our model, thus providing an explanation for layer-specific sparsity observed experimentally.

**Model generates sensorimotor mismatch error signals consistent with experimental observations**
Our study demonstrates the ability of our cortical model to predict upcoming sequences. We next sought to investigate the model's response when those predictions are violated and to determine if these responses vary between superficial and deep cortical layers. Additionally, we wanted to test whether our network generates mismatch error signals that resemble those observed in cortical networks of behaving animals[19,33,52].

More precisely, we verified whether our model could reproduce the mismatch responses in both L2/3 and L5, recently observed experimentally[19]. In the study by Jordan and Keller[19] the authors explored the mismatch error responses in a setting where animals learn to couple the speed of the visual flow (that is, sliding vertical gratings) to the animal's locomotion (Fig. 7a). This paradigm allows for the systematic investigation of how neural responses in the primary visual cortex are shaped by the interplay of external sensory stimuli (visual flow) and internal expectations (running speed). Using whole-cell recordings, Jordan and Keller[19] showed that when the visuomotor coupling is temporarily broken (i.e., "visuomotor mismatch"), the majority of L2/3 pyramidal neurons depolarize, whereas a smaller fraction of L2/3 neurons hyperpolarize. In contrast, almost all L5 excitatory neurons hyperpolarize in visuomotor mismatch. We propose that the differences in L2/3 and L5 mismatch responses observed experimentally[19] can be explained in a network implementing self-supervised predictive learning.

To gain deeper insights into how our cortical model responds to violations of expectations, we extended our model to examine its computational principles under visuomotor mismatch conditions. Specifically, we aimed to investigate how the mechanisms underlying self-supervised predictive learning shape responses to unexpected disruptions in sensory input. In our setup, the top-down input to L2/3 was defined as the running speed of an animal, modeled by a random walk process where the speed at any given moment depended on the preceding speed plus a random variation (see Methods). Under normal conditions, changes in visual flow were linked to changes in speed by

scaling the visual flow linearly with running speed. Occasionally, we uncoupled visual flow and speed (locomotion) by setting visual flow to zero, thus generating visuomotor mismatches (Fig. 7b). This approach allowed us to probe how predictive learning mechanisms adapt to disruptions in sensory-motor integration, providing fundamental insights into cortical computation beyond merely replicating previous experimental findings.

We first studied the sign and magnitude of layer-specific mismatch errors in our model. Since we do not model explicit error neurons, we use a proxy for error activity. Specifically, we chose to use the gradient of the cost function with respect to the activity of either L2/3 ($ME_{L2/3} \propto \frac{\partial C_{total}}{\partial z_t^{L2/3}}$) or L5 ($ME_{L5} \propto \frac{\partial C_{total}}{\partial z_t^{L5}}$), which reflects the error in each layer during learning (see Methods). To quantify the mismatch error, we calculate the gradient in those mismatch phases (visual flow and running speed uncoupled) relative to the gradient that would occur in the absence of a mismatch (baseline). If this error is positive relative to the error baseline (see Methods), we refer to it as a positive mismatch error. Conversely, if the error is negative relative to the error baseline, we refer to it as a negative mismatch error.

When visual flow was randomly halted during locomotion, we observed, on average, a positive mismatch error in L2/3 and a negative mismatch error in L5 (Fig. 7c), consistent with mismatch responses observed experimentally (Fig. 7c). The discrepancy between L2/3 and L5 arises from their roles, as postulated by our self-supervised learning cost (or error function, Eq. 5): L2/3 output predicts the upcoming visual flow while L5 encodes the actual visual flow. Hence, when visual flow halts and given top-down running speed, L2/3 output predicts a non-zero flow, thereby resulting in a positive mismatch error ($z_t^{L2/3} > z_t^{L5}$, $ME_{L2/3} > 0$). L5, in contrast, encodes the actual zero flow, and, hence, generates a negative mismatch error due to the (positive) prediction provided by L2/3 output targeting L5 ($z_t^{L2/3} > z_t^{L5}$, $ME_{L5} < 0$). Furthermore, we found that the mismatch responses in our model scale linearly with the running speed (Fig. 7d), in line with experiments[19,52]. These differential mismatch errors between L2/3 and L5 disappear when each layer optimizes its own error function rather than a shared one (Fig. S11), suggesting that both layers are jointly optimized towards common objectives.

To further test whether the mismatch errors across neurons resemble those experimentally[19], we analyzed the distribution of mismatch errors in our model. Our results show that the majority of L2/3 neurons exhibit positive mismatch errors while the majority of L5 neurons exhibit negative mismatch errors (Fig. 7e; see also Fig. S12 for mismatch errors across different model conditions), in line with Jordan and Keller[19].

A key finding from Jordan and Keller[19] was that mismatch responses invert under an open-loop paradigm, where visual flow is decoupled and presented independently of locomotion. In this open-loop condition, L2/3 neurons, which were previously depolarized during visuomotor mismatch in the closed-loop condition, instead exhibited hyperpolarization, while L5 neurons showed the opposite effect. This suggests that mismatch responses are shaped by the expected visual input rather than simply reflecting inherited sensory signals.

To better understand the computational principles underlying these experimental observations, we examined whether our model could capture this inversion of mismatch responses. Our model leverages top-down contextual information to help L2/3 in predicting incoming sensory input. However, predictive signals can also emerge intrinsically from the data itself, independent of top-down input, as shown in Fig. S3c. To investigate how our self-supervised predictive mechanism influences error signals, we trained our model in an open-loop paradigm where visual flow varied independently of running speed (see SM for details). After training, we examined how the model responds when the expected visual flow pattern is disrupted by presenting visual flow after a brief period of zero flow. Our model

qualitatively reproduced the experimental findings (Fig. S13), showing an inversion of mismatch errors between L2/3 and L5.

This sign flip arises from the predictive mechanism: in the closed-loop condition, where visual flow halts, L2/3 neurons overestimate sensory input, leading to a positive mismatch error. In contrast, in the open-loop condition, where visual flow is introduced randomly, L2/3 neurons underestimate sensory input, resulting in an error of the opposite sign. These results align with experimental observations and further support the presence of predictive mechanisms in the cortical circuit.

Here, we have focused on a feedforward implementation of the model, but cortical layers are also known to integrate information over time[53]. To test the effect of recurrence, we extended our model by incorporating an RNN in L2/3 (Fig. S14). Modeling L2/3 as an RNN enables it to integrate information from past inputs over longer timescales and use this temporal context to predict incoming inputs in layer 5. Interestingly, in addition to capturing the key findings described above, our model also captures the qualitative differences in neural responses between playback halt and visuomotor mismatch conditions observed in coupled training (CT) and non-coupled training (NT) paradigms, as reported by Attinger et al.[54]. Specifically, the model reproduces the selective mismatch responses in CT (response to mismatch, but not playback halt) versus generalized responses in NT (responses to both conditions). However, while the experimental data shows that CT mismatch responses are two to three times larger than NT responses, our model produces similar response magnitudes under both conditions.

## Differential role of L2/3 and L5 during sensorimotor mismatches

We have shown that the mismatch error responses observed in L2/3 and L5 of awake, behaving animals[19] can be explained by the distinct roles played by cortical layers engaged in self-supervised predictive learning. This suggests that layer-specific manipulations, for example, simulated optogenetic stimulation of L2/3 and L5 neurons, may reveal their unique functions. To test this in our model, we selectively increased the neuronal response of either L2/3 or L5 neurons during sensorimotor mismatches to determine their contributions to the mismatch errors in the other layer, L5 and L2/3, respectively.

To this end, we first scaled the output of L5 neurons. With moderate scaling, both positive and negative mismatch errors in L2/3 decrease in magnitude (Fig. 8b). As L5 activity increased further, neurons that displayed a positive mismatch error before scaling switched their sign to display a negative mismatch error. Likewise, neurons with a negative mismatch error before scaling signal a positive mismatch error after scaling (Fig. 8a, b). These results are a consequence of L5 neurons representing zero input during sensorimotor mismatch, while L2/3 neurons predict non-zero visual flow due to non-zero top-down input. Hence, as the L5 output increases, the feedback signal from L5 eventually surpasses the L2/3 prediction, leading to a reversal in the signs of mismatch errors in L2/3.

Thus far, we have scaled the L5 output as a whole, without differentiating between neurons exhibiting a positive or negative mismatch error. When we scaled only the outputs of L5 neurons with a positive mismatch error, the changes in L2/3 mismatch errors were heterogeneous (Fig. S15). This is likely due to the low number of neurons with positive mismatch errors in L5, limiting their impact on L2/3. In contrast, scaling only the output of L5 neurons with negative mismatch errors reversed the mismatch errors in L2/3 (Fig. S15), similar to the effect of scaling the entire L5 output (Fig. 8b).

Next, we studied how scaling the output of L2/3 affects the mismatch errors in L5. When the output of L2/3 neurons is scaled as a whole, the mismatch errors in L5 are amplified: neurons with a negative mismatch error before scaling exhibit an even stronger negative error, while those with a positive mismatch error show an even stronger positive error after scaling (Fig. 8c, d). This remains true even when we only scale the neurons with a positive mismatch error in L2/3 (Fig. S15). However, scaling only the L2/3 neurons with negative mismatch errors

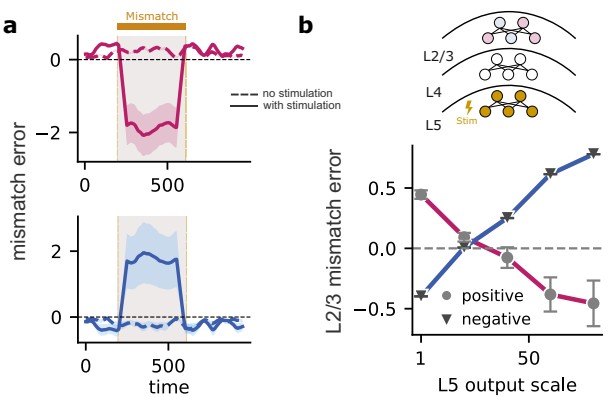

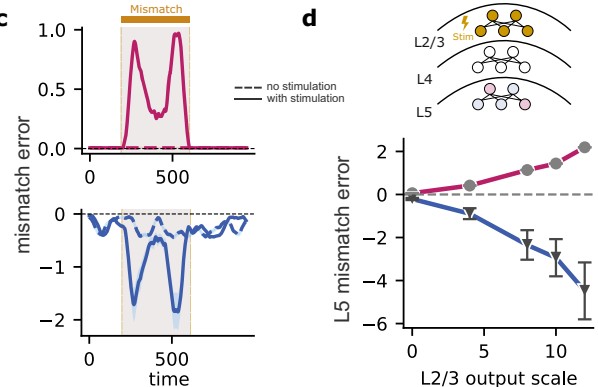

**Fig. 8 | Differential role of L5 and L2/3 activation in generating mismatch errors. a** Stimulating L5 during the mismatch interval causes mismatch errors in L2/3 to switch signs. When stimulating L5 neurons, L2/3 neurons that exhibited positive mismatch errors (magenta) become negative (top) and vice versa (blue; bottom). **b** Positive errors gradually shift towards negative and vice versa, demonstrating a direct relationship between L5 stimulation and L2/3 mismatch error modulation.

**c** Stimulating L2/3 during the mismatch interval amplifies existing mismatch errors within L5. When stimulating L2/3 neurons, L5 neurons that exhibited positive mismatch errors remain positive (top), and L5 neurons that exhibited negative mismatch errors remain negative (bottom). **d** Plot demonstrates a proportional increase or decrease in mismatch error magnitude as the output of L2/3 is scaled. Error bars represent the standard error of the mean over five different initial conditions.

reverses the mismatch errors in L5 (Fig. S15. Hence, our model predicts a dynamic interplay between L2/3 and L5, with L2/3 stimulation generally amplifying L5 mismatch errors. Interestingly, this effect is predominantly driven by L2/3 neurons signaling positive mismatch errors, while stimulating the neurons in L2/3 with negative mismatch errors reverses the direction of mismatch errors in L5.

These results can be best explained by the asymmetric contribution of L5 and L2/3 in generating mismatch errors. The strong effect of neurons with a positive mismatch error in L2/3 aligns with their role in predicting sensory input; increasing their activity strengthens this prediction signal within L5. Conversely, neurons with a negative mismatch error in L2/3 likely represent neurons suppressed by a greater-than-expected input. Enhancing their activity during a mismatch further emphasizes this "less-than-expected" signal, ultimately reversing the sign of the mismatch error in L5.

Overall, these targeted manipulations of layer-specific mismatch errors provide a valuable method for dissecting the distinct functional roles of L5 and L2/3 populations. This approach could be further explored experimentally to deepen our understanding of how neocortical layers contribute to the predictive learning of sensory streams.

## Discussion

Inspired by a refreshed view of the canonical neocortical circuitry and modern self-supervised learning algorithms, we introduce a computational theory wherein L2/3 output learns to anticipate incoming sensory input. We demonstrated L2/3's capacity to predict incoming sensory information using temporal-contextual tasks. As a result, L2/3 develops latent sensory representations that are resilient to sensory noise and occlusions, improving the ability of cortical networks to encode partially observable information. Additionally, the proposed optimization leads to layer-specific sparsity, in line with experimental findings. Subsequently, by employing a sensorimotor task, we reveal that the model's mismatch errors align with L2/3 and L5 mismatch responses observed in awake, behaving mice. Finally, using manipulations, we generated predictions for the role of specific circuit elements in self-supervised predictive learning.

Our study focuses on the canonical L4-L2/3-L5 three-layered motif[17,18]. This classical view of the neocortical microcircuit emphasizes the feedforward flow of information across layers. However, feedback projections are also evident[39,40] and both anatomical and electrophysiological data suggest the existence of direct thalamic input into L5 pyramidal cells that effectively bypasses this feedforward circuit[22,23,25,26,55]. Our model explores the computational significance of these pathways by

mapping them onto a self-supervised learning framework. Our results suggest that the two parallel thalamic pathways serve critical but distinct functions: the L4-L2/3 pathway generates temporal predictions, while the thalamic-L5 pathway provides the self-supervised target (i.e., incoming sensory input) against which these predictions are tested. Feedback from L5-to-L2/3 connects the two parallel systems to guide learning. However, in principle, the L2/3 lateral connectivity can also provide similar spatiotemporal targets for self-supervised learning. This possibility remains to be explored in future work.

Our model proposes that a critical feature of L2/3's integrative capacity is to use past information to predict incoming sensory input. This is in line with the stronger temporal integration of sensory information in superficial layers compared to deep layers[56,57], which is further reinforced by our variant of the model, where we represent L2/3 as an RNN (Fig. S14). Moreover, our work indicates that the delay introduced by the thalamic-L4-L2/3 pathway is critical for the emergence of these properties. It also suggests that the delay introduced by these neurons and synapses sets the time scale that L2/3 neurons use for temporal prediction. Since this delay is on the order of a few milliseconds[26,58], our model suggests that the temporal resolution for prediction in L2/3 of primary sensory cortices is constrained.

In principle, it is possible to achieve hierarchical spatiotemporal learning by stacking cortical areas. This is because the inherent delay in the L4-to-L2/3 circuit within each cortical area introduces a lag between the information available to L2/3 and the target represented in L5 (Fig. S16). Although simulating this hierarchical model is beyond the scope of this work, we can speculate on its computational and functional benefits. First, such an organization enables learning temporal dependencies beyond a single timestep, allowing the system to integrate information over longer timescales. Although learning extended temporal structures can be challenging, this architecture mitigates the difficulty by making predictions in the latent space rather than in the input space. By leveraging hierarchical feature extraction, the model focuses on high-level, abstract features while discarding irrelevant details. Specifically, higher-order areas make predictions based on more past information, but their task is simplified as they predict higher-level, more invariant features. In contrast, lower-level areas, which have access to more recent inputs, must predict finer details. This hierarchical model suggests that higher-order areas integrate sensory information over progressively extended durations, consistent with evidence showing that cortical areas incorporate increasingly longer temporal windows at higher levels of the hierarchy[59,60].

Furthermore, experimental findings indicate that superficial layers exhibit stronger temporal integration than deeper layers[56,57], further supporting the role of hierarchical structure in temporal learning.

Beyond facilitating multi-timestep predictions, this hierarchy also serves a regularizing function. The joint self-supervised loss between L2/3 and L5 encourages L5 to encode not only spatially invariant features but also temporally invariant representations. This aligns with findings that hierarchical temporal prediction can explain receptive field properties across the visual cortex[61], suggesting that the brain's predictive mechanisms inherently favor stable, abstract representations at higher levels.

These properties suggest a computational advantage for hierarchical predictive learning, where extended temporal context enhances abstraction and stability in neural representations. However, further work is needed to validate these hypotheses through simulations and experimental studies.

It has been well documented that superficial layers respond more sparsely to sensory stimuli than deep layers[47]. However, it was not known how this feature emerges. In our model, layer-specific sparsity occurs naturally due to the proposed predictive function of L2/3 output (Fig. 6). Our results also imply that simple measures like sparsity can help infer the optimization/learning processes of various brain structures. Indeed, we show how selective ablations of individual components of the network alter sparsity in a layer- and pathway-specific manner. While these findings provide important insights into the nature of sparsity in cortical networks, fully understanding the interplay between input, connectivity, and the emergence of sparsity during learning requires further investigation.

While our model has been mapped onto the canonical six-layered structure of the neocortex, it operates with only three layers. This raises an intriguing consideration: the evolutionarily conserved three-layered structures found in other brain regions, such as the hippocampus and piriform cortex in mammals, as well as in the cortices of other species, such as turtles[62], may represent the foundational blueprint for self-supervised learning. This structure was subsequently elaborated upon throughout evolution, firstly by increasing the number of layers, and secondly by expanding the primary locus of self-supervised learning in L2/3. Consistent with this view, we also find that sparsity increases as the number of neurons in L2/3 increases (Fig. 6b). L2/3 is greatly expanded in human evolution, even when compared to other layers[49,50]. Our results suggest that this expansion could enhance network function, particularly by broadening the predictive capabilities of the human neocortex.

L2/3's predictive capability can be enhanced by the contextual information it receives via top-down inputs. This contextual information can originate from higher-order areas within the same sensory modality or from other sensory and motor areas. According to our model, L5 neurons primarily encode the current input received through thalamic projections and provide learning signals to L2/3 via feedback connections within a cortical column. Because L5 represents the present sensory input, as opposed to L2/3's potentially inaccurate predictions, it is well-positioned to supply other areas with contextual information about the sensory state it is currently encoding. For example, L5 pyramidal cells in the auditory cortex might provide auditory context to L2/3 neurons in the visual cortex[63,64]. Within L5, different cell types likely serve distinct roles: L5 pyramidal tract neurons may facilitate the form of local self-supervised learning proposed here, while L5 intratelencephalic neurons might specialize in relaying feedback to L2/3 across cortical areas[40,65].

Our model predicts the need for feedback between the neocortical layers that carry information about error signals. This provides a form of credit assignment within the neocortical microcircuit. Although neocortex models often overlook feedback pathways,

numerous experimental studies demonstrate their existence[39,40,66]. Despite this, these pathways are understudied and their organization is not well understood, in contrast to feedforward, which often shows highly organized subnetwork architectures[67–69]. Our findings indicate that both structured (i.e., reciprocal), as well as sparse, random feedback enable learning, with the former potentially advantageous for more complex tasks. A further question is how feedback error signals may be computed in a biologically plausible manner. Recent work shows how this can be achieved using dendritic compartments and interneuron cell types[70–72]. Different interneuron subtypes, with distinct connectivity, control feedforward and feedback processing in the L2/3-L5 circuitry[73–76], which may underlie distinct aspects of the self-supervised learning proposed here.

While these studies suggest biological mechanisms through which self-supervised learning may emerge, is there evidence that it occurs within the brain? Recent experimental studies support the ability of the neocortex to perform self-supervised learning[77]. Consistent with these observations, deep networks trained using self-supervised learning better capture experimentally observed representations compared to networks trained via supervised learning[13]. For example, training deep networks using self-supervised predictive error functions yields representations that resemble visual cortical features[8,16,61,78–80]. Taking a step towards understanding the underlying learning mechanisms, recent research has introduced a combination of Hebbian and predictive synaptic plasticity[12]. This body of work supports the notion that sensory cortices engage in self-supervised learning, yet the specific circuit-level computations facilitating this process have remained unclear. Our work ties self-supervised learning to specific neocortical layers, suggesting that L2/3 and L5 provide complementary roles for implementing self-supervised learning. Consistent with these findings, the L2/3-to-L5 pathway is highly conserved across cortical regions[81,82], and behavioral studies have highlighted its importance for learning[83]. In future work, it would be interesting to test our theory by performing layer-specific experiments[84,85].

Predictive coding offers a framework for understanding how sensory representations are learned in the brain[86–88]. It postulates that the brain learns an internal model of the world from sensory streams by directly updating neuronal dynamics through prediction errors[1,7]. In temporal predictive coding, the neural networks constantly attempt to predict the incoming stimulus. Lotter et al.[89] demonstrated that a deep convolutional network that is trained using predictive coding learned sensory representations useful for downstream tasks. This contrasts with our model, where prediction violations drive plasticity rather than directly altering neuronal dynamics.

A related point is the fact that in our model, self-supervised errors and mismatch signals are derived directly from the error functions to drive the plasticity of model parameters. Implementing such error-driven signals in a biologically plausible manner remains an open question. However, we suggest two potential approaches to further develop our model, combining both inference and learning through neural dynamics (see also extended discussion in the SM). The first approach involves building on multiplexing theories of the backpropagation algorithm[72,90,91]. Building on the multiplexing framework, error signals originating in layer 5 could be kept separate from inference signals. These error-like events, potentially in the form of bursts, would propagate from L5 to L2/3, representing prediction errors in neuronal activity. The second approach would be to recast our model within a predictive coding framework[30,92]. Predictive coding jointly optimizes both model parameters and neuronal activities, which could naturally lead to prediction errors observable in the activity of both L2/3 and L5 neurons. Note that these two views are not mutually exclusive, as has previously been demonstrated[30].

Temporal predictive coding models are also often relatively abstract and do not consider how predictive coding is implemented. A notable exception is the work of Bastos et al.[1] in which it was proposed that L5 encodes input expectation while L2/3 encodes positive and negative prediction errors in separate populations. In contrast, our model proposes that L2/3 output predicts the incoming input to L5, which encodes the current sensory input. Additionally, L5 locally computes the self-supervised error between the L2/3 prediction and its current state. This approach helps explain a range of experimental observations. A distinguishing feature of our model, compared to existing predictive coding models, is its ability to both predict incoming sensory input in L2/3 and represent the current input in L5, aligning with recent advancements in deep learning[93]. In future work, it would be of interest to explicitly contrast our model with existing predictive coding frameworks.

In general, our work suggests that the circuit motifs found throughout the neocortex implement self-supervised predictive learning in the brain.

## Methods

We model the neocortical circuitry by using a network of interconnected neuronal layers. The architecture includes distinct layers corresponding to L4, L2/3, and L5 of the neocortex, with all-to-all connectivity between layers unless otherwise specified.

To represent the delayed input from L4 to L2/3, we denote the encoding of past sensory (or thalamic) input $\mathbf{x}_{t-1}$ by L4 as follows:

$$\mathbf{z}_{t-1}^{\text{L4}} = \sigma(W_{\text{Thal.}\to\text{L4}} \cdot \mathbf{x}_{t-1}) \tag{1}$$

where $\sigma$ is a sigmoid function and $W_{\text{Thal.}\to\text{L4}}$ is the weight matrix that models the connectivity from the thalamus to all L4 neurons. We model the neuronal and synaptic delay by explicitly representing L4's encoding of input from the previous time step, $t-1$, which is then processed by L2/3. More precisely, L2/3 integrates the delayed input, $\mathbf{z}_{t-1}^{\text{L4}}$, from L4 with top-down inputs at $t$ from higher-order cortical areas, $\mathbf{I}_t^{\text{td}}$. Hence, L2/3 is modeled as

$$\mathbf{z}_t^{\text{L2/3}} = \sigma(W_{\text{L4}\to\text{L2/3}} \cdot \mathbf{z}_{t-1}^{\text{L4}} + W_{\text{td}\to\text{L2/3}} \cdot I_t^{\text{td}}) \tag{2}$$

where $\mathbf{z}^{23}$ is a vector with all neurons in L2/3 and $W_{\text{L4}\to\text{L2/3}}$ is the weight matrix from L4 to L2/3. As above, all neurons are subject to the sigmoid non-linearity $\sigma$. L5 receives direct thalamic input, $\mathbf{x}_t$, and L2/3 input. It is modeled as,

$$\mathbf{z}_t^{\text{L5}} = \sigma(\alpha W_{\text{L2/3}\to\text{L5}} \cdot \mathbf{z}_t^{\text{L2/3}} + W_{\text{Thal.}\to\text{L5}} \cdot \mathbf{x}_t) \tag{3}$$

where $\mathbf{z}^5$ is a vector with all L5 neurons, $W_{\text{L2/3}\to\text{L5}}$ and $W_{\text{Thal.}\to\text{L5}}$ are the weight matrices from L2/3-to-L5 and thalamus-to-L5, respectively. The constant $\alpha$ models the dendritic-to-somatic attenuation of L2/3-to-L5

input. We set $\alpha = 0.3$, but the exact value does not qualitatively change our results.

In our network, the weight matrices $W_{\text{L2/3}\to\text{L5}}$, $W_{\text{L4}\to\text{L2/3}}$, $W_{\text{Thal.}\to\text{L4}}$, $W_{\text{Thal.}\to\text{L5}}$, and $W_{\text{td}\to\text{L2/3}}$ are subject to optimization through gradient descent. The learning rules for these connections are derived from cost functions inspired by those commonly used in self-supervised machine learning. In particular, we use a combination of two cost functions,

$$\mathcal{C}_{\text{total}} = \underbrace{\lambda_p \mathcal{C}_{\text{L2/3}\to\text{L5}}}_{\text{predictive}} + \underbrace{\lambda_r \mathcal{C}_{\text{L5}}}_{\text{reconstruction}}, \tag{4}$$

where $\lambda_p$ and $\lambda_r$ are hyperparameters that scale the predictive and reconstruction costs, respectively. The first component of $\mathcal{C}_{\text{total}}$ is the temporal self-supervised cost, where L2/3 predictions based on L4 input at time $t-1$ are compared with L5 activity at time $t$

$$\mathcal{C}_{\text{L2/3}\to\text{L5}} = \frac{1}{2}(\underbrace{W_{\text{L2/3}\to\text{L5}} \cdot \mathbf{z}_t^{\text{L2/3}}}_{\text{prediction, } \hat{\mathbf{z}}_t^{\text{L5}}} - \mathbf{z}_t^{\text{L5}})^2. \tag{5}$$

The second component of $\mathcal{C}_{\text{total}}$ encourages the model to learn non-trivial representations by *reconstructing* L5 thalamic input given its own activity, as follows

$$\mathcal{C}_{\text{L5}} = \frac{1}{2}(x_t - W_{\text{decoder}} \cdot \mathbf{z}_t^{\text{L5}})^2. \tag{6}$$

The reconstruction cost serves as a regularization term to prevent representational collapse (i.e., trivial representations) in L5. It is applied only in L5, as it receives direct thalamic input and acts as the primary representational layer, whereas L2/3 predicts the input representation without directly encoding it or accessing thalamic input. To maintain this distinction, $W_{\text{L2/3}\to\text{L5}}$ and $W_{\text{L4}\to\text{L2/3}}$ are updated solely to minimize prediction loss, while $W_{\text{Thal.}\to\text{L5}}$ is also adapted to minimize both prediction and reconstruction loss.

Hence, to ensure that the reconstruction error is not propagated to L2/3, we block the resulting error signals (i.e., gradients) from adjusting $W_{\text{L2/3}\to\text{L5}}$ weights and $W_{\text{L4}\to\text{L2/3}}$. To be precise, $W_{\text{L2/3}\to\text{L5}}$ weights and $W_{\text{L4}\to\text{L2/3}}$ synapses are only adapted to minimize the prediction error from the past, while the $W_{\text{Thal.}\to\text{L5}}$ synapses are adapted in addition to minimize both the prediction and reconstruction loss. This particular separation of learning helps prevent representational collapse[29], by ensuring that L2/3 and L5 follow different learning objectives. However, other approaches, such as variance maximization[12,37], also work (see Fig. S5).

The learning rule for $W_{\text{L2/3}\to\text{L5}}$ can be derived from the cost function as,

$$
\begin{aligned}
\Delta W_{\text{L2/3}\to\text{L5}} &= -\eta \left( \frac{\partial \mathcal{C}_{\text{total}}}{\partial W_{\text{L2/3}\to\text{L5}}} \right) \\
&= -\eta \left( \frac{\lambda_p \partial \mathcal{C}_{\text{L2/3}\to\text{L5}}}{\partial W_{\text{L2/3}\to\text{L5}}} + \frac{\lambda_r \partial \mathcal{C}_{\text{L5}}}{\partial W_{\text{L2/3}\to\text{L5}}} \right) \\
&= -\eta \lambda_p \left( W_{\text{L2/3}\to\text{L5}} \cdot \mathbf{z}_t^{\text{L2/3}} - \mathbf{z}_t^{\text{L5}} \right) \cdot \left( \mathbf{z}_t^{\text{L2/3}} - \frac{\partial \mathbf{z}_t^{\text{L5}}}{\partial W_{\text{L2/3}\to\text{L5}}} \right) \\
&= -\eta \lambda_p \left( W_{\text{L2/3}\to\text{L5}} \cdot \mathbf{z}_t^{\text{L2/3}} - \mathbf{z}_t^{\text{L5}} \right) \cdot \left[ \mathbf{z}_t^{\text{L2/3}} - \alpha \cdot \sigma' \left( \alpha W_{\text{L2/3}\to\text{L5}} \cdot \mathbf{z}_t^{\text{L2/3}} + W_{\text{Thal.}\to\text{L5}} \cdot \mathbf{x}_t \right) \cdot \mathbf{z}_t^{\text{L2/3}} \right] \\
&= -\eta \lambda_p \left( W_{\text{L2/3}\to\text{L5}} \cdot \mathbf{z}_t^{\text{L2/3}} - \mathbf{z}_t^{\text{L5}} \right) \cdot \mathbf{z}_t^{\text{L2/3}} \\
&\quad - \left( W_{\text{L2/3}\to\text{L5}} \cdot \mathbf{z}_t^{\text{L2/3}} - \mathbf{z}_t^{\text{L5}} \right) \cdot \left[ \alpha \cdot \sigma' \left( \alpha W_{\text{L2/3}\to\text{L5}} \cdot \mathbf{z}_t^{\text{L2/3}} + W_{\text{Thal.}\to\text{L5}} \cdot \mathbf{x}_t \right) \cdot \mathbf{z}_t^{\text{L2/3}} \right] \\
&= -\eta \lambda_p \left( W_{\text{L2/3}\to\text{L5}} \cdot \mathbf{z}_t^{\text{L2/3}} - \mathbf{z}_t^{\text{L5}} \right) \cdot \mathbf{z}_t^{\text{L2/3}} - \mathcal{A}
\end{aligned}
\tag{7}
$$

where we denote the effect of the attenuation factor by $\mathcal{A}$. The term $\frac{\lambda_r \partial \mathcal{C}_{L5}}{\partial W_{L2/3 \to L5}}$ is set to zero to prevent the reconstruction cost-related gradient from flowing back to L2/3.

Similarly, one can derive the learning rule for $W_{L4 \to L2/3}$ is given by the derivative of the cost function with respect to this weight matrix,

$$
\begin{aligned}
\Delta W_{L4 \to L2/3} &= -\eta \left( \frac{\partial \mathcal{C}_{total}}{\partial W_{L4 \to L2/3}} \right) = -\eta \left( \frac{\lambda_p \partial \mathcal{C}_{L2/3 \to L5}}{\partial \mathbf{z}_t^{L2/3}} \frac{\partial \mathbf{z}_t^{L2/3}}{\partial W_{L4 \to L2/3}} + \frac{\lambda_r \partial \mathcal{C}_{L5}}{\partial W_{L4 \to L2/3}} \right) \\
&= -\eta \lambda_p \left( W_{L/23 \to L5} \cdot \mathbf{z}_t^{L2/3} - \mathbf{z}^{L5} \right) \cdot \left( \frac{\partial \mathbf{z}_t^{L5}}{\partial \mathbf{z}_t^{L2/3}} - W_{L2/3 \to L5} \right) \cdot \frac{\partial \mathbf{z}_t^{L2/3}}{\partial W_{L4 \to L2/3}} \\
&= -\eta \lambda_p (W_{L2/3 \to L5} \cdot \mathbf{z}_t^{L2/3} - \mathbf{z}_t^{L5}) \left( W_{L2/3 \to L5}^{T/random} - \alpha \cdot \sigma' \left( \alpha W_{L2/3 \to L5} \cdot \mathbf{z}_t^{L2/3} + W_{Thal. \to L5} \cdot \mathbf{x}_t \right) W_{L2/3 \to L5}^{T/random} \right) \\
&\quad \cdot \sigma' \left( W_{L4 \to L2/3} \cdot \mathbf{z}_{t-1}^{L4} + W_{td \to L2/3} \cdot \mathbf{I}_t^{td} \right) \mathbf{z}_{t-1}^{L4}
\end{aligned}
\tag{8}
$$

where $W_{L2/3 \to L5}^{T/random}$ is the transpose of $W_{L2/3 \to L5}$ or a random matrix, depending on the experiments we performed (see Fig. 6). As before, the term $\frac{\lambda_r \partial \mathcal{C}_{L5}}{\partial W_{L4 \to L2/3}}$ is set to zero to prevent the reconstruction cost-related gradient from flowing back to L2/3. Finally, the learning rule for $W_{Thal. \to L5}$ can also be derived following a similar procedure:

$$
\begin{aligned}
\Delta W_{Thal. \to L5} &= -\eta \left( \frac{\partial \mathcal{C}_{total}}{\partial W_{Thal. \to L5}} \right) = -\eta \left( \frac{\lambda_p \partial \mathcal{C}_{L2/3 \to L5}}{\partial \mathbf{z}_t^{L5}} + \frac{\lambda_r \partial \mathcal{C}_{L5}}{\partial \mathbf{z}_t^{L5}} \right) \cdot \frac{\partial \mathbf{z}_t^{L5}}{\partial W_{Thal. \to L5}} \\
&= -\eta \left[ -\lambda_p (W_{L2/3 \to L5} \cdot \mathbf{z}_t^{L2/3} - \mathbf{z}_t^{L5}) + \lambda_r (W_{decoder} \cdot \mathbf{z}_t^{L5} - \mathbf{x}_t) \cdot W_{decoder} \right] \mathbf{x}_t.
\end{aligned}
\tag{9}
$$

## Tasks

**Gabor contextual-temporal task**. This task aims to investigate how Gabor patches at $t$ can be predicted based on their orientation determined by a top-down variable. To do so, we generate synthetic sequential data where each data point is a $28 \times 28$ Gabor patch. The frequencies of these patches are sampled from $\mathcal{N}(0.2, 0.1)$, with variability along the x and y axes drawn from $\mathcal{U}(3, 8)$, and orientations fixed for each class, $\theta = [0, 18°, 36°, \ldots, 162°]$. The top-down inputs can take values of $[-18°, 0°, +18°]$.

At each timestep, we randomly sample a data point $\mathbf{x}_t$ with orientations $\boldsymbol{\theta}_i$ where $i$ denotes the index in the $\theta$ list, and a top-down contextual input $\mathbf{I}_t^{td}$. The next input $\mathbf{x}_{t+1}$ is then generated by sampling a data point with orientation $\boldsymbol{\theta}_i + \mathbf{I}_t^{td}$. This setup allows for three orientations at time step $t+1$ (orientation shifts to the left, shifts to the right, or remains the same), except for angles 0° and +162°, which only have two possible successors.

According to our model, at each time step, L5 receives $\mathbf{x}_t$ as its sensory input. Simultaneously, the L2/3 network processes the output of L4 combined with the top-down contextual input.

**Noise and occlusion tests**. These experiments assess the model's robustness to input degradation (Fig. 4). We focus on two forms of degradation: noise and occlusion. For noise, Gaussian noise is added to the input as $\mathbf{x}_t^* = \mathbf{x}_t + \lambda \boldsymbol{\epsilon}_t$, where $\boldsymbol{\epsilon}_t \sim \mathcal{N}(\mathbf{0}, \mathbf{I})$, and $\lambda$ scales the noise level (values from 0 to 1, in increments of 0.2). To examine the importance of the L2/3 → L5 connection, we selectively disable the self-supervised cost during training. This prevents updates to $W_{L2/3 \to L5}$, $W_{L4 \to L2/3}$, and $W_{Thal. \to L4}$ through this loss, isolating the effects of this connection. Reconstruction performance across layers is measured using the mean squared error between the reconstructed and the original (denoised) input.

For occlusion, random image sections are obscured with a dark patch (pixel values set to zero). After training, a Support Vector Machine is used to classify outputs of L2/3 and L5 based on the $\mathbf{x}_t$ label.

Classification accuracy on a held-out test set indicates how well the model copes with occlusions.

### Visuomotor task

**Simulating the experimental setup.** To closely replicate the visuomotor task from Jordan and Keller[19], we generated synthetic sensorimotor data to model visual flow and motor speed. In our model, each vector dimension encapsulates a distinct aspect of visual flow, which is essential for simulating the sensory inputs typical in motion perception tasks. In particular, visual flow was calculated as $\mathbf{x}_t$ at any given time $t$ following

$$
\mathbf{x}_t = f(\mathbf{s}_t) + \boldsymbol{\epsilon}_t
$$

where, $f$ denotes a function that converts speed into visual flow. We set $f(\mathbf{s}_t) = \mathbf{s}_t$ to model a linear relationship (Figs. 7, 8) or $f(\mathbf{s}_t) = \sin(\mathbf{s}_t)$ to model a nonlinear interaction (Fig. S11c). The term $\mathbf{s}_t$ represents the speed at time $t$. To mimic more realistic conditions, we also add Gaussian noise, $\boldsymbol{\epsilon}_t \sim \mathcal{N}(\mathbf{0}, \mathbf{I})$. In our simulations, we model speed following a random walk. At each timestep, the speed $\mathbf{s}_t$ is determined with equal probability between the following options

- Decreasing by 1: $\mathbf{s}_t = \mathbf{s}_{t-1} - 1$
- Remaining the same: $\mathbf{s}_t = \mathbf{s}_{t-1}$
- Increasing by 1: $\mathbf{s}_t = \mathbf{s}_{t-1} + 1$

This approach simulates natural fluctuations of running speed, in which an individual might slightly accelerate, decelerate, or maintain pace from moment to moment. After sampling speeds, we generate visual flow input $\mathbf{x}_{t-1} = f(\mathbf{s}_{t-1})$ and $\mathbf{x}_t = f(\mathbf{s}_t)$, with noise as defined above.

The speed variable provides top-down context, an important factor in our model that provides contextual information. This information helps L2/3 to predict the incoming visual flow given past visual flow and the current speed.

**Mismatch simulation and measurement.** After training, we obtain a baseline error signal by averaging the gradient for each neuron across the dataset, that is, $ME_{i,BL} = \langle \frac{\partial \mathcal{C}_{total}}{\partial \mathbf{z}_t^i} \rangle_{BL}$. Next, we simulate a mismatch (i.e., breaking the coupling between locomotion and visual feedback) by randomly setting the visual flow to zero. Each mismatch period lasted for $k$ timesteps ($k = 600$ timesteps). During this mismatch period, we record the average gradient for each neuron, that is, $ME_{i,MM} = \langle \frac{\partial \mathcal{C}_{total}}{\partial \mathbf{z}_t^i} \rangle_{MM}$. To isolate the mismatch response for each neuron $\mathbf{z}_i$, we use the formula: $ME_i = ME_{i,MM} - ME_{i,BL}$. We provide a simulation of the contribution of the reconstruction and predictive losses to the total gradient in Fig. S17. This analysis shows that backpropagating the L5 reconstruction cost to L2/3 does not have a significant impact on the L2/3 gradients.

**L2/3 and L5 mismatch errors (ME).** The mismatch errors in L2/3 and L5 analysed in Figs. 7, 8 were calculated using the gradients of the self-supervised learning cost with respect to the activity of L2/3 and L5 neurons during mismatch and baseline conditions (see above). The L2/

3 gradient is calculated as

$$\frac{\partial \mathcal{C}_{\text{total}}}{\partial \mathbf{z}_t^{\text{L2/3}}} = \lambda_p \left( W_{\text{L2/3} \to \text{L5}} \cdot \mathbf{z}_t^{\text{L2/3}} - \mathbf{z}_t^{\text{L5}} \right) \tag{10}$$

$$\left( W_{\text{L2/3} \to \text{L5}} - \alpha \cdot \sigma\prime \left( \alpha W_{\text{L2/3} \to \text{L5}} \cdot \mathbf{z}_t^{\text{L2/3}} + W_{\text{Thal.} \to \text{L5}} \cdot \mathbf{x}_t \right) W_{\text{L2/3} \to \text{L5}} \right).$$

Similarly, the L5 gradient is calculated as

$$\frac{\partial \mathcal{C}_{\text{total}}}{\partial \mathbf{z}_t^{\text{L5}}} = -\lambda_p (W_{\text{L2/3} \to \text{L5}} \cdot \mathbf{z}_t^{\text{L2/3}} - \mathbf{z}_t^{\text{L5}}) - \lambda_r (\mathbf{x}_t - W_{\text{decoder}} \cdot \mathbf{z}_t^{\text{L5}}) W_{\text{decoder}}. \tag{11}$$

### Sparsity metric

To measure the sparsity of each layer in our model, we used the Treves–Rolls metric[48,94]. The population sparseness, $S$, of each layer for a single stimulus was measured as:

$$S = \frac{\left[ \sum_{i=1}^{N} r_i / N \right]^2}{\sum_{i=1}^{N} [r_i^2 / N]}$$

where $N$ is the number of neurons, and $r_i$ the activation rate of neuron $i$. To get the average population sparseness for the entire sequence, we average $S$ over a trial.

### Feedback and feedforward connection probabilities

In our work, we investigated the importance of feedback from L5 to L2/3 for learning. As part of this, we tested a range of connection probabilities, $P_{\text{connectivity}}$, for this feedback pathway. The feedback connections from L5 to L2/3 are removed with the probability of $(1 - P_{\text{connectivity}})$. The connection probability of all forward connections was set to 1.

### Reporting summary

Further information on research design is available in the Nature Portfolio Reporting Summary linked to this article.

## Data availability

Source data for all figures are provided as a Source Data file. Note that this is a computational study for which all the data can be generated using the code described below. Source data are provided with this paper.

## Code availability

The source code for the model proposed here and the respective analyses are available at https://github.com/neuralml/neoSSL (https://doi.org/10.5281/zenodo.15359568). For this implementation, PyTorch 2.2.2 was used (full list of dependencies at https://github.com/neuralml/neoSSL/blob/main/environment.yml).

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

## Acknowledgements

We would like to thank the Neural & Machine Learning group, Randy Bruno, Nicol Harper, Andrew King, Katherine Willard, Leopoldo Petreanu, David Schneider, Rebecca Jordan and Georg Keller for useful feedback. K.K.N. was funded by the UKRI Centre for Doctoral Training in Interactive Artificial, L.H. by the DFG (460088091), P.G.A by the Academy of Medical Sciences (SBF006\1047), Brain and Behaviour Research Foundation (YI Award 29300) and BBSRC (BB/X016331/1) and R.P.C. by the Medical Research Council (MR/X006107/1), BBSRC (BB/X013340/1), EPSRC (EP/X029336/1) and an ERC-UKRI Frontier Research Guarantee Starting Grant (EP/Y027841/1). We would like to thank Shuzo Sakata and Kenneth Harris for sharing their layer-specific sparsity data. This work made use of the HPC system Blue Pebble at the University of Bristol, UK. We would like to thank Dr Stewart for a donation that supported the purchase of GPU nodes embedded in the Blue Pebble HPC system.

## Author contributions

K.K.N. developed the computational framework with guidance from L.H. and R.P.C. K.K.N. performed all simulations and data analysis. K.K.N., P.A., L.H. and R.P.C. jointly wrote the manuscript. R.P.C. and L.H. jointly co-supervised the project.

## Competing interests

The authors declare no competing interests.
