## [Transparent Peer Review file · Nature Communications]

Self-supervised predictive learning accounts for cortical layer-specificity

Corresponding Author: Dr Rui Ponte Costa

Version 0:

Reviewer comments:

Reviewer #1

(Remarks to the Author)

In this manuscript, a circuit model for the neocortex is introduced that mediates unsupervised predictive learning. In the model, the direct thalamic input to layer 5 means that neurons here track the current sensory input, and the the disinaptic delay from thalamus to layer 2/3 enables layer 2/3 to generate predictions about the current sensory input based on the preceding sensory input. The model nicely recapitulates a variety of experimental observations, including layer 2/3 to layer 5 plasticity, and layer specific sparsity and visuomotor mismatch response distributions. This is an exciting model that potentially turns the canonical predictive coding view of the functional difference between superficial and infragranular layers on its head. However, I have some comments that could help improve comprehension of the model and its generalizability:

- A more thorough explanation of the model would be beneficial when introduced in the main text. I think that 'prediction error' and 'prediction' should be defined more clearly at the start. The text indicates that both layers contain prediction errors (e.g., line 61), but also states that layer 2/3 is providing predictions, while layer 5 is providing the teaching signal (prediction error), so clarity on the definition of these terms would be useful early on. This extends to the second version of the model, used for the visuomotor mismatches: it could be explained in the main text how this model works (i.e., with top-down input set to running speed).

- While the match of prediction error to the distribution of visuomotor mismatch response in layer 2/3 and infragranular layers is remarkable, a key set of evidence that layer 2/3 represents prediction errors is the fact that 1) the response to visuomotor mismatch far exceeds the responses to visually identical playback halts, and 2) there is an anticorrelation (opposing signs) between visual and motor inputs in layer 2/3 neurons (Attinger et al., 2017; Jordan and Keller, 2020; Zmarz and Keller, 2016). While the paper is already quite dense, it would be useful to know if the model can recapitulate these features for layer 2/3 mismatch responses.

- In the model, the prediction errors are not necessarily reflected in neuronal activity (e.g., line 397, which states PE leads to plasticity, but no direct change in neural dynamics). This makes it tricky to envision how the 'prediction error' in the model can explain the somatic membrane potential responses described in Jordan and Keller.

- Not all cortical areas receive sensory thalamic input in the same manner as primary sensory cortex. It would be good to discuss how the circuit model could generalize to higher order cortex that does not receive primary sensory thalamic input.

- In classical predictive processing models, the idea behind layer 2/3 providing prediction errors and layer 5/6 providing predictions arose due to the directionality of their connections between cortical areas. How do the authors envision that top-down predictions are provided in the context of their model, given that layer 5/6 provides these feedback connections in real cortical circuits? E.g., would it need to be a separate type of layer 5 neuron from those presented in the model?

Minor:

- Line 86: it is not clear to me in what manner the top-down prediction of a gabor patch from a preceding gabor patch 'mimics' the top-down motor prediction described in Leinweber et al. 2016; This may be due to the wording choice.

- Axis labels are missing from 6c and 6d.

- Figure 8a and 8c – it took me a while to understand what these plots were showing. If the plot could be shown with and without 'stimulation', that could help. For 8c, the figure legend is very short, and does not explain what the bottom and top plots show. More complete legends would be useful throughout the paper (e.g., defining error bars).
- Positive and negative mismatch errors could be defined at first use: as it stands this could lead to confusion with other currently used terminology (e.g., in common usage, positive and negative prediction errors don't necessarily refer to the sign of the response)
- Line 296: word 'input' is missing.

(Remarks on code availability)

Reviewer #2

(Remarks to the Author)

This is an interesting ms that addresses the question what the different layers within a cortical area could compute. The ms makes novel links to data with regard to the temporal processing and functional connectivity within the cortical layers.

Yet, there are some issues that first need to be clarified before publishing, we (two reviewers from the same lab) think. You'll find our detailed comments in the attached pdf.

(Remarks on code availability)

Reviewer #3

(Remarks to the Author)

This article explores a laminar model of cortical interactions based on the idea that propagation delays across the layers can serve to implement a context-dependent self-supervised learning algorithm, that is, an algorithm that attempts to learn the representations best able to predict upcoming stimuli, while taking contextual information into account.

The core idea is really nice because self-supervised algorithms is a class of algorithms most generally consistent with the brain as training does not require other signals than stimulus (and potentially context). The idea of using intrinsic delays is quite fitting and natural. The challenge is to establish concrete evidence for such a theory. Convincing evidence to this end would be a very important contribution to the field, forming a basis for the core principle of cortical computation.

The main strength of this paper is that it pins a mainly two observed features and relates it to the theory of self-supervised learning. The features selected are at a detailed level of description, one which makes this effort stand out with respect to other laminar theories of cortex (e.g. Bastos et al.). Specifically, the theory appears to explain a surprising feature of experiments from the Keller lab showing a different ratio of positive and negative prediction errors in different lamina, this phenomenon is puzzling otherwise.

There are, however, a number weaknesses, with some being serious. Currently, these make the quality of the paper fall below expectations for Nature Communications, but it is entirely possible the authors can address these issues.

1. The narrative is unfocused. I have the impression that the paper started by building a model of the mismatch response data by the Keller lab and then grew a number of tentacles, making a body of work that is hard to digest in a single narrative. Figure 2 shows a model of a rather special scenario of integrating expectations with sensory evidence, this scenario makes sense mostly in the context of the Keller data arriving at Fig. 7. Figure 2 also contains results about plasticity, which would belong to another narrative. I am not sure what the main point of Fig 3 is. Fig 4 is about normative reasons for the model, which does not necessarily help with establishing this model as a model of cortex and not a ML model and does not require the context signal. Fig 5 and 7 is about capturing data, great, but then Fig 6 is about an aspect of the model that hasn't been discussed so far: the 'gradient-communicating' feedback connections. It wasn't clear to me that the biological learning model had to be part of this paper or if one could focus on the phenomena that is explained by the self-learning objective, since it seems essential to all result, the centrality of the learning model should be put up front. Relatedly, I could not understand what the authors meant in Fig 8. All in all, the narrative jumps back and forth on several topics. Thus, the manuscript would have to be improved in terms of : explaining the full model at the start (e.g. the feedback connections, the cost function), make clear whether the focus is about puzzling features of neural data or normative, or both, given the central focus on the context integration, make clear that this is a central and general computation that needs to happen.

2. Missing controls or relevant comparisons. Fig 3 aims at providing relevant comparisons by showing that cutting some features of the model removes some of its capabilities. The argument is that these features are necessary for the capabilities. But the authors have not considered simpler architectures, such as putting a cost function on L2-3 (it would have to be different, but it can still be self-supervision) without any L5 neurons, or a single RNN receiving both thalamic input and top-down input. Also, I am not entirely convinced that the algorithm actually implements a self-supervised learning. It would seem to me that the model can learn two separate populations, one for matching with the L2-3 input and one for reconstruction. In the case of the sparsity result, there is no check that this isn't due to optimization hyperparameters (e.g. the scaling of the gradient can affect sparsity), this result is really missing an effort to identify the core mechanism.

3. Scarce experimental validation. My understanding is that there are two main pieces of experimental support for the theory: the sparsity gradient (but see point 2) and the Keller data. This seems rather thin. The problem with having only this particular data is that this ends up being a big theory to explain a small dataset. It is possible to explain the Keller data with

more trivial, ad hoc, models (like, L23 inhibits L5; Pluta and Adesnik 2019), but this becomes more difficult when the diversity of data explained by the model increases. The model would need to capture more than these few things if it is not to be an overfit speculation (e.g. the playback case in Keller 2012, orientation tuning, change in surround suppression)

4. Model consistency. When using the model in Fig 7, there is bit of a slay of hand taking place: while the activity of the L5 neurons is z_5 , what is used to capture the observation is the prediction error, $z_5 - w_{z3}$. It is fair to expect that a neuron whose potential is $z_5 - w_{z3}$ should fire according to this quantity, which would contradict the premise that the neuron activity is z_5 . Similarly for L23, if what's being plotted is the gradient received from L5, then this is not equal to the activity of that neuron. Here, I find the model breaking down in the details, but in a way that should be addressable by more carefully ascribing the ML quantities to observables. If the rate is actually the prediction error, then the layer-specific sparsity should relate to the gradient, not the activity.

Minor

6. It wasn't clear what was time zero in Fig 1b.

7. Fig 1a, I am not sure we need to stress this is updated view, I think Randy Bruno's point has been adopted by now.

8. Figure 1c has three times but the equations seem to consider only two times. I am utterly confused by either this graph or the equations. What are the gratings in the box supposed to represent within the blue boxes or next to the arrow? Why is there no link from L23 to L4 at $t=1$?

9. The model uses a pivoting of the grating, but then when it comes to simulating the virtual environment, it is intended as a translation of the grating. It seems that the authors did not adapt the algorithm to the translation situation. Are we sure it works with translation?

10. Unjustified assumptions. The authors should explain why they disregard the effect of top-down inputs onto L5s. Also, there is a peculiar scheduling to the learning, by detaching the gradient for learning some of the connections, I understand that the authors wouldn't be able to meaningfully simulate the model otherwise, but if there is no biological justification for this, it would seem to mean that the model is inconsistent and flawed. It can probably be explained in terms of the 'frozen' nature of thalamo-cortical inputs (although this has been debated lately).

11. The whole model seems limited to 14 ms of input presentation. It would be worth mentioning that the model doesn't currently account for the context signals that come later in time, as unprepared feedback from higher order area for instance.

12. I struggled with the mathematical notation sending states at time t to states at time t , except in one case. It seems that these equations should all be update equations that sends only states at time $t-1$ to states at time t . But maybe this confusion comes from my expecting that there would be a delay on z_{23} in Eq.3, which would require two timesteps. If these are not update equations and we see the equations as setting different variables, then we have z_{4^t} defined in Eq.1, but it is $z_{4^{t-1}}$ that's used in Eq. 2. So this is undefined.

13. After Eq. 6 "this reconstruction cost is only used to optimise ..." this statement is unclear to me. Are there two phases or only a detaching of the gradient? Is the detaching of the gradient for both cost functions or for the reconstruction cost only? Formally, Eq 7-9 says that there is no detaching of the gradient and no change in λ_p / λ_r for the different weight updates, but also no effect of the reconstruction cost. I find this very confusing.

14. Fig 7d, I imagine that the top is L23 and bottom is L5, but should be mentioned.

15. Note that because you are assuming biological learning, equation 10 isn't consistent since this assumes the gradient follows the transpose of the weights going up. It would be preferable to expand equation 10 to lay out more clearly what actually models the data. The sign inversion is clear from the math, it is also clear how this depends on the average sign of W_{23-5} .

16. The predictions do not navigate the E-I nature of cortex very well. Scaling all inputs in an E-I network does not necessarily lead to net excitation. Note also that microstim studies have mapped the effects of subgranular and supragranular stimulation in humans and in animal models.

(Remarks on code availability)

Version 1:

Reviewer comments:

Reviewer #1

(Remarks to the Author)

The authors have added a great amount of clarification and important additional analyses that have greatly improved the manuscript - in particular, the extension of the model to the difference observed experimentally between mismatch responses and playback halt responses in layer 2/3, and the dependence of this difference on experience.

I have only one minor comment regarding this new analysis:

The phrasing used to describe the result in S10b seems a little inaccurate: 'this model is also able to describe distinct neural responses to visuomotor mismatch observed in coupled training (CT) and

non-coupled training (NT) paradigms, as reported by (Attinger et al. 2017)'.

Given the mismatch response is the same magnitude in each condition for the model, while the response is 2-3 x higher to mismatch for CT mice compared to NT mice, I don't think the above is quite accurate. The next sentence does go on to clarify that the similarity with the data primarily concerns the difference between playback halt and mismatch, which is very nice, but the sentence above indicated to me that the model's visuomotor mismatch response itself would be different in the two conditions.

I would perhaps also note in figure S10 panels b and c that these responses are from layer 2/3.

(Remarks on code availability)

Reviewer #2

(Remarks to the Author)

As judged in our original review, the model makes an interesting claim on the role of L2/3 pyramidal neurons. It is postulated that L2/3 pyramids represent a prediction of the L5 stimulus representation based on the past, say the past 10ms. Beside the 10 ms delayed sensory activity, L2/3 also receive top-down inputs, and with these two types of information sources try to predict the sensory representation in L5. This is an interesting idea that is nicely supported by data.

While the idea of the cortical layers as a delay-structure helping to predict the next activity frame is interesting, we still think that the paper should work out some points more clearly. Some of our major points were addressed, but many remain, unfortunately. Below we try to reformulate these major issues based on the revision.

(Remarks on code availability)

Reviewer #3

(Remarks to the Author)

The revised manuscript is much improved, particularly the supplementary figures. I have no further comments.

(Remarks on code availability)

Version 2:

Reviewer comments:

Reviewer #2

(Remarks to the Author)

Dear Authors

Thanks for having taken our feedback as seriously and having written a really great paper now.

Your suggestion of seeing L2/3 as representing a memory of previous sensory inputs, modulated by top-down expectations, and out of this L2/3 memory predicting the next input to L5, is very interesting. You also make valuable links to experimental literature to show how this idea is supported.

From my side, the paper merits a publication in this journal. Congratulations!

Best wishes

Walter

(Remarks on code availability)

Self-supervised predictive learning accounts for cortical layer-specificity

Response to reviewers

Dear Reviewers,

We would like to thank you for the positive and constructive feedback. We have now addressed all your points in detail. In the revised manuscript we have:

1. Included 4 **new control/variants of the proposed computational model** and **new model-data comparisons**;
2. Updated **3 main figures** (Figs. 1, 2 and 8) to better explain the proposed model and our results;
3. Added new **5 supplementary figures** (Figs. S4, S5, S10, S11, S12);
4. **Rewritten and rearranged substantial parts of the manuscript** to clarify the points raised by the reviewers.

Overall, we believe that our manuscript has been greatly improved in the process.

Best regards,
Kevin Nejad,
Paul Anastasiades, PhD
Loreen Hertag, PhD
Rui Ponte Costa, PhD

Reviewer #1

In this manuscript, a circuit model for the neocortex is introduced that mediates unsupervised predictive learning. In the model, the direct thalamic input to layer 5 means that neurons here track the current sensory input, and the disynaptic delay from thalamus to layer 2/3 enables layer 2/3 to generate predictions about the current sensory input based on the preceding sensory input. The model nicely recapitulates a variety of experimental observations, including layer 2/3 to layer 5 plasticity, and layer specific sparsity and visuomotor mismatch response distributions. This is an exciting model that potentially turns the canonical predictive coding view of the functional difference between superficial and infragranular layers on its head. However, I have some comments that could help improve comprehension of the model and its generalizability:

- A more thorough explanation of the model would be beneficial when introduced in the main text. I think that 'prediction error' and 'prediction' should be defined more clearly at the start. The text indicates that both layers contain prediction errors (e.g., line 61), but also states that layer 2/3 is providing predictions, while layer 5 is providing the teaching signal (prediction error), so clarity on the definition of these terms would be useful early on. This extends to the second version of the model, used for the visuomotor mismatches: it could be explained in the main text how this model works (i.e., with top-down input set to running speed).

Reply: Thank you for pointing out this lack of clarity. We have now revised the manuscript in several parts to explain the model more thoroughly:

First in the introduction (lines 49-54):

"Inspired by this refreshed view of the canonical microcircuit and the predictive capabilities of self-supervised machine learning algorithms (Radford et al. 2019, Grill et al. 2020), we propose a model in which L2/3, informed by past sensory input from L4 and top-down context from higher-order cortical areas, predicts incoming sensory input. In this model, the delay from L4 to L2/3 enables L2/3 to generate predictions based on previous sensory information (Fig. 1c). Direct thalamic input to L5 provides the new sensory information, which serves as an implicit target to compare with the predictions generated by L2/3-to-L5 connections. During learning or when the model's predictions are violated, this comparison triggers prediction errors in both L5 and L2/3, thus driving circuit plasticity in a self-supervised manner. This perspective of neocortical circuitry suggests that the L4-L2/3-L5 laminar structure with parallel thalamic innervation enables the brain to learn rich temporal representations."

And also in the main text (lines 74-85):

"... At the same time, L2/3 receives thalamic input delayed by L4 via $W_{L4 \rightarrow L2/3}$ synapses, as well as top-down contextual input via the weights $W_{top-down} \rightarrow L2/3$. We hypothesize that

this combination of inputs enables L2/3 to make predictions about upcoming sensory information. In our model, we define predictions as the output of L2/3, that is $W_{L2/3 \rightarrow L5} z_t^{L2/3}$. These predictions are compared with the activity of L5 neurons z_t^{L5} ('target') that also receive the actual sensory input at timestep t . This comparison results in a self-supervised error when both are not equal and is at the core of a self-supervised learning cost, defined as $C_{L2/3 \rightarrow L5} = 0.5 (W_{L2/3 \rightarrow L5} z_t^{L2/3} - z_t^{L5})^2$. In our model, this error is fed back via L5-to-L2/3 connections to adjust the predictive model of the incoming inputs. In addition to this self-supervised learning cost, L5 is also trained to reconstruct its own input ("reconstruction cost", see Methods). During learning, we modify connections to minimize the total cost function and facilitate the encoding of sensory input. Consequently, our model requires both feed-forward connections from $L4 \rightarrow L2/3$, which relay sensory information, as well as feedback connections from L5 back to L2/3, to transmit self-supervised error signals. All weights are optimized via gradient descent."

Additionally, we have specified what we mean by the term 'prediction' in our model (lines 76-77):

"In our model, we define the predictions as the output of L2/3, that is, $W_{L2/3 \rightarrow L5} \cdot z_t^{L2/3}$ "

Furthermore, we now make a clearer distinction between the 'prediction error' involved in learning the predictive model and the 'prediction error' observed in response to an artificial mismatch between visual flow and locomotion, as described by Jordan and Keller (2020). To differentiate these concepts, we use the term 'self-supervised error' to refer to the discrepancy between the activity of L5 (the self-supervised target) and the prediction from L2/3 (the output of L2/3). This is now defined in lines 77-78. We reserve the term 'mismatch error' for describing the model's error to a mismatch between visual flow and running speed (lines 273-278).

Finally, we have revised the section on modelled visuomotor mismatches to provide a bit more detail as follows (lines 273-274):

"To this end, we used the same model as before and extended it so that the top-down input to L2/3 is equivalent to the running speed of an animal."

- While the match of prediction error to the distribution of visuomotor mismatch response in layer 2/3 and infragranular layers is remarkable, a key set of evidence that layer 2/3 represents prediction errors is the fact that 1) the response to visuomotor mismatch far exceeds the responses to visually identical playback halts, and 2) there is an anticorrelation (opposing signs) between visual and motor inputs in layer 2/3 neurons (Attinger et al., 2017; Jordan and Keller, 2020; Zmarz and Keller, 2016). While the paper is

already quite dense, it would be useful to know if the model can recapitulate these features for layer 2/3 mismatch responses.

Reply: Thank you for this comment and the suggestions to verify whether our model can also reproduce other observations that have been made in the context of visuomotor mismatch experiments. We have performed two new sets of simulations.

First, regarding ‘playback halts’, we have extended our model to capture sensory integration overtime in L2/3 by using an RNN for that layer. With this more realistic model we can also reproduce the playback halt results that the reviewer points out. This is because of the RNN’s ability to integrate information over time. We have included these new results in Figure S10 (also reproduced below) and the following text (lines 304-310):

“Here, we have focused on a feedforward implementation of the model, but cortical layers are also known to integrate information over time (Wester et al. 2012). To test the effect of recurrence we extended our model with an RNN in L2/3 (Fig. S10). Interestingly, in addition to capturing the key results above, this model is also able to describe distinct neural responses to visuomotor mismatch observed in coupled training (CT) and non-coupled training (NT) paradigms, as reported by (Attinger et al. 2017). The model accurately captured the fact that neurons in the CT condition responded to mismatch but not playback halt, while those in the NT condition responded to both.”

Supplementary Figure S10. Recurrent model captures distinct neural responses to visuomotor mismatch in coupled and non-coupled training conditions. a, Schematic of the coupled training (CT, blue) and non-coupled training (NT, red) paradigms. Visual flow (green) is coupled to locomotion (purple) in CT but not NT. Adapted from Visuomotor Coupling Shapes the Functional Development of Mouse Visual Cortex, 169 / 7, Attinger A., Wang B., Keller G. B., Cell, 1291-1302, Copyright (2017), with permission from Elsevier. b, Neural responses to mismatch (solid lines) and playback halt (dashed lines) events in CT (blue) and NT (red) conditions. Left: Experimental data. Right: Model predictions. Orange shading indicates mismatch duration. c, Correlation between mismatch error and locomotion speed during the mismatch period for the RNN model. d, Model's mismatch responses in L2/3 (top) and L5 (bottom) under CT conditions for both mismatch and playback halt events. e, Same as d but for NT conditions.

Second, regarding anticorrelation between visual and motor inputs in layer 2/3 neurons. We assume that the review is referring to Figure 4 in Jordan and Keller 2020. In the same sensorimotor task setup, but now in an open-loop paradigm (i.e. without visual-motor coupling) Jordan and Keller 2020 observed an interesting flip in the sensorimotor mismatch responses of L2/3 and L5 to visual flow onset. We have revisited the model from the perspective of this result, and our model naturally captures this result. This is because the model learns to couple visual and motor inputs, so that when this coupling is broken we see an anticorrelation effect. We have now added this new result to a new supplementary Figure S11, which we now discuss in lines 292-296. We reproduce this figure below:

Supplementary Figure S11: Model replicates reversed mismatch responses in open-loop conditions. a, Mismatch responses in L2/3 and L5 when visual input is halted during a sensorimotor task (same as in Fig. 7). b, Top: Schematic of the open-loop visual flow paradigm. Visual flow is initiated after a period of no stimulus (shaded green area). Bottom: Mismatch responses in L2/3 and L5. Neurons are sorted based on their responses in the visual flow halting experiment (Fig. 7). Experimental data (left)

demonstrates a reversal of mismatch response polarity compared to the closed-loop halting experiment (cf. panel a). Model (right) accurately captures the flipped sign of mismatch responses, indicating anti-correlation with visual flow onset. Reprinted from *Opposing Influence of Top-down and Bottom-up Input on Excitatory Layer 2/3 Neurons in Mouse Primary Visual Cortex*, 108/6, Jordan R. and Keller G., *Neuron*, 1194-1206, Copyright (2020), with permission from Elsevier. Colorbars indicate mismatch error magnitude.

- In the model, the prediction errors are not necessarily reflected in neuronal activity (e.g., line 397, which states PE leads to plasticity, but no direct change in neural dynamics). This makes it tricky to envision how the 'prediction error' in the model can explain the somatic membrane potential responses described in Jordan and Keller.

Reply: Thank you for pointing this out. In the interest of not making our story even longer we have decided to keep the model relatively abstract in terms of how the error signals are computed. This means that their computation is not biologically plausible in the current model. However, we agree that this is an important point that needs to be addressed going forward. Indeed, in the previous discussion we already highlighted this point (lines 269-279). We have now expanded on this brief discussion as follows (new lines 437-448):

"... in our model, self-supervised errors and mismatch signals are derived directly from the error functions to drive the plasticity of model parameters. Implementing such error-driven signals in a biologically plausible manner remains an open question. However, we suggest two potential approaches to further develop our model, combining both inference and learning through neural dynamics.

The first approach involves building on multiplexing theories of the backpropagation algorithm (Greedy et al. 2022, Payeur et al. 2021, Friedenberger and Naud 2023). Following on the multiplexing framework, error signals originating in layer 5 (L5) could be kept separate from inference signals. These error-like events, potentially in the form of bursts, would then propagate from L5 to layer 2/3 (L2/3), representing prediction errors in neuronal activity.

The second approach would be to recast our model within a predictive coding framework (Whittington et al. 2017, Whittington et al. 2019). Predictive coding jointly optimises both model parameters and neuronal activities, which could naturally lead to prediction errors observable in the activity of both L2/3 and L5 neurons. Note that these two views are not mutually exclusive as has been previously demonstrated (Whittington et al. 2019)."

Moreover, we have now included a more extensive discussion as part of a more detailed discussion on specific elements of the model in the 'Neural encoding of error signals' section of our extended discussion in the supplementary material.

- Not all cortical areas receive sensory thalamic input in the same manner as primary sensory cortex. It would be good to discuss how the circuit model could generalize to higher order cortex that does not receive primary sensory thalamic input.

Reply: Thank you for this suggestion. This is a very interesting suggestion, and indeed one that we have considered multiple times before. Because our current discussion is already quite long and this topic requires a more extensive discussion we have now added a new extended discussion ('Extrapolating across the cortical hierarchy') to the supplementary material in which we explore how these concepts might generalise to other cortical areas:

"How then might our model operate across the cortical hierarchy? Here, we focus on sensory information, relayed from the primary sensory thalamus, which innervates L4 and L5 and represents a sensory percept that functions as the target for L2/3 to predict. It is also possible to extrapolate this model to higher cortical areas. Ascending cortico-cortical connections can innervate middle layers of the higher-order cortex, including L4 and L5/6 (Minamisawa et al., Cell Reports 2018, Harris et al., Nature 2019), potentially replacing, or working in tandem with, thalamic inputs as described in our study. Consequently, as we move up the cortical hierarchy the target for learning is no longer external sensory input, but rather a lower-order prediction relayed by a separate input, such as the ascending projections described above (Keller and Mrsic-Flogel, Neuron 2018). For example, in higher-order regions such as the prefrontal cortex predictions are largely uncoupled from ongoing sensory activity. Instead, superficial layer 2/3 may predict a mnemonic sequence (Xie et al., Science 2022) which could be compared to stored hippocampal memories—acting as a reference, or target—which are sent to layer 5 of PFC (Anastasiades and Carter, Trends in Neurosciences 2021). In this example the hippocampus may generate sequential patterns of activity that act as an implicit target for sequences generated in the cortex. This is consistent with evidence showing the importance of hippocampal communication in learning and the role of hippocampus-to-PFC input in signaling trial outcome feedback (Brincat and Miller, Nature Neuroscience 2015). It is therefore possible that the cortical hierarchy exists along a gradient whereby predictions are grounded in sensory (external) vs internal models of the world as we move from lower-order to higher-order regions. Indeed, anatomically, regions thought to encode internal models in the brain, such as the cerebellum and hippocampus, are strongly connected with higher-order brain regions (Qiu et al. 2024). In such instances plasticity cannot rely on the tight temporal coincidence generated by simultaneous innervation of L4 and L5 by ascending axons from thalamus (Constantinople and Bruno 2013), however temporal synchrony across inputs and brain regions may instead be generated by neuronal oscillations (Brincat et al. 2015).

How is information then transmitted between brain regions within the cortical hierarchy? The classical assumption is that feed-forward information is sent from L2/3 while feedback emanates from L5. Our model builds on work showing that both motor cortex and cingulate cortex provide top-down inputs to sensory areas that play an important role in modulating sensory responses (Keller et al. 2012, Keller et al. 2018, Jordan et al. 2020). In both cases, both L2/3 and L5 pyramidal neurons mediate the projections to primary sensory areas (Mao et al. 2011, Zhang et al. 2014). Similarly, neurons in L2/3 and L5 of primary sensory areas also project to higher-order areas (Mao et al. 2011, Kim et al. 2020). In some cases, such as sensory-motor connections, there may be no strict hierarchical organisation, and S1 and M1 may reciprocally share predictions and error signals related to both forwards and inverse models via parallel connections involving L2/3 and L5 (Wolpert et al. 1995). There is also evidence for non-canonical feedback circuits such as L4 inputs which give feedback to both deep and superficial layers (Minamisawa et al. 2018).

These intracortical pathways are mediated by intratelencephalic (IT) projection neurons that populate both superficial and deep layers of the neocortex. However, within layer 5 there is also a separate subpopulation of subcortical projection neurons, so called pyramidal tract (PT) neurons, which typically do not project intracortically but instead target subcortical structures such as higher-order thalamus (Harris et al. 2015). Higher-order nuclei integrate information from across the cortex by sampling different layer 5 PT cell inputs (Harris et al. 2019) and have been shown to convey mismatches between self-generated and sensory-derived visual input (Roth et al. 2016), which helps amplify prediction error signals in superficial layers (Furutachi et al. 2024). Interestingly, while layer 2/3 stimulation causes an inhibition of layer 5 IT cells, consistent with both our model Fig. 5 and in vivo data (Keller et al. 2012), it also causes activation of layer 5 PT cells (Otsuka et al. 2021). Therefore, under conditions where a mismatch occurs within superficial layers, it may produce a corresponding activation of layer 5 PT cells which transmit this signal either up or down the cortical hierarchy via transthalamic pathways (Sherman et al. 2016).”

- In classical predictive processing models, the idea behind layer 2/3 providing prediction errors and layer 5/6 providing predictions arose due to the directionality of their connections between cortical areas. How do the authors envision that top-down predictions are provided in the context of their model, given that layer 5/6 provides these feedback connections in real cortical circuits? E.g., would it need to be a separate type of layer 5 neuron from those presented in the model?

Reply: Your thoughts on this matter have been very helpful. We now discuss the origin of our top-down input and the potential role of different L5 cells, as suggested by the reviewer, in more detail on lines 400-407:

“As we have shown, the L2/3's predictive capability can be greatly expanded through top-down input. L5 neurons are known to provide feedback to L2/3 in cortical areas (Harris et al. 2015). According to our model, L5 primarily encodes its own input, which may be further improved by also receiving top-down input (Manita et al. 2015, Harris et al. 2015). Such rich representations of sensory input are well-placed to contextualise sensory streams across modalities. For example, L5 pyramidal cells in the auditory cortex might provide auditory context for L2/3 neurons in the visual cortex (Harris et al. 2019, Yao et al. 2023). Alternatively, different cell types may mediate different functions. Whereas L5 PT neurons might mediate the form of local self-supervised learning that we propose here, L5 IT neurons might specialise in providing top-down feedback to L2/3 across cortical areas (Im et al. 2023, Hage et al. 2022).”

Minor:

- Line 86: it is not clear to me in what manner the top-down prediction of a gabor patch from a preceding gabor patch ‘mimics’ the top-down motor prediction described in Leinweber et al. 2016; This may be due to the wording choice.

Reply: We have now rephrased this sentence for extra clarity. Essentially in the Gabor task the top-down provides information about movement-related changes in the incoming stimuli. Predictive information about one’s own movements should help sensory cortices make better predictions about the incoming stimuli. We have now rephrased this part as (see lines 98-100): “This higher-order contextual cue to L2/3 is provided at each time step and conveys information about an animal's own locomotion which may be provided by the motor cortex (Leinweber et al. 2017)”.

- Axis labels are missing from 6c and 6d.

Reply: We had not included these to avoid repetition as they are the same for the 6c first panel. However as this may be confusing, we have now included all axis labels in all panels in 6c and 6d.

- Figure 8a and 8c – it took me a while to understand what these plots were showing. If the plot could be shown with and without ‘stimulation’, that could help. For 8c, the figure legend is very short, and does not explain what the bottom and top plots show. More complete legends would be useful throughout the paper (e.g., defining error bars).

Reply: Thank you for the suggestions. We have now added with and without stim as suggested for 8a and 8c. We have also expanded the figure legend for panels a and c. We now also define error bars in the caption of all figures.

- Positive and negative mismatch errors could be defined at first use: as it stands this could lead to confusion with other currently used terminology (e.g., in common usage, positive and negative prediction errors don't necessarily refer to the sign of the response)

Reply: Thank you. We now better define these terms, by defining the term 'mismatch error' for describing the model's error to a mismatch between visual flow and running speed (lines 273-278): *“We define the mismatch error as the error derived from the self-supervised learning cost with respect to the activity of either L2/3 (L2/3-specific mismatch error) or L5 (L5-specific mismatch error) during periods when the visual flow is zero, but the running speed is non-zero (see Methods). If this error is positive relative to the error baseline (see Methods), we refer to it as a positive mismatch error. Conversely, if the error is negative relative to the error baseline, we refer to it as a negative mismatch error.”*

- Line 296: word 'input' is missing.

Reply: Corrected, thanks!

Reviewer #2

This is an interesting ms that addresses the question of what the different layers within a cortical area could compute. The ms makes novel links to data with regard to the temporal processing and functional connectivity within the cortical layers.

Yet, there are some issues that first need to be clarified before publishing, we (two reviewers from the same lab) think. You'll find our detailed comments in the attached pdf.

Reply: Thank you for the encouraging and constructive feedback. Below we address in detail all the points raised.

The authors propose that a subset of local cortical circuitry enables self-supervised predictive learning of thalamic afferents through the delayed stream $\text{Thal} \rightarrow \text{L4} \rightarrow \text{L2/3} \rightarrow \text{L5}$ which learns to match direct thalamic input to L5. This approach of identifying computational principles in known cortical circuitry is commendable. The manuscript represents an interesting contribution to the landscape of ideas on computation in canonical cortical circuits. However, addressing issues on the time scales, the hierarchy, plasticity and the match to experimental data, seems necessary before publication.

Major issues

1. Prediction time scale. The temporal prediction task predicts the thalamic input x_t , fed at time t into cortical layer 4 and layer 5 neurons (Eqs 1 and 2), based on the previous thalamic input x_{t-1} at a time step earlier (Eq. 6). In the Discussion it is said that the time step 1 corresponds to 10-20 milliseconds, citing Constantinople and Bruno (2013) and Vanni et al. (2020), [26, 54]. However, the Vanni paper does not directly measure the delays, but refers to other papers with delays measured e.g. in monkey cortex upon flashing gratings (say for 250 ms, every 1 s, see Maunsell and Gibson, 1992). Inspecting Maunsell and Gibson (e.g. Figs 12 & 13) gives a delay of rather 10ms in from layer 4 to higher layers.

The Constantinople paper shows an even shorter delay from the input layer 4 to the output within the same area. Note that it is this delay that enters in the model as time step 1, not the latency of the neuronal response with respect to the stimulus. Inspecting Fig. 1b reproduced from Constantinople yields delay of 3-5 ms from layer 4 (in average 8 ms after stimulus) to layer 2/3 (in average 12 ms after the stimulus, see also Figure 1b of the ms), not the 10-20 ms cited in the Discussion. In addition, these stimuli are triggered by single

quick pulses (a light flash in the monkey study, and a whisker deflection in the mouse study). Ongoing stimulations are known to have even shorter latencies due to higher conductances and noisy, so in the perhaps 1-3ms. We think the Discussion should mention these lower numbers, not the 10-20 ms, with corresponding explanations.

Reply: Thank you for bringing this to our attention. We apologise for overlooking this aspect of the discussion. Our proposal is a more conceptual one that does not commit to specific whether it should be 3ms, 5ms or 8ms, as correctly indicated by the reviewer. For this reason, and because we do not actually model precise timescales, we think it's more appropriate to keep the discussion more abstract and refer to 'a few milliseconds' (see **line 359**). However, we now include a new extended discussion (supplementary material) on this issue of timescales ('Timescales' section), which we provide below:

"The model we propose in this study operates over a narrow time horizon which is dictated by the delay associated with mono-synaptic communication across layers, which ranges from a few milliseconds up to ~10ms depending on study (Constantinople and Bruno, Science 2013, Maunsell and Gibson, Journal of Neurophysiology 1992,Plomp et al., Scientific Reports 2017,Wilent and Contreras, Journal of Neuroscience 2004). However, sensory responses in the neocortex are often extended by tens or hundreds of milliseconds via recurrent interactions within and between cortical layers (Wester and Contreras, Journal of Neuroscience 2012). The primary visual thalamus itself possesses late activity independent of retinal input that extends for hundreds of milliseconds (Funayama et al., Journal of Neuroscience 2016). We show that our model can be made compatible with such recurrent network architectures (Fig. S10). Although spike-time-dependent plasticity operates in the range of milliseconds to tens of milliseconds (Bi and Poo, Journal of Neuroscience 1998), dendritic plateau potentials last much longer (Gambino et al., Nature 2014) and could allow plasticity across wider, behaviorally relevant temporal horizons (e.g. Bittner et al., Science 2017)."

The timescales through which cortical networks operate are also known to differ along the cortical hierarchy, with higher-order cortical regions operating over longer timescales than primary sensory areas (Gao et al., eLife 2020, Murray et al., Nature Neuroscience 2014). Despite many similarities in the organisation of cortical networks across brain regions, there are also differences in gene expression, neuronal morphology, synapse density, receptor composition, and the relative ratio of certain types of inhibitory interneurons which are thought to influence the properties of the network (Gao et al., eLife 2020). Indeed, computational models of recurrent networks containing both excitatory and inhibitory units generate multiple timescales based on differences in network architecture (Stern et al., eLife 2023). This is consistent with the idea that the canonical cortical circuit represents a generic blueprint upon which region-specific adaptations can be overlaid (Jiang et al., Science 2015)."

2. Hierarchy and timescales. The authors discuss that longer timescales could arise when stacking multiple of these microcircuits. If we generously stack 10 of these areas, the task the authors look at is to predict the thalamic output x_t (fed into layer 4 and layer 5 neurons of area 10) based on the input x_{t-1} this area 10 receives. Area 9 at the same time likely predicts x_{t-1} from x_{t-2} , while area 1 likely predicts x_{t-9} from x_{t-10} . If we consider that the time step represents 5 ms (to still be generous after the above corrections), the 10 areas predict the thalamic output 50 ms before. We don't think the authors want to say that, but when thinking through their claim that "Our framework... may provide a circuit-based explanation for why higher-order brain areas show longer timescales", this is what one gets. We think the paper should give indications how the model can scale up in the hierarchy so that after 10 areas the cortex does not only predict the thalamic input by 50 ms ahead.

Reply: Thank you for bringing up this lack of clarity on the timescales the model can predict over. We agree that it is indeed interesting and important to think how our proposal could be applicable to longer timescales. First, we have now rephrased this part of the discussion to avoid misleading the reader, as suggested by the reviewer (lines 377-382):

"... the cortex is highly hierarchical with gradually longer timescales towards the top of the hierarchy (Hasson et al. 2008 and Yamins et al. 2016). It would be of interest to study whether our model, when equipped with cortical hierarchy, could develop predictions over longer timescales. For example, the secondary visual cortex (V2) also gets primary thalamic projections, indicating that V2 could introduce further delays in the sensory input via cortico-cortical projections onto the L4→L2/3 pathway. These delays could then be compared with the incoming sensory input in L5, allowing the brain to learn predictive representations on longer timescales."

Second, we now include a new variant of the model in which L2/3 is modelled as a recurrent neural network. Although we use this model to demonstrate that a RNN-based model also gives similar results to a feedforward architecture, while also reproducing playback observations by the Keller lab (Attinger et al., 2017; Jordan and Keller, 2020; see Fig. S10), we can, in principle, use this RNN-based model to make predictions over longer time constants – for example to predict tens or even hundreds of milliseconds into the future, in particular given the hierarchical nature of the cortex. In addition, we have now added an entirely new (supplementary) discussion on the issue of timescales and how the brain may achieve longer timescales under similar principles (see new section 'Timescales' in the supplementary discussion):

"The model we propose in this study operates over a narrow time horizon which is dictated by the delay associated with mono-synaptic communication across layers, which ranges from a few milliseconds up to ~10ms depending on study (Constantinople and Bruno, Science 2013, Maunsell and Gibson, Journal of Neurophysiology 1992,Plomp et al., Scientific Reports 2017,Wilent and Contreras, Journal of Neuroscience 2004). However,

sensory responses in the neocortex are often extended by tens or hundreds of milliseconds via recurrent interactions within and between cortical layers (Wester and Contreras, Journal of Neuroscience 2012). The primary visual thalamus itself possesses late activity independent of retinal input that extends for hundreds of milliseconds (Funayama et al., Journal of Neuroscience 2016). We show that our model can be made compatible with such recurrent network architectures (Fig. S10). Although spike-time-dependent plasticity operates in the range of milliseconds to tens of milliseconds (Bi and Poo, Journal of Neuroscience 1998), dendritic plateau potentials last much longer (Gambino et al., Nature 2014) and could allow plasticity across wider, behaviorally relevant temporal horizons (e.g. Bittner et al., Science 2017).”

The timescales through which cortical networks operate are also known to differ along the cortical hierarchy, with higher-order cortical regions operating over longer timescales than primary sensory areas (Gao et al. 2020, Murray et al. 2014). Despite many similarities in the organisation of cortical networks across brain regions, there are also differences in gene expression, neuronal morphology, synapse density, receptor composition, and the relative ratio of certain types of inhibitory interneurons which are thought to influence the properties of the network (Kim et al. 2017, Gao et al. 2020). Indeed, computational models of recurrent networks containing both excitatory and inhibitory units generate multiple timescales based on differences in network architecture (Stern et al. 2023). This is consistent with the idea that the canonical cortical circuit represents a generic blueprint upon which region-specific adaptations can be overlaid (Jiang et al. 2015).”

Beside the above question, more specific questions the authors may want to answer are: (i) Which neurons send feedforward projections? (ii) Do these projections replace core thalamic input in higher areas and then do they also target L5? (iii) Which neurons (or other structures such as matrix thalamus) convey top-down contextual information?

Reply: Thank you for these suggestions. We have now included a new supplementary discussion (section ‘Extrapolating across the cortical hierarchy’) which we hope clarifies the questions of (i) feedforward projections, (ii) thalamic input to L5 and the (iii) sources of top-down contextual information:

“How then might our model operate across the cortical hierarchy? Here, we focus on sensory information, relayed from the primary sensory thalamus, which innervates L4 and L5 and represents a sensory percept that functions as the target for L2/3 to predict. It is also possible to extrapolate this model to higher cortical areas. Ascending cortico-cortical connections can innervate middle layers of the higher-order cortex, including L4 and L5/6 (Minamisawa et al., Cell Reports 2018, Harris et al., Nature 2019), potentially replacing, or working in tandem with, thalamic inputs as described in our study. Consequently, as we move up the cortical hierarchy the target for learning is no longer external sensory input, but rather a lower-order prediction relayed by a separate input, such as the ascending projections described above (Keller and Mrsic-Flogel, Neuron 2018). For example, in

higher-order regions such as the prefrontal cortex predictions are largely uncoupled from ongoing sensory activity. Instead, superficial layer 2/3 may predict a mnemonic sequence (Xie et al., Science 2022) which could be compared to stored hippocampal memories—acting as a reference, or target—which are sent to layer 5 of PFC (Anastasiades and Carter, Trends in Neurosciences 2021). In this example the hippocampus may generate sequential patterns of activity that act as an implicit target for sequences generated in the cortex. This is consistent with evidence showing the importance of hippocampal communication in learning and the role of hippocampus-to-PFC input in signaling trial outcome feedback (Brincat and Miller, Nature Neuroscience 2015). It is therefore possible that the cortical hierarchy exists along a gradient whereby predictions are grounded in sensory (external) vs internal models of the world as we move from lower-order to higher-order regions. Indeed, anatomically, regions thought to encode internal models in the brain, such as the cerebellum and hippocampus, are strongly connected with higher-order brain regions (Qiu et al. 2024). In such instances plasticity cannot rely on the tight temporal coincidence generated by simultaneous innervation of L4 and L5 by ascending axons from thalamus (Constantinople and Bruno 2013), however temporal synchrony across inputs and brain regions may instead be generated by neuronal oscillations (Brincat et al. 2015).”

How is information then transmitted between brain regions within the cortical hierarchy? The classical assumption is that feed-forward information is sent from L2/3 while feedback emanates from L5. Our model builds on work showing that both motor cortex and cingulate cortex provide top-down (contextual) inputs to sensory areas that play an important role in modulating sensory responses (Keller et al., Neuron 2012, Keller and Mrsic-Flogel, Neuron 2018, Jordan and Keller, Neuron 2020). In both cases, both L2/3 and L5 pyramidal neurons mediate the projections to primary sensory areas (Mao et al., Neuron 2011, Zhang et al., Science 2014). Similarly, neurons in L2/3 and L5 of primary sensory areas also project to higher-order areas (Mao et al., Neuron 2011, Kim et al., Neuron 2020). In some cases, such as sensory-motor connections, there may be no strict hierarchical organisation, and S1 and M1 may reciprocally share predictions and error signals related to both forwards and inverse models via parallel connections involving L2/3 and L5 (Wolpert et al., Science 1995). There is also evidence for non-canonical feedback circuits such as L4 inputs which give feedback to both deep and superficial layers (Minamisawa et al., Cell Reports 2018). These intracortical pathways are mediated by intratelencephalic (IT) projection neurons that populate both superficial and deep layers of the neocortex. However, within layer 5 there is also a separate subpopulation of subcortical projection neurons, so called pyramidal tract (PT) neurons, which typically do not project intracortically but instead target subcortical structures such as higher-order thalamus (Harris and Shepherd, Nature Neuroscience 2015). Higher-order nuclei integrate information from across the cortex by sampling different layer 5 PT cell inputs (Harris et al., Nature 2019) and have been shown to convey mismatches between self-generated and sensory-derived visual input (Roth et al., Nature Neuroscience 2016), which helps

amplify prediction error signals in superficial layers (Furutachi et al., Nature 2024). Interestingly, while layer 2/3 stimulation causes an inhibition of layer 5 IT cells, consistent with both our model Fig. 7 and in vivo data (Keller et al., Neuron 2012), it also causes activation of layer 5 PT cells (Otsuka and Kawaguchi, Communications Biology 2021). Therefore, under conditions where a mismatch occurs within superficial layers, it may produce a corresponding activation of layer 5 PT cells which transmit this signal either up or down the cortical hierarchy via transthalamic pathways (Sherman, Nature Neuroscience 2016)."

3. Mismatches. The link with Jordan and Keller (ref. 19) initially appears strong. Mismatch signals seem to emerge in the circuit, despite the fact that they are not baked into the eqs. 1- 3 for neuronal activity. However, when looking at the methods, one sees that the reported mismatch signals are in fact $-\partial C_{\text{total}}/\partial z$, and not the activity z themselves. This weakens the comparison to experimental data, which measures actual neural activity. It should be made clearer in the main text why the authors expect the formal quantity $-\partial C_{\text{total}}/\partial z$ (that contains an error, but does not represent neuronal activity in the model) to correlate their z that does not represent an error.

Reply: Thank you for pointing this out. In the interest of not making our story even longer we have decided to keep the model relatively abstract in terms of how the error signals are computed. This means that their computation is not biologically plausible in the current model. However, we agree that this is an important point that needs to be addressed going forward. Indeed, in the previous discussion we already highlighted this point (lines 269-279). We have now expanded on this brief discussion as follows (new lines 437-448):

"... in our model, self-supervised errors and mismatch signals are derived directly from the error functions to drive the plasticity of model parameters. Implementing such error-driven signals in a biologically plausible manner remains an open question. However, we suggest two potential approaches to further develop our model, combining both inference and learning through neural dynamics.

The first approach involves building on multiplexing theories of the backpropagation algorithm (Greedy et al. NeurIPS 2022, Payeur et al. Nature Neurosci 2021, Friedenberger and Naud JPhysio 2023). Following on the multiplexing framework, error signals originating in layer 5 (L5) could be kept separate from inference signals. These error-like events, potentially in the form of bursts, would then propagate from L5 to layer 2/3 (L2/3), representing prediction errors in neuronal activity.

The second approach would be to recast our model within a predictive coding framework (Whittington et al. 2017, Whittington et al. 2019). Predictive coding jointly optimises both model parameters and neuronal activities, which could naturally lead to prediction errors

observable in the activity of both L2/3 and L5 neurons. Note that these two views are not mutually exclusive as has been previously demonstrated (Whittington et al. 2019)."

Moreover, we have now included a more extensive discussion as part of a more detailed discussion on specific elements of the model in the 'Neural encoding of error signals' section of our extended discussion in the supplementary material.

Along the same lines, the authors state: "We propose that the opposing mismatch responses between L2/3 and L5 observed experimentally can be explained by visuomotor prediction errors", citing Jordan and Keller (2020). But the experimental paper claims the opposing effects within the same L2/3 neurons, as it is expressed in the paper title ("Opposing Influence of Top-down and Bottom-up Input on Excitatory Layer 2/3 Neurons in Mouse Primary Visual Cortex"). This is a fundamental problem since in the model the cost function is roughly of the form (leaving out the irrelevant parts) $C = \|z_5 - z_3\| + \dots$, and hence $-\partial C/\partial z_5$ and $-\partial C/\partial z_3$ evolve in different directions. However, when closer looking at the data in the Jordan and Keller paper, more layer 2/3 neurons show depolarizing than hyperpolarizing responses (say 2-3 times more). Instead, much more L5 neurons show hyperpolarizing than depolarizing responses (say 7-9 times). Generously speaking one may in fact say that the L2/3 neurons show a tendency to depolarize, while L5 show a tendency to hyperpolarize, as done in the ms, even this is not the message Jordan and Keller apparently want to provide. In any case, the ms should explain these discrepancies and clarify that the notion of "opposing" is different in the ms and the cited paper.

Reply: Thank you very much for this comment, it highlights that we have created confusion by using the term "opposing".

First, Jordan and Keller indeed use the term "opposing" to refer to the fact that some neurons in L2/3 depolarize in response to visuomotor mismatch while others hyperpolarize, indicating that those cells integrate top-down and bottom-up inputs with opposing signs. Similarly, we also see a distribution of responses to visuomotor mismatch (see Fig. 7).

In addition to referring to the different outcomes within L2/3, we also used the term "opposing" to highlight that in our model, on average, the responses to visuomotor mismatch, which we now define as *mismatch errors* (lines 281-285), are more positive in L2/3 and negative in L5. While this is partly a consequence of our cost function (Equation 4), the cost function alone cannot explain the distribution of responses in L2/3 and L5 that we observe as elements like the learnt connectivity between L2/3 and L5 is also a factor.

To avoid any confusion associated with the word "opposing," which is closely linked to the paper by Jordan and Keller that we cite, we have rephrased the sentence (lines 268-272):

“Using whole-cell recordings, (Jordan et al. 2020) showed that when the visuomotor coupling is temporarily broken (i.e. 'visuomotor mismatch'), the majority of L2/3 pyramidal neurons depolarize, whereas a smaller fraction of L2/3 neurons hyperpolarize. In contrast, almost all L5 excitatory neurons hyperpolarize upon visuomotor mismatch. We propose that the differences in L2/3 and L5 mismatch responses observed experimentally (Jordan et al. 2020) can be explained in a network implementing self-supervised predictive learning.”

4. Plasticity. Plasticity mechanisms in general could be better discussed. For the data corroborating the plasticity rule, ref. 28 indeed states that, for synapses targeting L5 pyramidal cells, somatic action potentials backpropagated through the dendritic tree, putatively proportional to L5 somatic activity, are responsible for the switch between LTD and LTP. The link between the learning rule and the physiology data seems to me weak, not because we find the proposed learning rule implausible, but because the only paper cited, ref. 28, is about dendritic mechanisms and localization, absent in this work. Moreover, we couldn't find in ref. 28 the specific data presented in panel 2e. Where the data comes from could be made clearer. Synapses from layer 5 to layer 2/3 need to propagate errors in order to learn $W_{L4 \rightarrow L2/3}$, but these errors are never explicitly represented. This should be discussed. Errors would also need to be propagated to layer 4 in order to learn $W_{Thal \rightarrow L4}$. On a positive note, difficulties at the level of neural circuits in applying gradient descent and backpropagation are acknowledged.

Reply: Thank you for highlighting this lack of clarity regarding the plasticity rule. We address each sub-point at a time.

Dendritic mechanisms: We do not indeed model dendritic mechanisms. However, in order to implement the proposed L2/3-to-L5 learning rule we would require a learning rule in line with the observations from ref 28. Our L2/3-to-L5 learning rule takes the following form (we omit here the attenuation term for simplicity; see full Equation 7):

$$\begin{aligned} \Delta W_{L2/3 \rightarrow L5} &= \eta \left(\frac{\partial \mathcal{C}_{\text{total}}}{\partial W_{L2/3 \rightarrow L5}} \right) \\ &= -\eta \left(\frac{\partial \mathcal{C}_{L2/3 \rightarrow L5}}{\partial W_{L2/3 \rightarrow L5}} \right) \\ &= -\eta \left(W_{L2/3 \rightarrow L5} \cdot \mathbf{z}_t^{L2/3} - \mathbf{z}_t^{L5} \right) \cdot \mathbf{z}_t^{L2/3} \end{aligned}$$

Note that for simplicity, here we are discarding the attention term (alpha), but the reviewer can find the full equation in Equation 7. In order to implement this learning rule synapses from L2/3-to-L5 pyramidal cells, which are located at proximal/distal dendrites would require information from somatic activity, which can be back-propagated to those proximal/distal synapses and this to be compared with the L2/3 input, locally at

proximal/distal dendrites. The negative term needed for the local comparison could be implemented through a local interneuron, which also receives input from L2/3 pyramidal cells. This is inline with other models that implement backprop-like computations through the use of interneurons (e.g. Greedy et al. NeurIPS 2022 and Sacramento et al. NeurIPS 2018).

Relationship with data from ref 28: We have now clarified the link to the data by Sjostrom et al. 2003 in our Fig. 2b (previously Fig. 2e) and their Figure 4D, both are reproduced below. Specifically, we extracted data from [28] focusing on L2/3-to-L5 experiments, which involved paired recordings of L2/3 and L5 PCs to induce long-term plasticity at these synapses. Typically, this led to long-term depression (LTD; non-filled blue dots), interpreted in our model as a scenario where L5 somatic activity is insufficient ($z_5 < Wz_{23}$), resulting in LTD. Conversely, Sjostrom et al. (2006) showed that boosting L5 distal dendrite activity switched the outcome to long-term potentiation (LTP; filled blue dots), consistent with $z_5 > Wz_{23}$ in our model. This is also in line with their Fig. 6 where direct depolarization of distal dendrites flipped plasticity from LTD to LTP.

boosted L2/3 → L5 exp.

Data

Model

Representation of errors: In our model, the error signals used for all plasticities, including $W_{Thal \rightarrow L4}$ and $L4 \rightarrow L2/3$ are derived directly from the error functions to drive the plasticity of model parameters, without directly altering neural dynamics. Implementing such error-driven signals in a biologically plausible manner remains an open question. However, we propose two potential approaches to further develop our model, combining both inference and learning through neural dynamics, which we have now included in the discussion (see related point above; new lines 437-448):

“The first approach involves building on multiplexing theories of the backpropagation algorithm (Greedy et al. 2022, Payeur et al. 2021, Friedenberger and Naud 2023). Following on the multiplexing framework, error signals originating in layer 5 (L5) could be kept separate from inference signals. These error-like events, potentially in the form of bursts, would then propagate from L5 to layer 2/3 (L2/3), representing prediction errors in neuronal activity.

The second approach would be to recast our model within a predictive coding framework (Whittington et al. 2017, Whittington et al. 2019). Predictive coding jointly optimises both model parameters and neuronal activities, which could naturally lead to prediction errors observable in the activity of both L2/3 and L5 neurons. Note that these two views are not mutually exclusive as has been previously demonstrated (Whittington et al. 2019).”

Minor issues

1. Other remarks regarding experimental data. The sparsity data is interesting and well presented, though we would be curious to know if the authors have a mechanistic explanation onto why predictive learning leads to sparsity. In ref. 30, cited in this work to say that “L2/3 can learn to predict image sequences”, we can see that sequences effect are present already in layer 4 (see their Fig. 1). This does not seem to fit too well with the proposed mechanism here. The clear predictions for optogenetics are great, more theoretical work should do this!

Reply: Thank you for the encouraging feedback! Our explanation is computational rather than mechanistic. As we have shown in our results (Section ‘Layer-specific sparseness emerges from self-supervised learning’) the reason why we obtain higher sparsity in L2/3 is due to the key role that L2/3 plays in predicting incoming input. Thank you for pointing that the sequence effect is also present in L4 in ref. 30. L4 may have some ability to predict incoming input through different computational principles. However, L4 is known to receive less top-down, thus its unlikely to be able to provide rich contextual predictions. Indeed, the results of [30] use a passive task, which does not require context. We have rephrased that line as (new lines 100-101): “These results are in line with experimental findings showing that L2/3 can learn to predict image sequences in a passive task (Gavornik et al. 2014).”

2. Complexity. Of course any work that looks for computational primitives in cortical circuits must ignore a lot of complexity. We cite some of this ignored complexity, not to say that these should be integrated in the model, but that it could be interesting to discuss them. (i) The authors should comment on the type of layer 5 pyramidal cells that are supposedly modeled here (IT or ET/PT). (ii) Apical dendrites of these layer 5 pyramidals, situated in layer 1, would classically also receive the top-down contextual input. (iii) Lateral connections between cells of the same layer represent a large portion of synapses, especially in layer 2/3, but are absent here. (iv) L6CT pyramidal cells project back to core thalamus that provide innervation to the area, closing the thalamocortical loop. It would be helpful if the authors could at least comment on these points.

Reply: Thank you for the suggestions. Given that the main discussion is already relatively long, we have included a new extensive discussion on the links and missing elements of the model in the supplementary material (some of which is already given above in the main points). See supplementary sections ‘Extrapolating across the cortical hierarchy’, ‘Timescales’ and ‘Neural encoding of error signals’.

3. Simulations and machine learning. In terms of simulations, the ablations study and denoising and occlusion experiments lead to expected results but seem well conducted. The link to recent advances in self-supervised learning seems solid. It could also be worth discussing successor representations in reinforcement learning, a concept that seems also closely related to the proposed mechanism.

Reply: Thank you for this suggestion. There are indeed interesting links between these two approaches. To avoid making the discussion too long, we have now included a discussion on the link between discussing the link to successor representations in the Extended discussion (see SM, ‘Relationship with successor representations’).

Reviewer #3

This article explores a laminar model of cortical interactions based on the idea that propagation delays across the layers can serve to implement a context-dependent self-supervised learning algorithm, that is, an algorithm that attempts to learn the representations best able to predict upcoming stimuli, while taking contextual information into account.

The core idea is really nice because self-supervised algorithms is a class of algorithms most generally consistent with the brain as training does not require other signals than stimulus (and potentially context). The idea of using intrinsic delays is quite fitting and natural. The challenge is to establish concrete evidence for such a theory. Convincing evidence to this end would be a very important contribution to the field, forming a basis for the core principle of cortical computation.

The main strength of this paper is that it pins a mainly two observed features and relates it to the theory of self-supervised learning. The features selected are at a detailed level of description, one which makes this effort stand out with respect to other laminar theories of cortex (e.g. Bastos et al.). Specifically, the theory appears to explain a surprising feature of experiments from the Keller lab showing a different ratio of positive and negative prediction errors in different lamina, this phenomenon is puzzling otherwise.

There are, however, a number of weaknesses, with some being serious. Currently, these make the quality of the paper fall below expectations for Nature Communications, but it is entirely possible the authors can address these issues.

Reply: Thank you for the encouraging and constructive feedback. We have addressed all points raised in detail, as we clarify below.

1. The narrative is unfocused. I have the impression that the paper started by building a model of the mismatch response data by the Keller lab and then grew a number of tentacles, making a body of work that is hard to digest in a single narrative. Figure 2 shows a model of a rather special scenario of integrating expectations with sensory evidence, this scenario makes sense mostly in the context of the Keller data arriving at Fig. 7. Figure 2 also contains results about plasticity, which would belong to another narrative. I am not sure what the main point of Fig 3 is. Fig 4 is about normative reasons for the model, which does not necessarily help with establishing this model as a model of cortex and not a ML model and does not require the context signal. Fig 5 and 7 is about capturing data, great, but then Fig 6 is about an aspect of the model that hasn't been discuss so far: the 'gradient-communicating' feedback connections. It wasn't clear to me that the biological learning model had to be part of this paper or if one could focus on the phenomena that is explained by the self-learning objective, since it seems essential to all result, the centrality of the learning model should be put up front. Relatedly, I could not understand what the authors meant in Fig 8. All in all, the narrative jumps back and forth

on several topics. Thus, the manuscript would have to be improved in terms of : explaining the full model at the start (e.g. the feedback connections, the cost function), make clear whether the focus is about puzzling features of neural data or normative, or both, given the central focus on the context integration, make clear that this is a central and general computation that needs to happen.

Reply: Thank you for your input on the structure of our manuscript. Given your feedback we have improved the flow of the story. But first, we would like to clarify why we converged on this particular structure for our story. Our modelling effort started from the observation that both L4 and L5 get direct thalamic input (Constantinople and Bruno 2013), which deviates from the canonical view of microcircuits and self-supervised learning principles commonly used in AI. Therefore our modelling of the observations made by the Keller lab is a test for the theoretical framework rather than the primary focus. Note that it is natural for us to consider contextual input to L2/3 due to its role in predicting upcoming input, this is not specific to Keller's studies, but rather an important input for any predictive model. We have, however, rearranged parts of the manuscript to provide a better flow, as suggested.

We start with the key observation by Constantinople and Bruno 2013 and a schematic introducing the model/framework, which has now been clarified following your feedback (*Figure 1*). We have rearranged *Figure 2* to continue with links with experimental data (L2/3-to-L5 synaptic plasticity). From this point onwards we focus on our (toyish) contextual-gabor task until *Figure 6*. First we demonstrate what are the contributions of the different model components ablations (*Figure 3* for feedforward *connections* and *Figure 4* for feedback). Next, we highlight functional consequences of this form of predictive learning, first in terms of denoising and occluded stimuli (*Figure 5*) and next in terms of population sparseness (*Figure 6*). In *Figure 6* we start bringing the model back to in vivo more functional experimental observations, by comparing with layer-specific sparsity (Sakata and Harris 2009), which naturally transitions to the sensorimotor task (*Figure 7 and 8*). We use the sensorimotor task as an in vivo/learning-like test to the model, which is also why it comes towards the end. In this part of the paper we first contrast our model with observations made by Keller's lab (*Figure 7*; Jordan and Keller 2020). We then finish the paper by providing further experimental predictions made by the model (*Figure 8*), which can be tested in future experimental studies. Following the reviewers feedback we have clarified a number of these figures/results, which we think has improved the paper readability.

Figure 2 shows a model of a rather special scenario of integrating expectations with sensory evidence, this scenario makes sense mostly in the context of the Keller data arriving at Fig. 7. Figure 2 also contains results about plasticity, which would belong to another narrative. I am not sure what the main point of Fig 3 is. Fig 4 is about normative reasons for the model, which does not necessarily help with establishing this model as a

model of cortex and not a ML model and does not require the context signal. Fig 5 and 7 is about capturing data, great, but then Fig 6 is about an aspect of the model that hasn't been discussed so far: the 'gradient-communicating' feedback connections. It wasn't clear to me that the biological learning model had to be part of this paper or if one could focus on the phenomena that is explained by the self-learning objective, since it seems essential to all results, the centrality of the learning model should be put up front. Relatedly, I could not understand what the authors meant in Fig 8. All in all, the narrative jumps back and forth on several topics. Thus, the manuscript would have to be improved in terms of: explaining the full model at the start (e.g. the feedback connections, the cost function), make clear whether the focus is about puzzling features of neural data or normative, or both, given the central focus on the context integration, make clear that this is a central and general computation that needs to happen.

Reply: Thank you for the feedback. To create a more focused narrative we have restructured parts of the paper as explained above. Importantly, we have moved the feedback figure (previously figure 6) right to after the ablation figure (Figure 3), which focuses on highlighting the role of feedforward connections. We also better introduce the need for this figure. We think this has greatly improved the structure of the paper, thank you for the feedback. Moreover, as suggested by the reviewer, we have expanded the introduction to the model in lines 74-83, which includes clarifications:

Cost function: we have expanded our introduction of the cost function (lines 74-80).

Feedback connections: We already briefly mentioned before that our model requires feedback connections, but now we made this more explicit by stating that (lines 80 and 81):

"In our model, this error is fed back via L5-to-L2/3 connections to adjust the predictive model of the incoming inputs."

Focus on neural data/normative/central computation: our work is jointly about data and normative, as we state in the introduction "Inspired by this refreshed view of the canonical microcircuit and the predictive capabilities of self-supervised machine learning algorithms". For this normative framework to be realisable it needs specific properties to be present in the cortical circuit, which we highlight to start with (Figure 1a,b). Note that we have now clarified the link between our computational ideas and the cortical circuit in Figure 1c (as detailed in your minor point 3 below). We also hope that we now clarify that the *computational focus* is on (temporal) self-supervised learning, from which all the rest follows, such as: (i) the need for a delay (as mediated by L4), (ii) top-down input for more richer predictions, (iii) the functional consequences of predictive learning and (iv) the link with experimental observations.

2. Missing controls or relevant comparisons. Fig 3 aims at providing relevant comparisons by showing that cutting some features of the model removes some of its capabilities. The argument is that these features are necessary for the capabilities. But the authors have not considered simpler architectures, such as putting (i) a cost function on L2-3 (it would have to be different, but it can still be self-supervision) without any L5 neurons, or a single RNN receiving both thalamic input and top-down input. Also, I am not entirely convinced that the algorithm actually implements a self-supervised learning. (ii) It would seem to me that the model can learn two separate populations, one for matching with the L2-3 input and one for reconstruction. In the case of the sparsity result, (iii) there is no check that this isn't due to optimization hyperparameters (e.g. the scaling of the gradient can affect sparsity), this result is really missing an effort to identify the core mechanism.

Reply: Thank you for pointing out the need for additional controls and relevant comparisons. We agree that demonstrating the importance of specific features in our model is crucial. Indeed, this is why we have dedicated a whole figure which includes 10 different ablation tests to demonstrate the critical elements in the model (Fig. 3), which include:

1. L2/3 top down knock-out
2. Thalamic → L5 knock-out
3. L2/3 → L5 knock-out
4. L4-mediated delay knock-out

We discuss the implication of these ablation tests to identifying the key components of the model in the following text (lines 131-141):

“Next, we investigate how ablation of these different circuit elements affects the ability to decode current sensory information from both L2/3 and L5 representations. For current input decoding (Fig.3d), L5 demonstrated robust accuracy as long as it retained access to thalamic sensory input. This aligns with its role as the primary recipient of sensory data, together with L4 (Douglas 1991, Constantinople and Bruno 2013, Pluta et al. 2015). L2/3 accuracy, however, was more dependent on the overall circuit properties. While top-down input to L2/3 provided useful context-dependent input (Fig. S2), any disruption to the core pathways within the microcircuit, except the delay knockout, compromised L2/3's ability to represent the current sensory input.

Decoding the previous input (Fig.3e) further differentiated L2/3 and L5. As anticipated, L5 exhibited limited information about previous inputs due to its exclusive focus on current thalamic information. L2/3, however, encodes information about the past as a result of the delay introduced by L4. Complete loss of this past-input representation occurred only when critical learning pathways were ablated (Thal. > L5, L2/3 > L5), or when the delay was removed; thereby synchronizing L2/3 and L5 inputs.”

However, following on the suggestions made by the reviewer we now also provide a number of new simulations and clarifications:

1. SSL cost or RNN only in L2/3 without L5:

Thank you for this suggestion. We should point out that our focus here is on studying how SSL could be implemented across cortical layers and how this can explain layer-specific observations, not on a given layer alone. We now acknowledge that similar principles may be implemented in a given layer, especially once one considers within-layer lateral connectivity, which we do now discuss in lines 364-365. Given that our focus is on self-supervised learning across layers and on relating it to existing observations across layers, our model requires L2/3 predictions to be compared with the new sensory input, which is received by L5 (as illustrated in Figure 1). Consistent with this we show in Figure 3 (panel b), that when the L2/3-to-L5 connection is removed the model fails to learn as the predictions cannot be compared with the self-supervised target. For layer 2/3 to be able to do this on its own, we would have to assume that it can store previous input and compare it with new information in a precise manner. Although this may in principle be possible, we do not know any mechanisms by which this can be done, and this would deviate from our focus on learning principles across layers. Moreover, this setup wouldn't be able to capture the different L2/3, L5 mismatch responses observed by Keller et al. To demonstrate that a model in which L2/3 can do self-supervised learning on its own does not reproduce key findings of interest, we have now implemented a new model (Fig. S12) in which L2/3 has its own SSL cost function, as suggested by the reviewer. To make this a fair comparison with our main model we still need a separate population which encodes the input into a latent space, but now the SSL cost is directly implemented in L2/3. This model setup also learns, but as expected in this case the model produces mismatched responses in L2/3, but not in L5, which is a key component that we are aiming to capture. In addition, the model also fails to capture the layer-specific sparsity result. We discuss this new model in detail in a new SM section ('Self-supervision cost directly in L2/3') and reproduce the figure below:

Figure S12: Separate self-supervised loss for L2/3 facilitates temporal learning but fails to capture L5 mismatch responses. **a**, Schematic of the model architecture incorporating a dedicated temporal self-supervised learning loss for layer 2/3 (L2/3). **b**, Demonstration of successful temporal learning in both L2/3 and L5, as evidenced by accurate prediction of the subsequent input. **c**, Comparison of L5 activity with experimental data from (Jordan et al. 2020), revealing a discrepancy in mismatch responses. The model's L5 fails to exhibit the characteristic suppressed activity observed during mismatch periods, suggesting limitations of input-space reconstruction as a proxy for perceptual learning, consistent with observations in (Balestriero et al. 2024).

2. Need for L2/3 SSL cost and L5 reconstruction cost:

Thank you for pointing out this lack of clarity. Our model requires two costs for it to learn: L2/3 SSL and L5 reconstruction cost. This is because these two have synergistic roles. Whereas L2/3-to-L5 SSL cost is required for the temporal self-supervision that we model, whereas the second L5 reconstruction cost acts as a regulariser that helps prevent representational collapse – avoids that the model learns trivial solutions. This is a problem in SSL that is commonly known and including a reconstruction cost is a way by which this can be fixed. To demonstrate this we have now included a new set of simulations in which we removed L5 reconstruction cost. As expected this results in representational collapse and the model does not learn (**new Figure S4a**). Exactly which form of avoiding representational collapse is used is not critical for our model. To demonstrate this we have replaced the L5 reconstruction cost with variance maximization (**Halvagal and Zenke, Nature Neuroscience 2023, Bardes et al., arXiv 2021**) which also enables the model to learn (**new Figure S4b**). Finally, we tested another cost for L2/3 (simple regression cost), as expected, although the model learns decodable representations it does not learn to correctly predict the incoming sensory input (Fig. S4c). We describe these new results in more detail in a new supplementary subsection ('The role of L5 regularization and L2/3 temporal self-supervision in learning'), and also briefly in the main text (lines 829-842).

3. Role of hyperparameters/core mechanism on sparsity

Thank you for the suggestions. We should point out that we already tried to identify what are the core mechanisms by ablating four of the critical components in the model (Figure 5f). This panel shows, for example, that removing top-down input had a minimal effect on sparseness. However, ablating thalamic input to L5 during learning selectively decreases L5 sparseness, which is likely due to the resulting random L5 responses. Finally, ablating L2/3-to-L5 connections or the delay component completely abolished sparsity across all layers, demonstrating their crucial role in encouraging sparseness over learning. We discuss these results in lines 843-853.

We have also now conducted, as suggested by the reviewer, a new set of simulations to investigate the robustness to optimiser hyperparameters in our findings on neural response sparsity. First, we tested a range of learning rates, our results show that the sparsity results do not depend on the exact learning rate provided that the model learns

(Fig. S5b). Second, we explored the impact of a different optimizer and regularization choices. First, we replaced the Adam optimizer with vanilla stochastic gradient descent (SGD) while retaining the reconstruction loss for L5. In this configuration, the model failed to learn, likely becoming trapped in local minima due to the shallow architecture and the limitations of SGD in navigating flat loss landscapes (Fig. S5a). However, when variance maximization was employed as the L5 regularizer, both SGD and Adam optimizers yielded qualitatively similar sparsity patterns across cortical layers (Fig. S5b). These patterns are in line with our main results in Figure 6, indicating that the observed sparsity trends are robust to variations in the optimizer as long as a suitable regularization strategy is employed for L5. These new set of results are discussed in a new supplementary section ('The influence of optimizer and regularization on learning and sparsity') and also in the main text (lines 243-244).

4. Extended model with RNN:

To help demonstrate that our results can also apply to non-feedforward architectures we have now included a new model variant in which L2/3 is model as a recurrent neural network (RNN). In this case we obtain similar results to our main results (Fig. S10). In addition, this more realistic model is also able to capture the distinct neural responses to visuomotor mismatch observed in coupled training (CT) and non-coupled training (NT) paradigms, as reported by Attinger et al. 2017. These new set of results are discussed in a new supplementary section ('Effect of visuomotor coupling and non-coupling on neural responses') and also in the main text (lines 854-879).

3. Scarce experimental validation. My understanding is that there are two main pieces of experimental support for the theory: the sparsity gradient (but see point 2) and the Keller data. This seems rather thin. The problem with having only this particular data is that this ends up being a big theory to explain a small dataset. It is possible to explain the Keller data with more trivial, ad hoc, models (like, L23 inhibits L5; Pluta and Adesnik 2019), but this becomes more difficult when the diversity of data explained by the model increases. The model would need to capture more than these few things if it is not to be an overfit speculation (e.g. the playback case in Keller 2012, orientation tuning, change in surround suppression)

Reply: Thank you for your feedback regarding the need for more experimental validation. Note that besides the Keller data and the sparsity gradient, our model is also consistent with and builds upon the known connectivity of thalamo-cortical circuits, indeed it provides a computational reason for the dual thalamo-cortical streams (L4 versus L5). We also capture long-term synaptic plasticity experiments (Fig. 2b). In addition, we now provide extensive discussion (see updated main Discussion and the new supplementary discussion), with links to the wider literature. However, we agree that additional

experimental evidence would strengthen our theory. We have therefore added two new sets of experimental results from the Keller lab, which our framework can also capture, as detailed below:

Anti-correlation in sensorimotor open-loop (Jordan and Keller 2020): An important finding by Jordan and Keller 2020 is a “anti-correlation” effect in which they observed a flipping of the mismatch responses when comparing mismatch periods to giving constant positive visual flow in the sensorimotor open-loop setup. Our model is naturally able to capture this because a positive visual flow signal flips the sign of the mismatch errors as predicted by our cost function. This is now discussed in lines 301-307 and these new results shown in Figure S11, also reproduced below.

Supplementary Figure S11: Model replicates reversed mismatch responses in open-loop conditions. **a**, Mismatch responses in L2/3 and L5 when visual input is halted during a sensorimotor task (same as in Fig. 7). **b**, Top: Schematic of the open-loop visual flow paradigm. Visual flow is initiated after a period of no stimulus (shaded green area). Bottom: Mismatch responses in L2/3 and L5. Neurons are sorted based on their responses in the visual flow halting experiment (Fig. 7). Experimental data (left) demonstrates a reversal of mismatch response polarity compared to the closed-loop halting experiment (cf. panel a). Model (right) accurately captures the flipped sign of mismatch responses, indicating anti-correlation with visual flow onset. Experimental data is reproduced from (Jordan and Keller 2020). Colorbars indicate mismatch error magnitude.

Playback halt (Attinger et al. 2017; Keller et al. 2012): We now show that our theoretical framework can also capture the playback halt effect as shown by Keller 2012 (lines 304-310). By adding recurrence to L2/3, for which there is evidence (Peron et al. 2020) and as suggested by the reviewer above, we are able to describe distinct neural

responses to visuomotor mismatch observed in coupled training (CT) and non-coupled training (NT) paradigms, as reported by (Attinger et al. 2017). The model captures the fact that neurons in the CT condition responded to mismatch but not playback halt, while those in the NT condition responded to both. Note that this is still the same overall computational framework, in which L2/3 predicts future input and L5 provides an implicit target signal. This is now discussed in lines 301-307 and these new results shown in Figure S10, also reproduced below.

Supplementary Figure S10. Recurrent model captures distinct neural responses to visuomotor mismatch in coupled and non-coupled training conditions. **a**, Schematic of the coupled training (CT, blue) and non-coupled training (NT, red) paradigms. Visual flow (green) is coupled to locomotion (purple) in CT but not NT. This figure is adapted from Attinger et al. 2017 **b**, Neural responses to mismatch (solid lines) and playback halt (dashed lines) events in CT (blue) and NT (red) conditions. Left: Experimental data. Right: Model predictions. Orange shading indicates mismatch duration. **c**, Correlation between mismatch error and locomotion speed during the mismatch period for the RNN model. **d**, Model's mismatch responses in L2/3 (top) and L5 (bottom) under CT conditions for both mismatch and playback halt events. **e**, Same as d but for NT conditions.

4. Model consistency. When using the model in Fig 7, there is bit of a slay of hand taking place: while the activity of the L5 neurons is z_5 , what is used to capture the observation is the prediction error, $z_5 - w_{23}$. It is fair to expect that a neuron whose potential is $z_5 - w_{23}$ should fire according to this quantity, which would contradict the premise that the neuron activity is z_5 . Similarly for L2/3, if what's being plotted is the gradient received from L5, then this is not equal to the activity of that neuron. Here, I find the model breaking down in the details, but in a way that should be addressable by more carefully ascribing the ML quantities to observables. If the rate is actually the prediction error, then the layer-specific sparsity should relate to the gradient, not the activity.

Reply: Thank you for pointing this out. We should clarify that we use two quantities: (i) activity z_5 and (ii) gradients (which we now refer to as mismatch errors in the context of the sensorimotor task). When we look at the sparsity of activations in Figure 6 we are purely using the activities (i.e. z_4 , $z_{2/3}$ and z_5), which is what is analysed in the experimental paper we compare with (Sakata and Harris 2009). With respect to the gradients (mismatch errors) it is true that we are comparing with subthreshold activity in the experimental setting. This is because we use an optimisation framework, in which error signals that are used for learning are calculated separately. In the interest of not making our story even longer we have decided to keep the model relatively abstract in terms of how the error signals are computed. This means that their computation is not biologically plausible in the current model. However, we agree that this is an important point that needs to be addressed going forward. Indeed, in the previous discussion we already highlighted this point (lines 269-279). We have now expanded on this brief discussion as follows (new lines 437-448):

“... in our model, self-supervised and mismatch signals are derived directly from the error functions to drive the plasticity of model parameters. Implementing such error-driven signals in a biologically plausible manner remains an open question. However, we suggest two potential approaches to further develop our model, combining both inference and learning through neural dynamics.

The first approach involves building on multiplexing theories of the backpropagation algorithm (Greedy et al. 2022, Payeur et al. 2021, Friedenberger and Naud 2023). Following on the multiplexing framework, error signals originating in layer 5 (L5) could be kept separate from inference signals. These error-like events, potentially in the form of bursts, would then propagate from L5 to layer 2/3 (L2/3), representing prediction errors in neuronal activity.

The second approach would be to recast our model within a predictive coding framework (Whittington et al. 2017, Whittington et al. 2019). Predictive coding jointly optimises both model parameters and neuronal activities, which could naturally lead to prediction errors

observable in the activity of both L2/3 and L5 neurons. Note that these two views are not mutually exclusive as has been previously demonstrated (Whittington et al. 2019)."

Moreover, we have now included a more extensive discussion as part of a more detailed discussion on specific elements of the model in the 'Neural encoding of error signals' section of our extended discussion in the supplementary material."

Furthermore, we now make a clearer distinction between the 'prediction error' involved in learning the predictive model and the 'prediction error' observed in response to an artificial mismatch between visual flow and locomotion, as described by Jordan and Keller (2020). To differentiate these concepts, we use the term 'self-supervised error' to refer to the discrepancy between the activity of L5 (the self-supervised target) and the prediction from L2/3 (the output of L2/3). This is now defined in lines 77-78. We reserve the term 'mismatch error' for describing the model's error to a mismatch between visual flow and running speed (lines 273-278).

We hope we could clarify this aspect of our model.

Minor

1. It wasn't clear what was time zero in Fig 1b.

Reply: We now clarify in the caption of Fig. 1 that time 0 represents the time of the sensory input (whisker deflection, as specified in Constantinople and Bruno 2013).

2. Fig 1a, I am not sure we need to stress this is updated view, I think Randy Bruno's point has been adopted by now.

Reply: The feedback we have received from people suggests that most people are not aware of this more recent view of cortical circuits. For this reason we think it's important to emphasise the thalamus-to-L5 connection to ensure that all readers are aware of this important but often overlooked pathway. Given Nature Communications' broad readership, we anticipate that some readers may be less familiar with Randy Bruno's and previous work.

3. Figure 1c has three times but the equations seem to consider only two times. I am utterly confused by either this graph or the equations. What are the gratings in the box supposed to represent within the blue boxes or next to the arrow? Why is there no link from L23 to L4 at $t=1$?

Reply: Sorry for the confusion induced by our schematic. We have now simplified this schematic, which we reproduce below. The new schematic is a more direct representation of equations 1-3. The schematic represents a simple task setup in which L2/3 predicts incoming input given past input and top-down contextual input. We now simply represent

how L2/3 uses previous sensory input to predict current sensory input arriving at L5. We provide three examples of this prediction (t-1, t and t+1):

4. The model uses a pivoting of the grating, but then when it comes to simulating the virtual environment, it is intended as a translation of the grating. It seems that the authors did not adapt the algorithm to the translation situation. Are we sure it works with translation?

Reply: Thank you for the question. Yes, the model also works for translation. Indeed, we already demonstrate this when we test the model for robustness to noise and occlusion (see panel Figure 4c) where the gabor patch also moves in any direction.

5. Unjustified assumptions. The authors should explain why they disregard the effect of top-down inputs onto L5s. Also, there is a peculiar scheduling to the learning, by detaching the gradient for learning some of the connections, I understand that the authors wouldn't be able to meaningfully simulate the model otherwise, but if there is no biological justification for this, it would seem to mean that the model is inconsistent and flawed. It can probably be explained in terms of the 'frozen' nature of thalamo-cortical inputs (although this has been debated lately).

Reply: Thank you for raising these important points. We appreciate the opportunity to clarify our approach.

We chose to disregard the effect of top-down inputs onto L5 to simplify the model and focus on the primary mechanisms we wanted to investigate. Our intention was to isolate and study the specific interactions between L2/3 and L5, as well as the thalamic input to L5, without the added complexity of top-down modulation. However, we agree that

top-down input onto L5 cells may play an important role and now discuss this further on lines 401-402:

“According to our model, L5 primarily encodes its own input, which may be further improved by also receiving top-down input (Manita et al. 2015; Harris et al. 2015).”

Regarding the “scheduling”: The key reason for why we do not use the reconstruction cost to change the L2/3-to-L5 connections is to avoid representational collapse, which is a common issue in self-supervised learning. We have now clarified this in lines 81-85 and lines 476-481. However, this does not have to be the only way by which to achieve this. Indeed, we have now tested a commonly used strategy, called variance maximisation, which achieves a similar outcome without gradient detachment and that has recently been linked to Hebbian learning (Halvagal and Zenke 2023). These new results are given in Fig. S4 and discussed in the new control experiments section (lines 829-842), as:

“Here, we first demonstrate that removing the reconstruction loss causes representation collapse, where the model converges to a degenerate solution, producing the same output for all inputs (Fig. S4a). We then investigate various regularisation techniques, with a particular focus on variance maximisation, which has been shown to align with Hebbian plasticity rules (Fig. S4-b) (Halvagal and Zenke 2023, Bardes et al. 2021).”

6. The whole model seems limited to 14 ms of input presentation. It would be worth mentioning that the model doesn't currently account for the context signals that come later in time, as unprepared feedback from higher order area for instance.

Reply: We thank you for this suggestion. We have now revised the discussion on delays and acknowledge the model's limitations further (lines 371-382). We also discuss the issue of timescales in more detail in a new extended discussion (see SM, section 'Timescales').

7. I struggled with the mathematical notation sending states at time t to states at time t , except in one case. It seems that these equations should all be update equations that sends only states at time $t-1$ to states at time t . But maybe this confusion comes from my expecting that there would be a delay on z_{23} in Eq.3, which would require two timesteps. If these are not update equations and we see the equations as setting different variables, then we have z_4^t defined in Eq.1, but it is z_4^{t-1} that's used in Eq. 2. So this is undefined.

Reply: Thank you for pointing this out. The reviewer is right in that these are not updated equations. Therefore z_4 in Eq. 1 should indeed be encoding $t-1$ directly, we have updated Equation 1 to reflect this.

8. After Eq. 6 “this reconstruction cost is only used to optimise ...” this statement is unclear to me. Are there two phases or only a detaching of the gradient? Is the detaching of the gradient for both cost functions or for the reconstruction cost only? Formally, Eq 7-9 says that there is no detaching of the gradient and no change in λ_p / λ_r for the different weight updates, but also no effect of the reconstruction cost. I find this very confusing.

Reply: We appreciate you bringing this to our attention. There are no two phases, only gradient detachment of the reconstruction term. We now clarify this point in lines 476-481 as follows:

“This reconstruction cost is a regularisation term that avoids representational collapse (i.e. trivial representations) in L2/3 and hence only used to optimise $W_{Thal. \rightarrow L5}$ and $W_{decoder}$ weights. To ensure the reconstruction error is not propagated to L2/3, we block the resulting error signals (i.e. gradients) from adjusting $W_{L2/3 \rightarrow L5}$ weights and the remaining weights. This particular separation of learning helps prevent representational collapse (Tian et al., ICML 2021), by ensuring that L2/3 and L5 follow different learning objectives. However, other approaches, such as variance maximization (Halvagal and Zenke 2023, Bardes et al. 2021), also work (see Fig. S4).”

9. Fig 7d, I imagine that the top is L23 and bottom is L5, but should be mentioned.

Reply: Yes, indeed. Thank you, we have now added the layer in the caption. In the plot, we only show the correlation of speed and mismatch error in L2/3 (Figure 2E,F of Jordan & Keller 2020).

10. Note that because you are assuming biological learning, equation 10 isn't consistent since this assumes the gradient follows the transpose of the weights going up. It would be preferable to expand equation 10 to lay out more clearly what actually models the data. The sign inversion is clear from the math, it is also clear how this depends on the average sign of W23-5.

Reply: Thank you for pointing this out. We have expanded equation 10 and 11 (lines 533-535).

$$ME_{L2/3} = \frac{\partial \mathcal{C}_{total}}{\partial \mathbf{z}_t^{L2/3}} = (W_{L2/3 \rightarrow L5} \cdot \mathbf{z}_t^{L2/3} - \mathbf{z}_t^{L5}) \left(W_{L2/3 \rightarrow L5} - \alpha \cdot \sigma' \left(\alpha W_{L2/3 \rightarrow L5} \cdot \mathbf{z}_t^{L2/3} + W_{Thal. \rightarrow L5} \cdot x_t \right) W_{L2/3 \rightarrow L5} \right). \quad (10)$$

$$ME_{L5} = \frac{\partial \mathcal{C}_{total}}{\partial \mathbf{z}_t^{L5}} = - (W_{L2/3 \rightarrow L5} \cdot \mathbf{z}_t^{L2/3} - \mathbf{z}_t^{L5}). \quad (11)$$

11. The predictions do not navigate the E-I nature of cortex very well. Scaling all inputs in an E-I network does not necessarily lead to net excitation. Note also that microstim studies have mapped the effects of subgranular and supragranular stimulation in humans and in animal models.

Reply: Thank you, but our model is not at the level of E-I. Therefore, the predictions we make in Figure 8 are in terms of mismatch errors, which can in principle arise from both changes in excitation or/and inhibition. We have now added discussion on how, in future work, we aim to bring our model closer to biology (lines 435-446), which would allow us to look into the influence of E-I.

Self-supervised predictive learning accounts for cortical layer-specificity

Response to reviewers

Dear Reviewers,

We would like to thank you for the positive and constructive feedback. We have addressed the remaining points raised by R2 as explained below, including new figures and changes in the text. We have also addressed the remaining minor points by R1.

Overall, we believe that our manuscript has been greatly improved in the process.

Best regards,
Kevin Nejad,
Paul Anastasiades, PhD
Loreen Hertag, PhD
Rui Ponte Costa, PhD

Reviewer #1:

The authors have added a great amount of clarification and important additional analyses that have greatly improved the manuscript - in particular, the extension of the model to the difference observed experimentally between mismatch responses and playback halt responses in layer 2/3, and the dependence of this difference on experience.

Reply: Thank you for your feedback which has greatly improved our manuscript.

Minor points:

I have only one minor comment regarding this new analysis:

The phrasing used to describe the result in S10b seems a little inaccurate: 'this model is also able to describe distinct neural responses to visuomotor mismatch observed in coupled training (CT) and non-coupled training (NT) paradigms, as reported by (Attinger et al. 2017)'.

Given the mismatch response is the same magnitude in each condition for the model, while the response is 2-3 x higher to mismatch for CT mice compared to NT mice, I don't think the above is quite accurate. The next sentence does go on to clarify that the similarity with the data primarily concerns the difference between playback halt and mismatch, which is very nice, but the sentence above indicated to me that the model's visuomotor mismatch response itself would be different in the two conditions.

Reply: Thank you for bringing this to our attention. We have rephrased this section for improved clarity as follows (new lines 335-342):

“Interestingly, in addition to capturing the key findings described above, our model also captures the qualitative differences in neural responses between playback halt and visuomotor mismatch conditions observed in coupled training (CT) and non-coupled training (NT) paradigms, as reported by (Attinger et al. 2017). Specifically, the model reproduces the selective mismatch responses in CT (response to mismatch, but not playback halt) versus generalized responses in NT (responses to both conditions). However, while the experimental data shows that CT mismatch responses are 2-3 times larger than NT responses, our model produces similar response magnitudes under both conditions.”

I would perhaps also note in figure S10 panels b and c that these responses are from layer 2/3.

Reply: Thank you, this is a good point. We have revised the figure captions for panels b and c to explicitly state that the neural responses and correlations are from layer $\frac{2}{3}$ (see updated figure, now Fig. S11).

Reviewer #3:

The revised manuscript is much improved, particularly the supplementary figures. I have no further comments.

Reply: Thank you for your valuable feedback, which has significantly enhanced our manuscript.

Reviewer #2:

As judged in our original review, the model makes an interesting claim on the role of L2/3 pyramidal neurons. It is postulated that L2/3 pyramids represent a prediction of the L5 stimulus representation based on the past, say the past 10ms. Beside the 10 ms delayed sensory activity, L2/3 also receive top-down inputs, and with these two types of information sources try to predict the sensory representation in L5. This is an interesting idea that is nicely supported by data.

While the idea of the cortical layers as a delay-structure helping to predict the next activity frame is interesting, we still think that the paper should work out some points more clearly. Some of our major points were addressed, but many remain, unfortunately. Below we try to reformulate these major issues based on the revision.

Reply: Thank you for your continued interest in our work and for the effort you have dedicated to the review. We have provided detailed clarifications below.

Major issues

1. What type of cortical computation serves the 10 ms prediction? The essence of the model of cortical layers is that L2/3 gets a delayed sensory input from thalamus (since it turns through L4) while L5 gets the direct thalamic input. Based on the roughly 10ms delay of the L2/3 sensory representation, L2/3 is able to predict the immediate representation in L5 (which is actually a post-diction, since the presence is predicted by the past). This is the version of 'self-supervised predictive coding' put forward in the paper. But what is this type of self-supervised 'post-diction' good for? If the model is stacked to 3 subsequent cortical areas, L2/3 pyramids in the 3rd area will again postdict L5 pyramids in the same area by again 10 ms, and that area overall is delayed by 30 ms as compared to the first area.

What is the purpose of these 'post-dictions' in each layer? The Discussion explains that the delays from L2/3 to L5 could also be longer, sure. The revision is also replacing L2/3 by a recurrent neuronal network (RNN) to get a more biological version of the 1-time-step delay of the original L2/3 activity model. While this RNN still captures neuronal data, it is not clear what it would add computationally. In that sense, our question of what one would gain by this model when considering other areas (whether hierarchically stacked or recurrently connected) is not really answered.

Reply: Thank you for pointing out this lack of clarity. Below we clarify/address each sub-point raised:

1. **The benefit of the 10ms delay (per se) in our model?** In our model, the delay between L4 and L2/3 is a crucial feature enabling the self-supervised predictive coding mechanism. Specifically, L2/3 receives delayed input, z_{t-1} , and generates a prediction of the current state, z_t , which is then compared with the sensory information arriving directly in L5 via thalamic input. This mechanism of post-diction offers several advantages:
 - a. It enhances the model's robustness to noise and occlusion, as illustrated in Fig. 5.
 - b. Our simulated ablation experiments (Figure 3) explicitly demonstrate that disrupting this delay significantly impairs predictive learning, underscoring its importance.
 - c. By enabling the comparison of past predictions with current sensory representations, the 10ms delay helps drive circuit plasticity in a self-supervised manner.

2. **What are the benefits of stacking the model up in a cortical hierarchy?** Stacking cortical areas yields several potential computational benefits by progressively extending the temporal scale of predictions across hierarchical levels:
 - a. *New schematic:* To address questions about information flow in hierarchically organized cortical areas, we have developed a schematic with the respective mathematical formulation that illustrates the temporal dynamics across three cortical areas, with equations generalizing to N areas (see below; new Figure S15). Our model's architecture is grounded in established experimental observations of cortical connectivity patterns (see point below). While we acknowledge that actual cortical microcircuitry exhibits additional complexity, our hierarchical model captures the essential pathways supported by current empirical evidence. A key insight revealed by our schematics is the cascading nature of predictive processing across hierarchical levels. While each area's L2/3 generates predictions with a consistent 10ms delay, the sensory information available to L2/3 becomes progressively removed from the original input signal as one ascends the hierarchy. Specifically, in area N , L2/3 operates not on direct sensory input, but on a chain of predictions derived from previous areas. This creates a natural temporal hierarchy where higher areas must learn to generate predictions based on increasingly processed and abstracted representations of past sensory states.

Targets: representation by L5

$$\begin{aligned}
 Z_t^1 &= g(S_{t-1}) \\
 Z_{t-1}^2 &= g(Z_t^1) = g(g(S_{t-1})) \\
 Z_{t-2}^3 &= g(Z_{t-1}^2) = g(g(g(S_{t-2}))) \\
 Z_{t-(n-1)}^n &= g^{(n)}(S_{t-(n-1)}) \quad \{n > 1\}
 \end{aligned}$$

Predictions of the targets: output of L2/3

$$\begin{aligned}
 \hat{Z}_t^1 &= f(g(S_{t-2}), I_t^1) \\
 \hat{Z}_{t-1}^2 &= f(f(g(S_{t-3}), I_{t-1}^1), I_t^2) \\
 \hat{Z}_{t-2}^3 &= f(f(f(g(S_{t-4}), I_{t-2}^1, I_{t-3}^2), I_t^3) \\
 \hat{Z}_{t-(n-1)}^n &= f^{n-1}(g(S_{t-n}), I_{t-n}^1, I_{t-n+1}^2, \dots, I_t^n) \quad \{n > 1\}
 \end{aligned}$$

Figure: Schematic of model across the cortical hierarchy. Top, schematic of the hierarchical organisation of cortical columns. In each column, L2/3 integrates prior input with top-down contextual information to generate a prediction in latent space, which is

then compared with the activity in L5. The first cortical column predicts the current input using the previous input, while higher-order cortical columns generate increasingly abstract predictions conditioned on predictions from the previous time step and extracting higher-level spatio-temporal features. Bottom, equations describing the flow of information within each cortical column, specifically at L2/3 and L5.

While we have not run simulations for the hierarchical model, as it is beyond the scope of this study, we would like to speculate on its computational and functional benefits.

Computational/functional benefits of hierarchy of predictions:

Extended Temporal Context: As illustrated in the schematic above, the inherent delay in the L4-to-L2/3 circuit within each cortical area introduces a lag between the information available to L2/3 and the target represented in L5. This delay in the context of a hierarchical structure (see schematic above) enables learning temporal dependencies beyond a single timestep, allowing the system to integrate information over longer timescales. Although learning extended temporal structures can be challenging, this architecture mitigates the difficulty by making predictions in latent space rather than input space. By leveraging hierarchical feature extraction, the model focuses on high-level, abstract features while discarding irrelevant details. Specifically, higher-order areas make predictions based on past information, but their task is simplified as they predict higher-level, more invariant features. In contrast, lower-level areas, which have access to more recent inputs, must predict finer details. Therefore, it enables a hierarchical version of our model to, in principle, capture long-range dependencies in sequential data.

Regularisation of L5 and prediction with more abstract representations: Beyond facilitating multi-timestep predictions, this hierarchy may also serve a regularising function. The self-supervised loss encourages L5 to learn not only spatially invariant features but also temporally stable representations, reducing sensitivity to short-term fluctuations. This allows higher cortical areas to focus on predicting abstract, slowly evolving features, while lower areas handle finer, rapidly changing details.

Planning and downstream tasks: Such hierarchical temporal predictions can prove particularly beneficial in contexts where the brain must infer future states “*in the absence of sensory input*”, for example for planning or other downstream tasks. The predictions generated by L2/3 at each level, possibly incorporating top-down contextual information, can thus be used to simulate future scenarios and inform decision-making.

- b. **Experimental/computational support:** The cortex operates as a hierarchical system, where increasingly richer representations of sensory information are built as one ascends the hierarchy (Felleman and Van Essen, *Cereb. Cortex*, 1991; Riesenhuber and Poggio, *Nature Neurosci.*, 1999; Harris et al., *Nature*, 2019). Combining this hierarchical principle with our temporal self-supervised framework implies that higher levels of the cortical hierarchy (e.g., L2/3 circuits in V4) learn temporal predictions based on progressively more abstract representations of sensory inputs. Importantly, layer 2/3 neurons in one cortical area project to layer 4 in downstream areas, thus facilitating the hierarchical stacking of our model motif (Harris and Shepherd 2015 *Nature Neurosci*; Harris et al., *Nature* 2019). For example, neurons in V1 have small and simple receptive fields, allowing them to detect fine details and local features of stimuli. In contrast, neurons in V2 exhibit larger and more complex receptive fields, enabling them to integrate information across a broader spatial context (Van den Bergh et al., *J Comp Neurol.*, 2011). Within the context of our model, this suggests that V2 generates 10 ms temporal predictions using richer, more integrated representations than V1.

This principle has been supported by modelling by Singer et al. (eLife, 2023), who demonstrated hierarchical temporal predictions in the sensory cortex, which in turn explain receptive field dynamics. However, while their work provides compelling observations, it does not address how these temporal predictions are learned. In our study, we bridge this gap and propose a specific mechanism for self-supervised learning.

We now provide a summary discussion of these points in the main text (lines 408-431):

“In principle, it is possible to achieve hierarchical spatiotemporal learning by stacking cortical areas. This is because the inherent delay in the L4-to-L2/3 circuit within each cortical area introduces a lag between the information available to L2/3 and the target represented in L5 (Fig. S15). Although simulating this hierarchical model is beyond the scope of this work, we can speculate on its computational and functional benefits. First, such an organisation enables learning temporal dependencies beyond a single timestep, allowing the system to integrate information over longer timescales. Although learning extended temporal structures can be challenging, this architecture mitigates the difficulty by making predictions in latent space rather than input space. By leveraging hierarchical feature extraction, the model focuses on high-level, abstract features while discarding irrelevant details. Specifically, higher-order areas make predictions based on more past information, but their task is simplified as they predict higher-level, more invariant features. In contrast, lower-level areas, which

have access to more recent inputs, must predict finer details. This hierarchical model suggests that higher-order areas integrate sensory information over progressively extended durations, consistent with evidence showing that cortical areas incorporate increasingly longer temporal windows at higher levels of the hierarchy (Hasson et al. 2008, Yamins et al. 2016). Furthermore, experimental findings indicate that superficial layers exhibit stronger temporal integration than deeper layers (Pitas et al. 2017, Ayaz et al. 2019), further supporting the role of hierarchical structure in temporal learning.

Beyond facilitating multi-timestep predictions, this hierarchy also serves a regularising function. The joint self-supervised loss between L2/3 and L5 encourages L5 to encode not only spatially invariant features but also temporally invariant representations. This aligns with findings that hierarchical temporal prediction can explain receptive field properties across the visual cortex (Singer et al. 2023), suggesting that the brain's predictive mechanisms inherently favour stable, abstract representations at higher levels.

These properties suggest a computational advantage for hierarchical predictive learning, where extended temporal context enhances abstraction and stability in neural representations. However, further work is needed to validate these hypotheses through simulations and experimental studies.”

3. Advantage of modelling L2/3 as an RNN? Although we added the RNN to our experiments at the request of reviewers in a previous round — leading us to capture additional experimental data — its use in modeling L2/3 offers not only greater biological plausibility but also computational advantages. In a hierarchically stacked structure, an RNN enables continuous integration of information, enhancing contextual depth for more accurate predictions as we explain below:

a. Enhanced Contextual Integration: The recurrent structure allows L2/3 to accumulate and integrate information over a longer history, thereby providing a richer contextual basis for its predictions. We have now clarified this in lines 334-335: *“Modeling layer 2/3 as an RNN enables it to integrate information from past inputs over longer timescales and use this temporal context to predict incoming inputs in layer 5.”*

b. Increased Biological Plausibility: Neocortical circuits are characterised by extensive recurrent connectivity. By modelling L2/3 as an RNN, we achieve a closer alignment with the biological structure and dynamics of the cortex.

c. Improved Predictive Accuracy: With the ability to integrate past inputs more effectively, the RNN implementation of L2/3 is well equipped to generate accurate predictions, which allows us to capture more observations (see Figure S11 added in

the previous round), which is in line with its wide use in computational neuroscience to model neural dynamics (e.g. Mante et al. Nature 2013).

Although we have not yet simulated stacked cortical areas, we consider this an important direction for future work.

2. Not gradient learning on the total cost, as incorrectly claimed. While the main text claims that plasticity of the $W_{L2/3 \rightarrow L5}$ synapses and the $W_{L4 \rightarrow L2/3}$ synapses would be gradient descent on the total cost, the calculations in Eqs 7 and 8 does not reflect that total- cost-gradient, since the gradient on the reconstruction loss is missing. These two gradient calculations only consider the prediction loss, i.e. the error in predicting the L5 activity at time t from the past at time $t - 1$. It is crucial to also consider the reconstruction loss, i.e. the error in reconstructing the original stimulus x_t based on the L5 activities. So far, only plasticity of the thalamic inputs to L5, $W_{\text{Thal.} \rightarrow L5}$ considers the reconstruction loss (Eq. 9).

The mathematical flaw should either be corrected (yielding different learning rules and implying redoing the simulations), or the statement should be adapted. A correct statement would be that the $W_{L2/3 \rightarrow L5}$ and $W_{L4 \rightarrow L2/3}$ synapses are only adapted to minimize the reconstruction from the past, while the $W_{\text{Thal.} \rightarrow L5}$ synapses are adapted in addition to minimize both the prediction and reconstruction loss. Of course, the next question is then why these different purposes, and the authors may find a good computational reason (in the sense of Major point 1), beside telling where this assumption helps to fit the data.

The authors explain after Eq. 6 "To ensure the reconstruction error is not propagated to L2/3, we block the resulting error signals (i.e. gradients) from adjusting...". But why should one ensure this? The statement reads like "To ensure that plasticity does not follow the gradient on the total cost, as claimed in the text and the Methods, we block...". This should be resolved in the sense above.

Reply: Thank you for pointing out this inconsistency and lack of clarity. The differences in the equations (see Eqs. 7–9) stem from the original motivation behind introducing the reconstruction loss. We introduced the reconstruction loss to prevent *representational collapse* (Tian et al. 2021, Halvagal et al. 2023) in L5 as we originally stated in the previous version (now lines 525-529). Representational collapse is a well-documented phenomenon in artificial neural networks where a model's learned representations (e.g., embeddings or feature mappings) become overly simplified, trivial, or uniform, losing their ability to differentiate between distinct inputs. To address this, we implemented a reconstruction loss for L5. Specifically, we applied the reconstruction loss exclusively to the synapses from the thalamus to L5 (a modelling choice that does not affect the

results, see below). We now clarify this point by removing the word *total* in the main text, revising the Methods, and by stating the reasons for our choices more explicitly in the Methods section (lines 525-529):

“The reconstruction cost serves as a regularisation term to prevent representational collapse (i.e. trivial representations) in L5. It is applied only in L5, as it receives direct thalamic input and acts as the primary representational layer, whereas L2/3 predicts the input representation without directly encoding it or accessing thalamic input. To maintain this distinction, $(W_{L2/3 \rightarrow L5})$ and $(W_{L4 \rightarrow L2/3})$ are updated solely to minimise prediction loss, while $(W_{\text{Thal.} \rightarrow L5})$ is also adapted to minimise both prediction and reconstruction loss.

Hence, to ensure that the reconstruction error is not propagated to L2/3, we block the resulting error signals (i.e. gradients) from adjusting $(W_{L2/3 \rightarrow L5})$ weights and $(W_{L4 \rightarrow L2/3})$. To be precise, $(W_{L2/3 \rightarrow L5})$ weights and $(W_{L4 \rightarrow L2/3})$ synapses are only adapted to minimise the prediction error from the past, while the $(W_{\text{Thal.} \rightarrow L5})$ synapses are adapted in addition to minimise both the prediction and reconstruction loss. This particular separation of learning helps prevent representational collapse (Tian et al. 2021), by ensuring that L2/3 and L5 follow different learning objectives. However, other approaches, such as variance maximization (Halvagal et al. 2023, Bardes et al. 2021), also work (see Fig. S5).”

We incorporated a reconstruction loss specifically in L5, because L5 serves as the primary representational layer for the input and acts as the target for L2/3. Given that L5 receives direct thalamic input, it is a natural candidate for reconstruction, allowing it to compress as much information as possible about the input. In contrast, L2/3 functions to predict the input representation rather than directly encoding it and does not have direct access to thalamic input.

Moreover, in the predictive coding framework, reconstruction typically occurs in deeper layers (L5/6) based on hierarchical representations, with signals transmitted downward to lower layers, where they are compared against actual sensory input. Following this principle, we restricted the reconstruction loss to L5, ensuring that its learning objective remains distinct from L2/3.

Since the brain is likely also having to deal with representational collapse (Halvagal and Zenke 2023) our approach provides a possible solution to this problem. In biological systems, neural representations are naturally constrained by behavioral demands and downstream processing requirements, which inherently prevent representational collapse. However, since our model learns sensory representations in a passive setting without any downstream tasks, we needed to introduce an auxiliary objective to avoid

this collapse. We chose reconstruction as our auxiliary task because it is well-established in the predictive coding literature and effectively maintains representational diversity. However, alternative approaches, such as variance maximization (Halvagal et al., 2023; Bardes et al., 2021), are also effective (see Fig. S5).

Additionally, we modified the equations to visually indicate the reconstruction loss being crossed out (see Methods, page 18, Eq. 7 and 8).

3. Experimental mismatch errors in L2/3 and L5 are not what is simulated. We still have serious issues with the definition of the ‘mismatches’ in the simulations. The model reproduces the Jordan & Keller results on encoding positive and negative prediction errors (i.e. error encoding ‘visual input minus motor predictions’ = PPE and ‘motor paper predictions minus visual input = NPE) in L2/3 and L5. When the visual flow is stopped, PPE neurons hyperpolarize, while NPE depolarize, both due to the lack of visual input. This is what is experimentally observed in L2/3. In L5, prediction PE neurons mainly hyperpolarize. The model seems to reproduce this finding.

Yet, the definition of mismatch errors in the manuscript is rather obscure. Eqs 10 and 11 define the mismatch errors in L2/3 and L5, respectively, as gradient of the total cost with respect to the representational activities in L2/3 ($z_{L2/3}$) and in L5 (z_{L5}), respectively. First, in Eq. 11, the derivative of the reconstruction loss is again forgotten. It looks like the authors are aware of this when mentioning after Eq. 11 that the reconstruction loss would vanish after learning, and conclude that it does NOT have to be considered. But mismatch errors DO represent a reconstruction loss. They represent errors in reconstructing the visual stimulus based on the past and the top-down input, and hence this DOES change the reconstruction loss, that is not 0 anymore and hence needs to be considered.

Second and more importantly, these definitions do not reflect how mismatch activities are produced in the experiment by either halting the visual flow, or the motor input. Why not simply stopping the visual input and record the activities of the error neurons in L2/3 and L5? This yields error activities that are conceptually very different from a cost change induced by a small increase in L2/3 and L5 representational activities.

When making the claim that the Jordan & Keller results are reproduced, the paper should also hint to a formula derived from the model, why this happens. Currently, no rational explanation is given, why Eqs 10 and 11 (the definitions of the mismatch errors via partial derivatives) explain (i) the simulation results, and explain (ii) the experimental data. So far, simulations are shown, but given the partial-derivative-definition of

mismatch responses, it remains obscure what these simulation results really tell about the experimental mismatch responses.

Reply: Thank you for highlighting these points.

Equation 11 missing reconstruction term: We agree with the reviewer that the reconstruction loss was not shown in Eq. 11. This was indeed a typo and has been corrected, and we have now updated the equations to include the reconstruction loss (see new Eq. 11, page 19). However, we would like to clarify that the reconstruction loss was applied in our simulations. Importantly, as shown in our new analysis (see below), the sign of the total error in L5 is dominated by the sign of the gradient of the self-supervised cost function, as the gradient of reconstruction loss is very small.

Additionally, we have incorporated the baseline into Eqs. 10 and 11.

Error neurons: Unlike classical predictive coding approaches, in this work, we do not use separate representation and error neurons. Instead, each neuron in our model generates activity during the forward pass, which is then used to compute its respective error signals through gradient calculations. To clarify this distinction, we have now included a new supplementary figure (new Fig. S1; see also below) illustrating the differences between classical predictive coding models and our ANN-based model. We also briefly make this clarification in the introduction to the model in the main text (lines 85-88).

Figure (new Fig. S1): Comparison between our model and a predictive coding architecture. **a**, Our model implements error-driven learning similar to standard artificial neural networks, where errors are propagated directly through backward synapses to update neuronal activities. Therefore, all neurons in our model are both representation and error neurons (blue neuron with dashed red outline). **b**, In contrast, predictive coding explicitly represents prediction errors using dedicated error neurons (red) that are distinct from representation neurons (blue). Each representation neuron is paired with an error neuron that computes the mismatch between predictions and actual values.

Since we do not model explicit error neurons and therefore cannot directly compare their activity with experimentally observed error neurons, we use a proxy for error activity. Specifically, we chose to use the gradient, which reflects the error levels in our network during learning. To quantify the error during a mismatch, we calculate the gradient in those mismatch scenarios relative to the gradient that would occur in the absence of a mismatch (baseline). We have now clarified this point in lines 586-596.

We would like to emphasize that this approach does not preclude the possibility of recasting our model within a classical predictive coding framework with explicit error neurons. However, for this initial study, we opted for a more abstract approach and chose to work with gradients instead.

Linking mismatch error with equations: During mismatch, L5 encodes the current target (input), which is near zero $z_t^{L5} \sim 0$, while L2/3 predicts a positive input for L5 given top-down input, such that $W_{L2/3 \rightarrow L5} z_t^{L2/3} > 0$. This discrepancy creates a prediction error, leading to a negative mismatch error in L5:

$$ME_{L5} \sim -W_{L2/3 \rightarrow L5} x_t^{L2/3} + z_t^{L5}, \text{ where } z_t^{L5} < W_{L2/3 \rightarrow L5} z_t^{L2/3}$$

Conversely, the mismatch errors in L2/3 is positive:

$$ME_{L2/3} \sim W_{L2/3 \rightarrow L5} x_t^{L2/3} - z_t^{L5}, \text{ where } z_t^{L5} < W_{L2/3 \rightarrow L5} z_t^{L2/3}$$

To provide extra clarity, we have derived equations (see the Appendix to this response attached) describing the sign and magnitude of error (gradients of cost functions w.r.t. neurons in L2/3 and L5). We also show that the reconstruction loss is naturally small.

Our analysis shows that (Section 2) during mismatch, L5 (z_t^{L5}) encodes zero input ($x_t \sim 0$), while L2/3 expects a positive input ($W_{L2/3 \rightarrow L5} z_t^{L2/3} > 0$), leading to a positive gradient for the self-supervised cost.

In Section 3-4 we show that although in our model, the reconstruction cost is specific to L5, the sign of the error in L2/3 will not be impacted if reconstruction error flows back to L2/3. This is because the gradient of reconstruction loss remains positive, maintaining the overall positive error in L2/3.

In section 5 we show that the gradient of the self-supervised cost w.r.t. L5 is negative. Finally in section 6, we show that the gradient of reconstruction loss has the opposite sign to the self-supervised cost for L5, but due to its smaller magnitude, the overall L5 error remains negative.

However, we also wanted to confirm that disregarding the reconstruction loss does not affect the results. To do so, we analyzed in simulations the gradients of the self-supervised and reconstruction cost with respect to both L2/3 and L5. Our analysis shows that propagating back the gradient of the reconstruction cost to L2/3 does not have a significant impact on the sign of mismatch responses in L2/3. Furthermore, the sign of the total gradient in L5 is dominated by the sign of the self-supervised cost function (we have attached to this response the derivations in detail). We also ran a simulation to visualise the sign of error in L5 during the mismatch period (see figure below).

Figure: Gradient analysis of self-supervised and reconstruction costs during the mismatch period. **a**, The gradient of the total cost $C_{total} = \lambda_p C_{L2/3 \rightarrow L5} + \lambda_r C_{L5}$ with respect to L5 neurons, computed using the default parameters learned during model training. The total gradient is primarily driven by the self-supervised cost gradient $C_{L2/3 \rightarrow L5}$, while the gradient of the reconstruction cost C_{L5} remains close to zero. **b**, The effect of varying the modulation factor α on the total cost gradient. The sign of the total gradient remains robust (negative) for different values of α , as long as the top-down input is greater than one, a condition characteristic of the mismatch period. **c**, The impact of scaling $W_{decoder}$ by a multiplicative factor. The total cost gradient with respect to L5 neurons remains robustly negative, indicating the stability of the mismatch-driven gradient dynamics (Note that Sigmoid activation function was used for this simulation, and the learning becomes unstable for larger values of $W_{decoder}$ scaling factor).

Further clarifications are provided in the main section "Model generates sensorimotor mismatch error signals consistent with experimental observations".

4. Error representations in L2/3 and L5 and sparsity. The above issues hint to a lack of description clarity of the model. Figures 1a and 1b only show the representation of the stimulus and its prediction, but a figure showing error neurons both in L2/3 and L5 is missing, say something akin to Supplementary Figure 13. These error neurons are a crucial element of the model, and should be shown in a sketch. Or better: they should be simulated, making it very clear that their activity represents the experimentally observed mismatch responses (see the above point – we doubt that the activity of these error neurons will be the derivatives in Eqs 10 and 11).

Related to this lack of clarity, it remains unclear how the sparsity statements have to be interpreted. According to the Methods and the text, sparsity only refers to the representational neurons in these layers. One can certainly assume that error neurons are silent, since after learning the stimulus is perfectly predicted. However, when adding noise (as in Figure 6e), those error neurons would certainly strongly reduce the sparsity (in contrast to the result in Figure 6e – where the noise should be quantified in percentage of the stimulus). Again, by declaring the mismatch errors being represented by the partial derivatives (Eqs 10 and 11), these error activities are likely not taken into account in the sparsity statements.

Apart from the above issue, it remains unclear why "layer-specific sparsity occurs naturally due to the proposed predictive function of L2/3 output", as explained in the Discussion. Do the authors now say that L2/3 is sparser because there, mostly errors are represented? This is not their model that does not consider the representation of errors in specific neurons. Or do the authors have another intuition why the representation of the prediction (L2/3) should be sparser than the representation of the stimulus (L5)? If the prediction is good, and the prediction task is not overly simple, the activity out of which the prediction is made, and predicted activity itself, should have similar sparsity, one may guess.

The rather obscure definition of the sparseness given in the Methods does not help (even this measure is taken from the literature with lots of citations). What would help, instead, is a histogram of activities of L2/3 and L5 neurons (that include error and representational neurons). Perhaps such an analysis would even be in favour of the model. As it is now, the quantitative comparison of model and data in Figure 6a is really bad (L2/3 is more than 4 times sparser than L5 according to the model, but perhaps 1.3 times sparser according to the data).

Reply: Thank you for highlighting this lack of clarity. As we clarify in the point above, unlike classical predictive coding approaches, our model does not separate representation and error neurons. Instead, each neuron generates activity during the forward pass, which is then used to compute its own error signals through gradient calculations. To make this distinction clear, we have added a new supplementary figure (new Fig. S1; see also above) comparing classical predictive coding models with our ANN-based approach. We also briefly address this clarification in the model introduction within the main text (lines 85–88).

Why L2/3 becomes sparser than L5: One possible explanation is that L2/3 is forced to extract and rely only on the most salient temporal features from the previous input in order to forecast the current latent state, whereas L5 is free to capture the full complexity, and more distributed representation of the input. In effect, L2/3 is tasked with filtering out the redundant (such as extraneous details or noise) or less informative parts of the previous encoding (by L4), which leads to a naturally sparser pattern of activation. It is important to note that the prediction is computed as $W_{L2/3 \rightarrow L5} z_t^{L2/3}$ and it is trained to match the full sensory representation of L5. To be precise, the sparsity computed here are for the neural activity within L2/3 $z_t^{L2/3}$, and not $W_{L2/3 \rightarrow L5} z_t^{L2/3}$. We now provide a shorter version of this explanation in lines 248-256.

Sparsity: As we clarify above, in our model, we do not have separate representation and error neurons. All neurons are used for representation, and over learning, have their own error signals (gradients). Since sparsity is calculated after learning, the errors (gradients) are near zero. To make this point clear, and as suggested by the reviewer, we now show the histograms of the activities and of the error signals (gradients) across all neurons after learning (see new Figure below).

Figure: Distribution of neural activity and sparsity across cortical layers. **a**, Kernel density estimate (KDE) plots showing the activation distributions of neurons in layers L2/3, L4, and L5. The inset displays the gradients (errors) with respect to neurons in each layer, illustrating their distribution across the network. **b**, Population sparsity of activations across layers, replicating the sparsity plot from Figure 6, panel (a) of the main paper. The bar plot quantifies the degree of sparsity in each layer, showing that L2/3 exhibits the highest activation sparsity, followed by L4 and L5 (cf. panel a).

Minor issues

1. Related to Major point 1: The intro claims that “..our model can learn Gabor-like inputs in contextual sequential tasks, highlighting its effectiveness in capturing complex patterns.”. Well, a Gabor-like input is not really a complex pattern, say as compared to a natural visual scene where an animal has to be detected. The latter is closer to what the visual is supposed to do. So that sentence should be adapted. It is shown that the model can learn to predict a sequence of MNIST digits. Yet, in principle, a single layer network with one digit as input can generate another digit as output too. Not clear, in what sense the cortical area with layered structure does genuinely more (in a computational sense). If so, this may also be added in Major point 1.

Reply: We acknowledge that Gabor-like inputs are not inherently complex patterns compared to natural visual scenes and revised the text accordingly (**lines 56-58**). We used Gabor patterns due to their simplicity, well-established role in experimental studies and their relevance to V1 neuronal encoding. MNIST was included as an example of a more complex temporal mapping. Regarding the ability of a single-layer network to map one digit to another, we agree that such transformations are possible in simpler

architectures. However, what our model allows is to learn these sequences in a self-supervised manner by leveraging on the proposed layered structure. In addition, as we discuss in Major Point 1 above, stacking up our model in a cortical hierarchy offers potential further computational benefits, such as allowing for predictions in increasingly more abstract representations. Furthermore, the layered structure in our model enables prediction to occur in latent space (representation space) rather than input space. Predictions in latent space can capture more informative features compared to those made in input space, aligning with findings from Balestrieri and LeCun (arXiv, 2024).

2. Related to Major point 4: "the model predicts layer-specific sparsity" appears all over, but in view of the factor 4 in the model and the factor 1.3 in the data, better to not emphasize this too much (see major issues – same for the various gradient statements).

Reply: Thank you for your feedback. The sentences were rephrased to indicate that the model captured qualitative differences in sparsity across layers.

"Moreover, the model captures the relative differences in sparsity across layers, aligning qualitatively with experimental findings in sensory systems." (lines 61-62) and "Our results capture qualitative differences in the level of sparsity, showing trends similar to experimental findings (Sakata et al. 2009)" (lines 236-237)

3. The sentence on line 145 somehow is incomplete.

Reply: Thank you for bringing this to our attention. The issue with the sentence was an incorrect "to" that should have been "we". We have now corrected it.

4. From Equation 2–5 the notation for the activities z changes several times with the upper and lower indices.

Reply: Yes, the subscripts and superscripts were indeed inconsistent—thank you for pointing this out. We now use superscripts to indicate the layer and subscripts to denote the timestep.

Appendix to reviewer responses

1 Definitions

1.1 Neural activity for each layer

$$z_{L4}^t = \sigma(W_{\text{Thal.} \rightarrow L4} \cdot x_t) \quad (1)$$

$$z_t^{L2/3} = \sigma(W_{L4 \rightarrow L2/3} \cdot z_{t-1}^{L4} + W_{td \rightarrow L2/3} \cdot I_t^{td}) \quad (2)$$

$$z_t^{L5} = \sigma(\alpha W_{L2/3 \rightarrow L5} \cdot z_t^{L2/3} + W_{\text{Thal.} \rightarrow L5} \cdot x_t) \quad (3)$$

where $0 < \alpha < 1$.

For readability, we denote

$$z_t^{L5-pre} = \alpha W_{L2/3 \rightarrow L5} \cdot z_t^{L2/3} + W_{\text{Thal.} \rightarrow L5} \cdot x_t \quad (4)$$

$$z_t^{L2/3-pre} = W_{L4 \rightarrow L2/3} \cdot z_{t-1}^{L4} + W_{td \rightarrow L2/3} \cdot I_{td}^t \quad (5)$$

Assumption In derivations, we assume weight matrices are predominantly positive matrix, which naturally emerges as result of gradient-based learning where the input are positive and neurons activations function are non-negative (e.g. Sigmoid and ReLU). In other words, weighted positive contributions (i.e. those entries where W_i is positive) outweigh the negative ones, therefore $Wx = \sum_{i=1}^{i=n} W_i x > 0$ for $x > 0$.

1.1.1 Cost functions

For simplicity, we assume $\lambda_p = \lambda_r = 1$.

$$\mathcal{C}_{\text{total}} = \underbrace{\mathcal{C}_{L2/3 \rightarrow L5}}_{\text{predictive}} + \underbrace{\mathcal{C}_{L5}}_{\text{reconstruction}} \quad (6)$$

$$\mathcal{C}_{L2/3 \rightarrow L5} = \frac{1}{2} (W_{L2/3 \rightarrow L5} \cdot z_t^{L2/3} - z_t^{L5})^2. \quad (7)$$

$$\mathcal{C}_{L5} = \frac{1}{2} (x_t - W_{\text{decoder}} \cdot z_t^{L5})^2. \quad (8)$$

2 Derivative of Self-Supervised Cost Function w.r.t $z_t^{L2/3}$

$$\frac{\partial \mathcal{C}_{L2/3 \rightarrow L5}}{\partial z_t^{L2/3}} = (W_{L2/3 \rightarrow L5} \cdot z_t^{L2/3} - z_t^{L5}) \frac{\partial (W_{L2/3 \rightarrow L5} \cdot z_t^{L2/3} - z_t^{L5})}{\partial z_t^{L2/3}}$$

now, let's derive $\frac{\partial (W_{L2/3 \rightarrow L5} \cdot z_t^{L2/3} - z_t^{L5})}{\partial z_t^{L2/3}}$ first.

$$\begin{aligned} \frac{\partial (W_{L2/3 \rightarrow L5} \cdot z_t^{L2/3} - z_t^{L5})}{\partial z_t^{L2/3}} &= \frac{\partial (W_{L2/3 \rightarrow L5} \cdot z_t^{L2/3})}{\partial z_t^{L2/3}} - \frac{\partial z_t^{L5}}{\partial z_t^{L2/3}} \\ &= W_{L2/3 \rightarrow L5} - \frac{\partial z_t^{L5}}{\partial z_t^{L2/3}} \\ &= W_{L2/3 \rightarrow L5} - \frac{\partial \sigma(\alpha W_{L2/3 \rightarrow L5} \cdot z_t^{L2/3} + W_{\text{Thal.} \rightarrow L5} \cdot x_t)}{\partial z_t^{L2/3}} \\ &= W_{L2/3 \rightarrow L5} - \left(\alpha \sigma'(\alpha W_{L2/3 \rightarrow L5} \cdot z_t^{L2/3} + W_{\text{Thal.} \rightarrow L5} \cdot x_t) \cdot W_{L2/3 \rightarrow L5} \right) \\ &= \left(1 - \alpha \sigma'(\alpha W_{L2/3 \rightarrow L5} \cdot z_t^{L2/3} + W_{\text{Thal.} \rightarrow L5} \cdot x_t) \right) \cdot W_{L2/3 \rightarrow L5} \\ &= \left(1 - \alpha \sigma'(z_t^{L5-\text{pre}}) \right) \cdot W_{L2/3 \rightarrow L5} \end{aligned}$$

Replacing $\frac{\partial (W_{L2/3 \rightarrow L5} \cdot z_t^{L2/3} - z_t^{L5})}{\partial z_t^{L2/3}}$ in $\frac{\partial \mathcal{C}_{L2/3 \rightarrow L5}}{\partial z_t^{L2/3}}$, we get

$$\begin{aligned} \frac{\partial \mathcal{C}_{L2/3 \rightarrow L5}}{\partial z_t^{L2/3}} &= (W_{L2/3 \rightarrow L5} \cdot z_t^{L2/3} - z_t^{L5}) \frac{\partial (W_{L2/3 \rightarrow L5} \cdot z_t^{L2/3} - z_t^{L5})}{\partial z_t^{L2/3}} \\ &= \left(W_{L2/3 \rightarrow L5} \cdot z_t^{L2/3} - z_t^{L5} \right) \left(\underbrace{\left(1 - \alpha \sigma'(z_t^{L5-\text{pre}}) \right) \cdot W_{L2/3 \rightarrow L5}}_{\Phi} \right) \end{aligned}$$

Where $\Phi \geq 0$ for Sigmoid and ReLU activation functions. Therefore, the sign of the $\frac{\partial \mathcal{C}_{L2/3 \rightarrow L5}}{\partial z_t^{L2/3}}$ only depends on the sign of the first term $(W_{L2/3 \rightarrow L5} \cdot z_t^{L2/3} - z_t^{L5})$

3 Derivative of Reconstruction Cost Function w.r.t $z_t^{L2/3}$

$$\frac{\partial \mathcal{C}_{L5}}{\partial z_t^{L2/3}} = \frac{\partial \mathcal{C}_{L5}}{\partial z_t^{L5}} \frac{\partial z_t^{L5}}{\partial z_t^{L2/3}}$$

The first term

$$\frac{\partial \mathcal{C}_{L5}}{\partial z_t^{L5}} = -W_{\text{decoder}}^\top (x_t - W_{\text{decoder}} z_t^{L5})$$

The second term

$$\begin{aligned} \frac{\partial z_t^{L5}}{\partial z_t^{L2/3}} &= \frac{\partial \sigma(\alpha W_{L2/3 \rightarrow L5} \cdot z_t^{L2/3} + W_{\text{Thal.} \rightarrow L5} \cdot x_t)}{\partial z_t^{L2/3}} \\ &= \alpha \sigma'(\alpha W_{L2/3 \rightarrow L5} \cdot z_t^{L2/3} + W_{\text{Thal.} \rightarrow L5} \cdot x_t) \cdot W_{L2/3 \rightarrow L5} \\ &= \alpha \sigma'(z_t^{L5 - \text{pre}}) \cdot W_{L2/3 \rightarrow L5} \end{aligned}$$

Substitute back into the main derivative

$$\frac{\partial \mathcal{C}_{L5}}{\partial z_t^{L2/3}} = -W_{\text{decoder}}^\top (x_t - W_{\text{decoder}} z_t^{L5}) \odot \alpha \sigma'(z_t^{L5 - \text{pre}}) \cdot W_{L2/3 \rightarrow L5} \quad (9)$$

where \odot represents the Hadamard product. The second term is positive, therefore, the sign of $\frac{\partial \mathcal{C}_{L5}}{\partial z_t^{L2/3}}$ is determined by the sign of the first term.

4 Effect of Reconstruction Loss on L2/3 Mismatch Responses

After learning converges, during mismatch interval, we halt the input while the top down is non-zero ($x_t = 0, I_t^{td} > 0$). Therefore, while L5 is encoding $x_t = 0$, L2/3 predicts an input greater than zero, as a result of the non-zero top-down input ($I_t^{td} > 0$). This results in $W_{L2/3 \rightarrow L5} \cdot z_t^{L2/3} > z_t^{L5}$ which causes a positive gradient flow to L2/3 from the $\mathcal{C}_{L2/3 \rightarrow L5}$ cost function. To analyze the sign of the gradient for \mathcal{C}_{L5} , we need to look at the signs for each term in Eq. 9.

Because of the modulation impact of L5 by L2/3 (with coefficient $0 < \alpha < 1$), we can show that $W_{\text{decoder}} \cdot z_t^{L5} > x_t$.

before mismatch : $W_{\text{decoder}} z_t^{L5} = W_{\text{decoder}} \sigma(\alpha W_{L2/3 \rightarrow L5} \cdot z_t^{L2/3} + W_{\text{Thal.} \rightarrow L5} \cdot x_t) \approx x_t$

after mismatch : $W_{\text{decoder}} z_t^{L5} = W_{\text{decoder}} \sigma(\alpha W_{L2/3 \rightarrow L5} \cdot z_t^{L2/3} + \Delta_{\text{pred}} + W_{\text{Thal.} \rightarrow L5} \cdot x_t) > x_t$

where the Δ_{pred} denote the over-prediction by L2/3 during mismatch interval. Therefore, during mismatch $W_{\text{decoder}} \cdot z_t^{L5} > x_t$, hence the sign of the gradient for \mathcal{C}_{L5} with respect to L2/3 neurons is also positive.

In summary, during mismatch, we have

$$W_{L2/3 \rightarrow L5} \cdot z_t^{L2/3} > z_t^{L5} \Rightarrow W_{L2/3 \rightarrow L5} \cdot z_t^{L2/3} - z_t^{L5} > 0$$

$$W_{\text{decoder}} \cdot z_t^{L5} > x_t \Rightarrow x_t - W_{\text{decoder}} \cdot z_t^{L5} < 0$$

Therefore

$$\frac{\partial \mathcal{C}_{L2/3 \rightarrow L5}}{\partial z_t^{L2/3}} > 0 \quad \& \quad \frac{\partial \mathcal{C}_{L5}}{\partial z_t^{L2/3}} > 0 \Rightarrow \frac{\partial \mathcal{C}_{\text{total}}}{\partial z_t^{L2/3}} > 0$$

5 Derivative of the Self-Supervised Cost Function with Respect to z_t^{L5}

First, consider the derivative of the self-supervised cost term, $\mathcal{C}_{L2/3 \rightarrow L5}$, with respect to z_t^{L5} :

$$\begin{aligned} \frac{\partial \mathcal{C}_{L2/3 \rightarrow L5}}{\partial z_t^{L5}} &= \left(W_{L2/3 \rightarrow L5} \cdot z_t^{L2/3} - z_t^{L5} \right) \frac{\partial \left(W_{L2/3 \rightarrow L5} \cdot z_t^{L2/3} - z_t^{L5} \right)}{\partial z_t^{L5}} \\ &= - \left(W_{L2/3 \rightarrow L5} \cdot z_t^{L2/3} - z_t^{L5} \right). \end{aligned}$$

During periods of mismatch, we have

$$W_{L2/3 \rightarrow L5} \cdot z_t^{L2/3} > z_t^{L5},$$

so the gradient above is negative.

6 Derivative of the Reconstruction Cost Function with Respect to z_t^{L5}

The reconstruction cost function \mathcal{C}_{L5} has the derivative:

$$\frac{\partial \mathcal{C}_{L5}}{\partial z_t^{L5}} = - \left(x_t - W_{\text{decoder}} \cdot z_t^{L5} \right) \cdot W_{\text{decoder}}.$$

Given that

$$W_{\text{decoder}} \cdot z_t^{L5} > x_t \quad \Rightarrow \quad x_t - W_{\text{decoder}} \cdot z_t^{L5} < 0,$$

the gradient of the reconstruction loss with respect to z_t^{L5} is positive.

7 Comparison of the Two Gradient Terms for z_t^{L5}

It is important to note that the gradients with respect to z_t^{L5} from the self-supervised cost and the reconstruction cost have opposite signs. However, our

analysis shows that the magnitude of the self-supervised gradient exceeds that of the reconstruction gradient. Consequently, the overall gradient is dominated by the self-supervised component.

During a mismatch, the reconstruction error arises from the modulation effect of layer L2/3 on L5. Denote the modulated activity by

$$\Delta_{\text{pred}} = \alpha W_{L2/3 \rightarrow L5} \cdot z_t^{L2/3}.$$

The corresponding reconstruction error is then given by $W_{\text{decoder}} \Delta_{\text{pred}}$, so that

$$\frac{\partial \mathcal{C}_{L5}}{\partial z_t^{L5}} = W_{\text{decoder}}^\top \Delta_{\text{pred}} W_{\text{decoder}}.$$

Thus, the overall sign of the total gradient depends on the relative magnitudes of the error terms from the self-supervised and reconstruction losses. Specifically, we have:

$$\text{if } W_{\text{decoder}}^\top \Delta_{\text{pred}} W_{\text{decoder}} > W_{L2/3 \rightarrow L5} \cdot z_t^{L2/3} - z_t^{L5}, \quad \text{then } \frac{\partial \mathcal{C}_{\text{total}}}{\partial z_t^{L5}} > 0; \quad (10)$$

$$\text{if } W_{\text{decoder}}^\top \Delta_{\text{pred}} W_{\text{decoder}} < W_{L2/3 \rightarrow L5} \cdot z_t^{L2/3} - z_t^{L5}, \quad \text{then } \frac{\partial \mathcal{C}_{\text{total}}}{\partial z_t^{L5}} < 0. \quad (11)$$

In our simulations, the gradient with respect to z_t^{L5} is consistently negative, indicating that the self-supervised gradient dominates the reconstruction loss gradient. This outcome is attributed to factors such as a small value of α (with $0 < \alpha \ll 1$) and the use of a sigmoid activation function. Additional simulations have been conducted to identify regimes in which the sign of the total gradient may reverse.

7.1 Simulation of Gradients w.r.t z_t^{L5}

Figure 1: Gradient analysis of self-supervised and reconstruction costs during the mismatch period **a**, The gradient of the total cost $C_{total} = \lambda_p C_{L2/3 \rightarrow L5} + \lambda_r C_{L5}$ with respect to L5 neurons, computed using the default parameters learned during model training. The total gradient is primarily driven by the self-supervised cost gradient $C_{L2/3 \rightarrow L5}$, while the gradient of the reconstruction cost C_{L5} remains close to zero. **b**, The effect of varying the modulation factor α on the total cost gradient. The sign of the total gradient remains robust (negative) for different values of α , as long as the top-down input is greater than one, a condition characteristic of the mismatch period. **c**, The impact of scaling $W_{decoder}$ by a multiplicative factor. The total cost gradient with respect to L5 neurons remains robustly negative, indicating the stability of the mismatch-driven gradient dynamics (Note that Sigmoid activation function was used for this simulation, and the learning becomes unstable for larger values of $W_{decoder}$ scaling factor).